# TimeSeAD: Benchmarking Deep Multivariate Time-Series Anomaly Detection

**Dennis Wagner**\*                                        *dwagner@cs.uni-kl.de*
*RPTU Kaiserslautern-Landau*

**Tobias Michels**\*                                        *tmichels@cs.uni-kl.de*
*RPTU Kaiserslautern-Landau*

**Florian C.F. Schulz**                                *florian.cf.schulz@tu-berlin.de*
*TU Berlin*

**Arjun Nair**                                                *naira@rptu.de*
*RPTU Kaiserslautern-Landau*

**Maja Rudolph**                                        *maja.rudolph@us.bosch.com*
*Bosch AI*

**Marius Kloft**                                            *kloft@cs.uni-kl.de*
*RPTU Kaiserslautern-Landau*

**Reviewed on OpenReview:** *https://openreview.net/forum?id=iMmsCIOJsS*

## Abstract

Developing new methods for detecting anomalies in time series is of great practical significance, but progress is hindered by the difficulty of assessing the benefit of new methods, for the following reasons. (1) Public benchmarks are flawed (e.g., due to potentially erroneous anomaly labels), (2) there is no widely accepted standard evaluation metric, and (3) evaluation protocols are mostly inconsistent. In this work, we address all three issues: (1) We critically analyze several of the most widely-used multivariate datasets, identify a number of significant issues, and select the best candidates for evaluation. (2) We introduce a new evaluation metric for time-series anomaly detection, which—in contrast to previous metrics—is recall consistent and takes temporal correlations into account. (3) We analyze and overhaul existing evaluation protocols and provide the largest benchmark of deep multivariate time-series anomaly detection methods to date. We focus on deep-learning based methods and multivariate data, a common setting in modern anomaly detection. We provide all implementations and analysis tools in a new comprehensive library for `Time Series Anomaly Detection`, called `TimeSeAD`[1].

## 1 Introduction

Anomaly detection (AD) on time series is a fundamental problem in machine learning and significant in various applications, from monitoring patients and uncovering financial fraud to detecting faults in manufacturing and critical process conditions in chemical plants (Ruff et al., 2021). The aim in AD is to automatically identify significant deviations to the norm—so-called *anomalies*. There are two principal approaches to AD on time series: Either an anomaly detector assigns a score to each time step separately (point-wise) or the entire time series (globally). This work focuses on unsupervised AD in the point-wise setting, which has

---

\*These authors contributed equally.
[1]https://github.com/wagner-d/TimeSeAD

become the standard in the literature and can easily be adapted to the global environment by aggregating the local labels. Moreover, point-wise methods lend themselves naturally to real-time prediction and anomaly localization, which are of practical importance in many applications.

Evaluating the accuracy of an anomaly detector for time series is not straightforward. Most authors commonly rely on point-wise metrics, particularly the $F1$-score (Choi et al., 2021; Audibert et al., 2022). However, by definition, point-wise metrics ignore any temporal dependencies in time series and, consequently, fail to distinguish predictive patterns, such as early and late predictions. The few prior attempts to introduce specialized evaluation metrics (Lavin & Ahmad, 2015; Tatbul et al., 2018) for time-series AD have not caught on in the community, primarily due to their complexity and the counterintuitive results they can produce (Huet et al., 2022). We explicitly discuss existing metrics for time series anomaly detection in Section 3.2. Another major problem in time-series AD is the datasets. Recently, Wu & Keogh (2021) have exposed significant flaws in many widely used univariate datasets, ranging from surface-level issues (such as mislabeled points and an unrealistic density of anomalies) to deep-rooted problems (such as positional bias and trivial features). These problems raise doubts about the reliability of existing evaluations obscuring the actual progress in the field. The analysis by Wu & Keogh (2021) does not address all critical aspects of time-series AD, such as distributional shift. Furthermore, many modern applications of AD reside in the multivariate regime. To the best of our knowledge, this is the first comprehensive analysis of many popular multivariate times-series AD datasets. The third crucial problem impeding progress in the field is the diverse and often incompatible evaluation protocols across publications. Many recent methods compare to previously reported results while using different evaluation protocols. Differences range from the way methods are trained, optimized, tuned, or evaluated, making the results inherently incomparable. The general lack of detailed specification of the evaluation protocol and methods, and official implementations additionally burden fair comparisons. Furthermore, common to many evaluations, some practices have already been proven to introduce bias and skew the results in favor of random predictions (Kim et al., 2022; Doshi et al., 2022).

The distinct flaws of datasets, metrics, and evaluation protocols are pervasive problems subverting evaluations in time-series AD, making it hard to determine the actual progress in the field. This work examines the most popular multivariate time-series datasets, evaluation metrics, and protocols in detail and proposes a general evaluation framework to address the identified problems. We have created a detailed, extendable, and user-friendly library, where we implemented 28 deep-learning based multivariate time-series AD methods. This library is an unprecedented asset enabling researchers to quickly and reliably develop, test, and evaluate new methods. It provides a set of tools to analyze datasets and methods alike. In our evaluation, we focus entirely on multivariate datasets and deep-learning based methods, a common setting in modern applications (Darban et al., 2022). Multivariate data are more complex than univariate data, allowing for complex dependencies between features, and deep learning based methods have been shown to outperform shallow baselines on multivariate data in multiple settings (LeCun et al., 2015).

Our main contributions are the following:

- We conduct a **thorough analysis** of the most widely used **datasets, metrics, and evaluation protocols** for multivariate time-series AD, revealing significant problems with all three.

- We propose a **new evaluation metric** that is provable *recall consistent* (a property we define in section Section 3.2) and empirically provides a reasonable ordering of evaluated methods.

- We present the **largest comprehensive benchmark so far** for multivariate time-series AD, comparing 28 deep-learning methods on 21 datasets.

## 2 Related Work

Several papers have attempted to summarize the vast number of time-series AD approaches (Darban et al., 2022). However, most prior work focuses either on a subclass of network architectures (Lindemann et al., 2021; Lee et al., 2021; Wen et al., 2022) or a specific application domain and methods specifically applied therein (Luo et al., 2021). Others discuss multiple methods and concepts in a high-level overview (Blázquez-García et al., 2021), with a strong focus on application. Choi et al. (2021) and Audibert et al. (2022) use

point-wise metrics to selectively evaluate nine and 14 methods, respectively, on three and five datasets, in which we identified several problems (see Section 3.1). Similarly, Lai et al. (2021) evaluate nine methods on four other datasets, one of which is not available anymore, using point-wise methods. Schmidl et al. (2022) evaluate a large collection of more than 20 deep and several shallow methods primarily on univariate or low-dimensional datasets. Their evaluation relies on a slow (quadratic in time) implementation of time-series precision and recall (Tatbul et al., 2018). Consequently, they had to exclude results where the computation took too long. Other libraries, such as (Bhatnagar et al., 2021), mainly focus on shallow or basic deep methods.

We expand the work of Wu & Keogh (2021) to analyze multivariate datasets. They thoroughly analyzed several of the most popular univariate time-series datasets, identified multiple flaws, and concluded that many datasets do not guarantee a fair evaluation of AD algorithms. Following their work, we find several similar problems with the most widely used multivariate time-series datasets.

Overcoming the inherent problems of point-wise metrics on time-series data is not an easy task. Huet et al. (2022) provide an overview of existing attempts and introduce a metric based on the distance of predicted anomaly windows to the nearest anomaly. However, they report their results on datasets in which we identify several problems. Others modify the predictions before evaluation (Xu et al., 2018; Scharwächter & Müller, 2020; Kim et al., 2022) or consider only the beginning of anomaly windows (Doshi et al., 2022). Some metrics are clearly biased towards extreme cases of anomaly detectors (Hundman et al., 2018). Lavin & Ahmad (2015) introduced the first metric to directly address the problems of point-wise evaluations by penalizing late predictions in anomaly windows. With its numerous hyperparameters, the metric was too complex and variable to be widely adapted (Xu et al., 2018). Tatbul et al. (2018) proposed time-series precision and recall, a generalization of previous concepts in many ways. For a long time, the only publicly available implementations were too slow and cumbersome to use in practice. Lastly, Garg et al. (2021) propose a variation of time-series precision and recall that ignores any overlap between prediction and anomalies in the recall.

## 3 The Illusion of Progress

In this section, we uncover several issues that make evaluations in (multivariate) time-series AD unreliable and thus create an illusion of progress in the field. Our analysis first examines some of the most commonly used datasets. These datasets are the backbone of time-series AD evaluation and have been used in virtually all major comparisons in the field (Schmidl et al., 2022; Garg et al., 2021; Choi et al., 2021; Jacob et al., 2020). Our analysis reveals several significant flaws in these datasets. Second, we investigate the shortcomings of frequently used evaluation metrics, particularly the point-wise $F1$-score and its adaptations. Lastly, we examine the inconsistencies and other problems within established evaluation protocols.

### 3.1 Datasets

A good dataset for benchmarking deep time-series AD methods should be as unbiased as possible and adhere to the underlying assumptions of AD (Ruff et al., 2021), but should also be complex enough to represent a significant challenge. It is impossible to determine exact statistics of a good benchmark dataset, but we can look at similar settings to gain some intuition. Time-series data contain strong temporal dependencies similar to spatial dependencies in images, where deep-learning based methods have outclassed most traditional approaches [cite]. Thus, a good benchmark dataset should contain a similar amount of samples to facilitate the training of deep models. To present a significant challenge, a good benchmark dataset should contain enough features to allow for complex inter-feature dependencies. In the following, we highlight several problems found in time-series datasets and summarize our findings on SWaT (Goh et al., 2016), WADI (Ahmed et al., 2017), SMAP, and MSL (Hundman et al., 2018).

**Anomaly density** refers to the fraction of anomalies in the test set. Anomalies are usually considered rare deviations, and their density in the test set should reflect this characterization. However, except for WADI, all considered datasets contain more than 10% anomalies, which might already be too high to be considered rare deviations from the norm.

**Positional bias** is introduced when the distribution of the relative positions of anomalies deviates significantly from a uniform distribution. For example, anomalies can be biased towards the end when an anomaly means a fatal error for the generating process, known as run-to-failure bias (Wu & Keogh, 2021). Algorithms accounting for this shift have an immediate advantage over any competitors. To investigate this bias, we examine the relative positions of anomalous time steps in each time series in the test sets and find clear evidence of positional bias in both SMAP and MSL (for example, see Figure 1a).

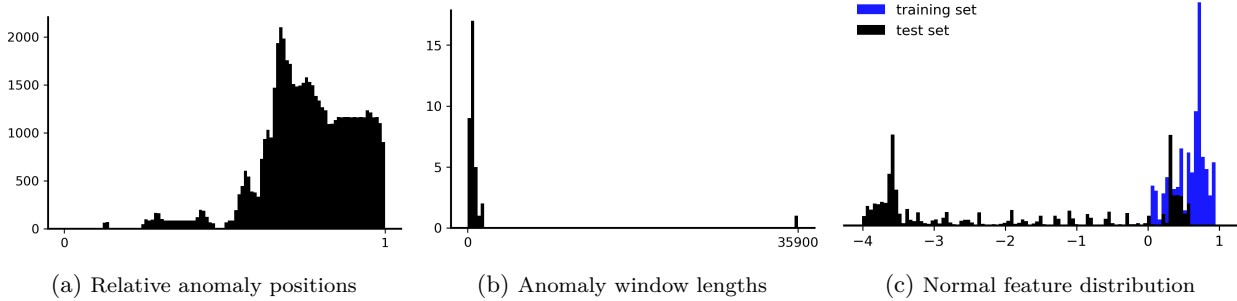

(a) Relative anomaly positions      (b) Anomaly window lengths      (c) Normal feature distribution

Figure 1: (a) The relative positions of anomalies in the test set of SMAP show clear positional bias towards the latter half of the time series. (b) The distribution of anomaly window lengths in SWaT shows the existence of exceptionally long anomaly windows. (c) Comparing the distribution of normal time steps of feature AIT201 in SWaT reveals clear signs of distributional shift.

**Long anomalies** can introduce problems in the evaluation. Some methods may rely on normal context in each window to predict subsequent anomalies and are thus at a disadvantage when long anomaly windows occur. Long anomalies also interact with adapted evaluation protocols, which we discuss in Section 3.3. Although not inherently negative, both effects should be kept in mind when using data containing long anomaly windows for evaluation. We found that the vast majority of anomalous time steps in all datasets belong to one or several long anomalies (for example, see Figure 1b).

**Constant features** appear in all considered datasets. Features that are constant only in the training or test set are generally desirable. However, some datasets contain features that remain constant across training and test set. While such features may be valuable in practical applications, they add unnecessary complexity to the benchmark.

**Distributional shift** occurs when the underlying process that generated normal training and test data is not the same. It breaks one of the fundamental assumptions of AD. For several datasets, we can examine distributional shift by simply inspecting their feature means and standard deviations (for example, see Figure 1c). Furthermore, anomalies should be labeled consistently where they occur in the data to ensure an unbiased and fair evaluation. Effects that show in a sensor only after the labeled anomaly (for example, see Figure 2a) pose an impossible problem for any anomaly detector. This holds especially true for long-lasting changes in the data. Some anomalies seem to permanently change the distribution of the system, causing clear distributional shift (for example, see Figure 2b).

### 3.1.1 Analyzing SWaT, WADI, SMAP, and MSL

In the following, we summarize our analysis of four of the most widely used datasets for multivariate time-series AD. We provide descriptions, detailed examples, and discussions for all datasets in Appendix B.

**SWaT and WADI** contain the clearest examples of delayed and long-term effects in the data. In both datasets, the distribution changes drastically in the second half of the test set. Additionally, we found exceptionally long anomalies in both datasets, especially in SWaT, where one anomaly spans nearly 36,000 time steps. Even if the former issue was addressed by experts, this could introduce even more anomalous time steps, longer anomaly windows, and positional bias. Thus we conclude that evaluations on these two datasets are highly unreliable and that these datasets are not suited for multivariate time-series AD evaluation.

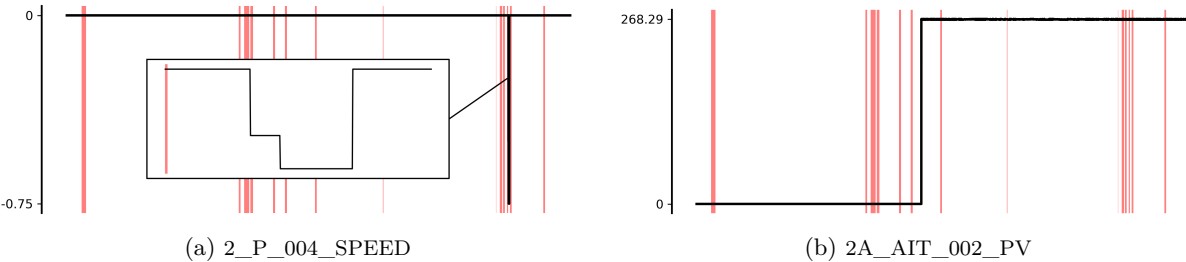

(a) 2_P_004_SPEED  (b) 2A_AIT_002_PV

Figure 2: Two features from the test set of WADI show where anomalies seem to cause (a) delayed or (b) long-term effects in the data. Red-shaded areas are ground truth anomalies. The feature in (b), normalized to range in $[0, 1]$ on the training set, jumps to unprecedented values on the test set.

**SMAP and MSL** contain time series with one feature representing a sensor measurement, while the rest represent binary encoded commands. The command features are often constant, particularly in sections where anomalies occur. Furthermore, since several sensors have been used to construct the dataset, each time series in both datasets should be considered independently. SMAP contains a clear positional bias towards the end, and both seem to contain significant distributional shifts caused by anomalies. Thus, we conclude that both MSL and SMAP are also not suited for general time-series AD evaluation.

### 3.2 Metrics

An anomaly detector produces an anomaly score for each time step in a time series. The higher the score at time $t$, the more confident the detector is that the point at that time is an anomaly. Anomalies are then predicted by thresholding these scores. Given predictions and labels, an evaluation metric produces a score based on their agreement. Different algorithms can then be compared based on the scores produced by their predictions. A good metric should not be unintentionally biased towards a specific group of bad algorithms, for example random predictions. Evaluation on time-series data is complicated due to the temporal dependencies between time steps, as anomalies often appear in intervals and two predictors might differ in the pattern of their predictions inside these anomaly windows. In the following, we examine the shortcomings of point-wise metrics and existing attempts to explicitly include the temporal dependency in evaluation metrics.

**Predictive patterns** describe the patterns of the predictions of an anomaly detector. Two methods making the same amount of predictions can still differ significantly depending on their predictive pattern. Predictive patterns matter for separating early and late or consistent and fragmented predictions, and their differences should be reflected in the metric used to compare them. Many proposed metrics ignore the predictive patterns for their computation. This is particularly true for point-wise metrics that consider each prediction separately. As a consequence, any two methods that differ only in a predictive pattern on anomalies are indistinguishable for any point-wise metric (see Figure 3a as an example). However, most papers rely on the point-wise $F1$-score for their evaluation, oftentimes reported alongside precision and recall.

**Recall consistency** refers to the monotonicity of point-wise recall with respect to the threshold of the evaluated anomaly detector, that is, the point-wise recall is monotonically decreasing with an increasing threshold. We argue that any derived metric replicating the intuition of recall should be recall consistent to avoid unexpected and unintuitive behavior. Such behavior can even lead to problems when computing aggregated metrics that assume recall consistency. Time-series precision and recall (Tatbul et al., 2018) is an attempt to incorporate predictive patterns in the computation of recall and precision. Consider the set of anomaly windows in a dataset $\mathcal{A}$, the set of predicted windows $\mathcal{P}$, and the set of predicted windows overlapping with a set $\mathcal{P}_A = \{P \in \mathcal{P} \mid |A \cap P| > 0\}$. Then time-series recall is defined as

$$TRec(\mathcal{A}, \mathcal{P}) = \frac{1}{|\mathcal{A}|} \sum_{A \in \mathcal{A}} \left[ \alpha \mathbb{1}(|\mathcal{P}_A| > 0) + (1-\alpha)\gamma(|\mathcal{P}_A|) \sum_{P \in \mathcal{P}} \frac{\sum_{t \in P \cap A} \delta(t - \min A, |A|)}{\sum_{t \in A} \delta(t - \min A, |A|)} \right], \qquad (1)$$

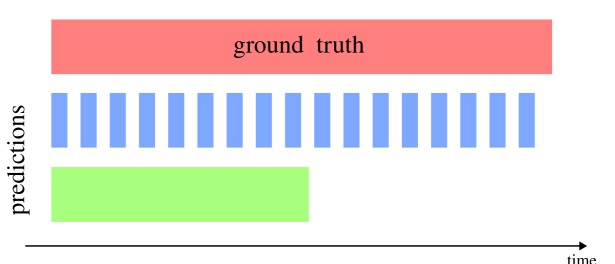

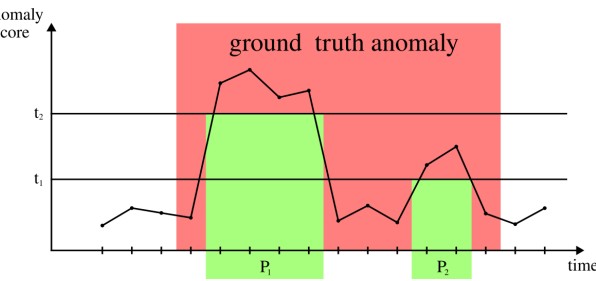

(a) Two predictions of equal size with distinct predictive patterns on an anomaly.

(b) Anomaly score and predictions $P_1, P_2$, where only the larger persists for both thresholds $t_1, t_2$.

Figure 3: (a) A point-wise metric cannot distinguish between two methods that differ only in their predictive pattern on anomalies. (b) Counterintuitively, $TRec$ with a constant bias and $\gamma(x) = x^{-1}$ increases when the threshold increases from $t_1$ to $t_2$.

with weight $0 \leq \alpha \leq 1$, monotone decreasing cardinality function $\gamma$ with $\gamma(1) = 1$, and bias function $\delta \geq 1$. This metric is not recall consistent in general, in particular for the recommended default parameter choices ($\gamma(x) = x^{-1}$) (Tatbul et al., 2018). Increasing the threshold and removing an entire window from the predictions will increase $\gamma(|\mathcal{P}_A|)$ which may "override" the decrease in the following sum that quantifies the overlap if $\gamma$ is not chosen carefully. For example, consider two disjoint predictions $P_1, P_2 \subset A \in \mathcal{A}$ for a threshold $\lambda$, such that $\sum_{t \in P_1 \cap A} \delta(t - \min A, |A|) > \sum_{t \in P_2 \cap A} \delta(t - \min A, |A|)$. Then, if there exists a threshold greater than $\lambda$ such that $P_1$ is kept intact while $P_2$ vanishes, $TRec$ increases (see Figure 3b for illustration).

**Implicit bias** is at the core of any evaluation metric defining the ideal behavior to strive for. Thus each metric needs to be carefully designed, to not encourage unwanted behavior by accident or introduce conflicting goals. Consider the precision associated with $TRec$ that is computed by interchanging the rolls of anomalies and predictions, i.e., $TPrec(\mathcal{A}, \mathcal{P}) = TRec(\mathcal{P}, \mathcal{A})$. This choice encourages algorithms to predict many small anomaly windows, such that the negative impact of falsely anomalous predictions is diminished since all predictions are weighted equally. The resulting behavior conflicts with the choice of a decreasing cardinality function, which encourages the opposite.

**Compensating flaws** generally introduces further, often subtle, bias into the evaluation. For many applications, precise predictions are important, for example, where false positives cause severe overhead. Introducing soft boundaries for anomalies to account for predictions only nearly missing ground truth anomaly windows erodes the required preciseness of predictions. Depending on the particular setting, such leeway might be desired behavior, but even then should be inserted conscious of its implicit biases. In a well-labeled dataset, each anomaly window should indicate where alarms are expected and acceptable. Thus, such issues should not be addressed in the metric, but rather in the dataset.

### 3.2.1 Analyzing evaluation metrics for time series

We discuss and analyze recent evaluation metrics for time-series data with respect to the identified flaws.

**Soft-boundary metrics**, such as distance-based precision and recall Huet et al. (2022) and range-based volume under surface metrics by Paparrizos et al. (2022), compensate for predictions outside of anomaly windows by relying on the distance between predictions and anomalies or extending anomaly windows. While the former introduces implicit bias on correct predictions towards the center of anomaly windows and makes no explicit distinction between predictions before or after anomalies, the latter carries over the indifference to predictive patterns of point-wise methods. Similarly, Scharwächter & Müller (2020) extend anomaly window and predicted windows in the computation of point-wise precision and recall, respectively.

**Early prediction metrics**, such as the NAB score (Lavin & Ahmad, 2015) and sequence precision delay (Doshi et al., 2022), only consider the first anomalous prediction after the beginning of an anomaly window. This emphasis on early predictions is largely motivated by specific applications, where early detection

of anomalies is vital. However, by ignoring the remaining predictions, these metrics cannot distinguish predictive patterns and thus distinguish subtle differences between methods.

**Time-series precision and recall** and their derived metrics solve most problems of point-wise metrics and provide an intuitive set of adjustable parameters. However, they suffer from recall inconsistency and conflicting implicit bias. Even though clearly flawed, they constitute a reasonable attempt to include the temporal structure of time series in an evaluation metric. Other attempts to adjust precision and recall for time series contain clear bias towards constant predictors (Hundman et al., 2018) or fail to account for predictive patterns similar to point-wise metrics (Garg et al., 2021).

### 3.3 Evaluation Protocol

An evaluation protocol comprises specifications of how the experiments are conducted, including the preprocessing of datasets, feature elimination, and parameter selection heuristics. Consistent evaluation protocols are necessary to guarantee fair comparison across different models. In the following, we outline some of the problems we identify in and around evaluation protocols in the literature.

**Point adjustment** is a technique modifying the predictions to complement the point-wise $F1$-score. Any anomaly window with at least one correctly predicted time step is considered predicted correctly. However, even random methods have a decent chance to predict at least one point in larger anomaly windows, where they can easily reach the performance of most complex methods or even outperform them (Kim et al., 2022; Doshi et al., 2022). Despite these flaws, this technique was adopted by many papers (Su et al., 2019; Audibert et al., 2020; Zhao et al., 2020; Zhang et al., 2021; Xiao et al., 2021; Chen et al., 2021; Wang et al., 2021; Challu et al., 2022; Hua et al., 2022; Chambaret et al., 2022; Zhang et al., 2022b;a). In light of our discussion on evaluation metrics and their apparent flaws, this technique should generally be abandoned for evaluations of time-series AD.

**Implementations and specifications** provided by the original authors are an important tool to ensure reproducibility. This becomes especially important when the evaluation protocols need to be adapted, or the methods need to be evaluated on new datasets or with different metrics. However, several works do not specify their evaluation protocol, hyperparameters, and architecture in enough detail to reproduce their results and do not publish any source code that contains them (e.g., Homayouni et al., 2020; Pereira & Silveira, 2018). As the employed evaluation protocol can significantly affect the final performance, a functioning implementation should be completely disclosed alongside an evaluation.

**Other potential inconsistencies** can be found across multiple evaluation protocols. Some papers seem to tune the model parameters on the test set, introducing more bias into the evaluation (Zhan et al., 2022). Others report aggregated metrics over datasets consisting of samples from different distributions or aggregated over multiple datasets (Su et al., 2019). Such metrics can be of interest in their own right, however, without a clear definition of how these aggregated values are computed or additional analysis, these evaluations often lack clarity, comparability, and reproducibility. We encountered many more minor inconsistencies, highlighting the importance of official implementations and thorough specifications.

## 4 TimeSeAD: Benchmarking Deep Multivariate Time-Series AD

In this section, we propose how to benchmark time-series AD methods in a way that mitigates the issues discussed in Section 3. We discuss the strengths and weaknesses of two recent datasets, and how their flaws can be mitigated. Further, we introduce modified versions of time-series precision and recall, to alleviate the biases of the precision and ensure recall consistency of the recall. Finally, we discuss our evaluation protocol and implementation.

### 4.1 Datasets

In the previous section, we uncovered flaws in several commonly used benchmark datasets, making them unfit for evaluation. However, there are also datasets we find more suited for benchmarking, namely SMD (Su et al., 2019) and Exathlon (Jacob et al., 2020).

**SMD** contains 28 time series generated from different processes and thus comprises 28 datasets with 38 features each. Some of its datasets suffer from distributional shift and have been removed from evaluations in the past (Li et al., 2021b). Detecting distributional shift in time-series data is no trivial task in itself. Therefore, we rely on manual inspection of all datasets in this study. We exclude several datasets from the final evaluation, where we suspect delayed or long-term effects caused by anomalies, and only report those results in Appendix E. In total, we remove 13 datasets leaving 15 datasets for evaluation.

**Exathlon** comprises eight datasets collected from applications run on a cluster. The time series in Exathlon suffer from missing values, which the creators suggest be replaced with default values. This inadvertently injects unlabeled anomalies in the data, where the default values follow a different distribution. Instead, we replace any missing values with the respective preceding value. We omit two applications, one, for which we identify a severe distributional shift, and one with a too-small test set, leaving six datasets. Overall, we find several more instances of possible delayed effects and distributional shift, which might be attributed to background effects. Nonetheless, we strongly encourage further careful inspection by application experts, especially to address the high anomaly density in all datasets in Exathlon.

All datasets in SMD and Exathlon are far from ideal benchmark datasets. Considering each time series individually leaves each dataset with fewer samples compared to other datasets, and the high anomaly density in Exathlon is worrying. While not perfect, these datasets are significantly better alternatives to SWaT, WADI, SMAP, and MSL for evaluating time-series AD methods.

## 4.2 Metrics

In Section 3.2 we discussed the potential and shortcomings of $TRec$ and $TPrec$. We propose new default parameters for $TRec$ and a variation of $TPrec$ to address their flaws. Let us first note the discrepancy between the two terms in Equation (1). The first term counts the number of anomaly windows for which at least one point was predicted correctly. In contrast, the second term is entirely concerned with the predictive structure within each anomaly window. Since the first term is completely oblivious to the size of the anomalies, the range of both terms could vary wildly between tasks, and the terms would need to be balanced for each task individually. Furthermore, the second term already implicitly acknowledges the existence of anomalies in their overlap. Thus, we suggest using $\alpha = 0$.

To prevent unintuitive results caused by recall inconsistency, we further require the cardinality function to guarantee recall consistency. Thus, we define a suitable class of cardinality functions for which recall consistency always holds.

**Theorem 1** $TRec$ is recall consistent for any cardinality function of the form

$$\gamma(1, A) = 1, \quad \gamma(n, A) = \max_{0 < m < n} \frac{\sum\limits_{t \in A} \delta(t - \min A, |A|) - n + m}{\sum\limits_{t \in A} \delta(t - \min A, |A|)} \gamma(m, A).$$

**Proof - Sketch[2]:** It is straight-forward to verify the monotonicity of $\gamma$. Thus, it suffices to show the recall-consistency of the resulting $TRec$. To ensure monotonicity of $TRec$, we only need to proof the monotonicity for each individual ground-truth anomaly window. Consider the predictions $\mathcal{P}_A, \mathcal{P}'_A$ on an anomaly $A \in \mathcal{A}$ for two thresholds $\lambda < \lambda'$, such that the cardinality of the predictions decreases with an increasing threshold, i.e. $|\mathcal{P}_A| > |\mathcal{P}'_A|$. Then the term in Equation (1) corresponding to $A$ is non-increasing, if the following holds

$$\frac{\sum\limits_{P \in \mathcal{P}'_A} \sum\limits_{t \in P \cap A} \delta(t - \min A, |A|)}{\sum\limits_{P \in \mathcal{P}_A} \sum\limits_{t \in P \cap A} \delta(t - \min A, |A|)} \leq \frac{\sum\limits_{t \in A} \delta(t - \min A, |A|) - (|\mathcal{P}_A| - |\mathcal{P}'_A|)}{\sum\limits_{t \in A} \delta(t - \min A, |A|)} \leq \frac{\gamma(|\mathcal{P}_A|, A)}{\gamma(|\mathcal{P}'_A|, A)}.$$

If this holds for all cardinalities smaller than the initial prediction, the statement holds, since recall consistency with respect to the threshold is equivalent to the term in Equation (1) corresponding to $A$ being non-increasing with respect to the cardinality of the predictions.

---

[2]We provide the detailed proof in Appendix A.

This definition of suitable cardinality functions depends on the choice of bias function. A particularly important choice is the constant bias, allowing the metric to be readily applied to most settings. Thus, the following theorem shows the closed-form solution of its corresponding cardinality function.

**Theorem 2** With constant bias the cardinality function has the closed-form solution

$$\gamma^*(n, A) = \left(\frac{|A| - 1}{|A|}\right)^{n-1}.$$

**Proof - Sketch[2]:** By an inductive argument, using the fact that the maximum over a set is larger or equal to all its elements, we can show that $\gamma$ is lower bounded by $\gamma^*$ for all cardinalities. By a similar argument, rewriting the maximum using a technical lemma, we can show $\gamma$ is upper bounded by $\gamma^*$ for all cardinalities. Combining both facts yields the statement.

We call $TRec$ with cardinality function $\gamma^*$ and constant bias $TRec^*$. While this gives an easy-to-compute metric, the general formulation preserves the bias function as a tunable parameter. It is important to retain this degree of generality in the definition, such that we can still adapt the metric to specific use cases, such as early prediction.

Finally, we address the bias of time-series precision. In the definition of $TPrec$, each prediction is weighted by the inverse of the cardinality of the predictions $|\mathcal{P}|^{-1}$. Instead of using equal weights, we propose to weigh each term inside the sum by $|P| \left(\sum_{P \in \mathcal{P}} |P|\right)^{-1}$. This choice penalizes fragmented predictions by eliminating their global effects on the total precision. Using these implementations of precision and recall, we can compute an $F1$-score and the area under the precision-recall curve (AUPRC). To further justify this choice, we examine the anomaly scores produced by different methods. We compare the order induced by the point-wise $F1$-score with that produced by the $F1$-score using $TRec^*$ and the adjusted $TPrec^*$ and find that the latter closer matches our intuition. For example, while LSTM-P produces scores that fluctuate within anomaly windows and spike near the end of and even outside anomaly windows, likely resulting in fragmented predictions, TCN-AE produces scores that smoothly increase and decrease over the duration of anomaly windows, resulting in continuous predictions, and spiking at the terminal failure (see Figure 4).

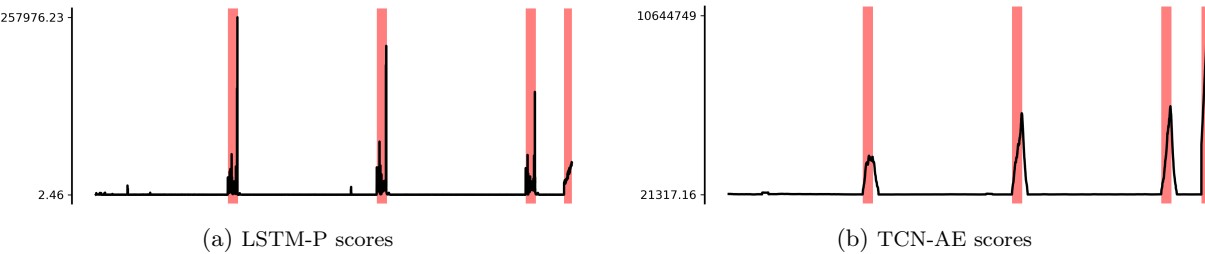

(a) LSTM-P scores        (b) TCN-AE scores

Figure 4: Scores from two different methods on a test time series of Exathlon 2, where (b) performs better according to our score, but worse according to the point-wise $F1$-score, providing an ordering aligned with our intuition.

### 4.3 Evaluation

To address the inconsistencies in evaluation protocols and provide the necessary tools for consistent evaluations, we introduce our TimeSeAD library. The library consists of a general training and evaluation framework geared towards deep learning based methods, several analysis tools for datasets and methods, as well as a large collection of architectural elements, methods, and baselines. The collections of architectural elements are implemented on top of PyTorch (Paszke et al., 2019) to provide reusable building blocks allowing for great customization. This setup allows researchers to prototype ideas quickly and users to adjust individual elements to any setting. Using these elements, we implemented 28 deep methods. Furthermore, the library provides a common interface for all datasets considered in this study alongside several analysis tools which we used in the evaluation in Section 3.1. Additionally, we provide a fast (linear in time) implementation of time series precision and recall along with the extensions proposed in the previous section. All elements of this library are specifically designed to work well with time-series data.

The general framework for training and evaluation provides the foundation for a unified evaluation and enables integration into customized experiment management systems. Thus we implemented a separate plugin based on sacred (Greff et al., 2017) to run and manage the experiments for the evaluation of methods and datasets. We used this setup to conduct our analysis of the datasets in Section 3.1 and create a benchmark of 28 methods on 21 datasets. Since performance can vary greatly between datasets for two sets of hyperparameters, we adapt grid search to tune the hyperparameters over a preselected set of parameter choices. To perform grid search without introducing significant bias in the evaluation, we remove part of the test set to tune the parameters on, before evaluating with the best performing parameters on the rest. Because of distributional changes in the test set, a fixed, arbitrary split can introduce further bias. To mitigate its effects, instead, we perform cross-validation on the test set, splitting it into multiple folds and using each fold once as a validation set. Finally, to mitigate the impact of temporal dependencies between folds, we remove the neighboring folds of each validation set. To ensure a fair evaluation, we choose a maximum training time and adjust the size of the parameter grid, such that each method can be fully evaluated within this time frame. We use this evaluation protocol for all methods[3].

### 4.4 Benchmark results

See Table 1 for the main results of our evaluation on SMD and Exathlon. Here, we show the best $F_1$ scores based on $TRec^*$ and $TPrec^*$, introduced in chapter Section 4.2. To maintain clear visibility, we only report the induced ranking, where 1 corresponds to the highest and 28 to the lowest score. Interested readers can find the *raw* scores alongside evaluations with different metrics in Appendix E.

On SMD, we can see a consistently strong performance by older (less complex) methods, such as LSTM-AE (Malhotra et al., 2016) and LSTM-P (Malhotra et al., 2015). In contrast, several modern approaches, such as the group of GAN-based methods, often perform poorly on SMD datasets. Interestingly, SMD and Exathlon do not share any methods in the top three best performing methods. In fact, methods performing well on SMD generally struggle on Exathlon and the other way around, indicating that there is currently no dominant architecture for multivariate time series AD. However, the autoencoder- and prediction-based methods perform consistently across multiple datasets. Whereas, the methods collected in *other* have the most difficulties across datasets. Our benchmark reveals that the variational autoencoder-based method GMM-GRU-VAE (Zhang et al., 2021), the prediction-based method GDN (Deng & Hooi, 2021), and the reconstruction-based STGAT-MAD (Zhan et al., 2022) perform the most consistent across SMD and Exathlon.

There are many reasons why our results might not reflect the promised advances of numerous papers. First, we use a standardized evaluation protocol with a fixed training time during hyperparameter searches to guarantee a fair comparison. As more complex models usually take longer to train, we shrink the grid of possible hyperparameters to fit our limited time budget (see Section 4.3 for details). Second, we tune the models by maximizing our novel $F_1$ score based on $TRec^*$ and $TPrec^*$. Lastly, authors do not always provide an official implementation, meaning we replicate some methods solely based on the corresponding paper.

## 5 Discussion

In this section, we discuss current and future challenges of time-series AD.

**Quality datasets** are the backbone of any evaluation. The analysis in Section 3.1 revealed multiple severe flaws in many widely used multivariate time-series datasets. Other datasets used for evaluation are sometimes not publicly available (Audibert et al., 2020; Park et al., 2018). Thus, assessing their quality is virtually impossible. This poses a huge problem for the field going forward. Publicly available high-quality datasets, such as CIFAR (Krizhevsky et al., 2009) or ImageNet (Deng et al., 2009) in computer vision, provide great platforms for comparative evaluations propelling their respective fields forward. Such a dataset is utterly needed for multivariate time-series AD. Our analysis shows that any new dataset needs to undergo careful scrutiny. The analysis tools in our TimeSeAD library provide a solid baseline, but future discussion will likely reveal more potential flaws and pitfalls. Automated detection of distributional shift especially yields great potential for future analysis.

---

[3]We provide a detailed description in Appendix D.

Table 1: Cross-validation results on Exathlon and SMD. We report the ranks according to the best $F_1$-score based on $TRec^*$ and $TPrec^*$ averaged over all test folds. $\mu_s^{Exa}$ and $\mu_s^{SMD}$ are the ranked average scores over all datasets in Exathlon and SMD, respectively. $\mu_s^{all}$ shows the ranked order of the weighted average scores over all datasets from both Exathlon and SMD. We weight $\mu_s^{all}$ by the number of datasets in Exathlon and SMD in order to treat both datasets equally. **Bold** are the top 3, normal size the top 9, and tiny all other methods for each dataset. Full results are provided in Appendix E.

| | ID | Exathlon | | | | | | | SMD | | | | | | | | | | | | | | | | $\mu_s^{all}$ |
|---|---|---|---|---|---|---|---|---|---|---|---|---|---|---|---|---|---|---|---|---|---|---|---|---|---|
| | | 1 | 2 | 4 | 5 | 6 | 9 | $\mu_s^{Exa}$ | 1 | 6 | 8 | 9 | 10 | 11 | 13 | 14 | 16 | 17 | 20 | 21 | 24 | 26 | 27 | $\mu_s^{SMD}$ | |
| reconstruction | LSTM-AE | 24 | **2** | 22 | 21 | 22 | 24 | 22 | 4 | **3** | 5 | 10 | **1** | 8 | 4 | **1** | **1** | 3 | 3 | **1** | **2** | 4 | **3** | **1** | 4 |
| | LSTM-Max-AE | 5 | 23 | 24 | 19 | 21 | 8 | 20 | **3** | 24 | 22 | 12 | 21 | 17 | 6 | 19 | 20 | 11 | 21 | 25 | 12 | 7 | 11 | 17 | 22 |
| | MSCRED | 10 | **1** | **2** | 14 | 4 | **1** | **1** | 9 | 19 | **1** | 20 | 19 | 26 | 21 | 26 | 13 | 9 | 24 | 23 | 21 | 22 | 18 | 20 | 12 |
| | FC-AE | 4 | 20 | 11 | 15 | 14 | 11 | 10 | 7 | 13 | 8 | 9 | 11 | 13 | 9 | 8 | 6 | 6 | 14 | 6 | 9 | 8 | **2** | 7 | 7 |
| | USAD | 7 | 18 | 10 | 11 | 17 | 20 | 15 | 24 | 21 | 20 | 18 | 20 | 11 | 10 | 10 | 24 | 13 | 12 | 15 | 20 | 23 | 15 | 15 | 16 |
| | TCN-AE | 8 | 5 | 15 | 9 | 19 | **2** | **3** | 21 | 15 | 6 | 23 | 22 | 25 | 23 | 23 | 16 | 16 | 5 | 20 | 22 | **1** | 23 | 21 | 17 |
| | GenAD | **3** | 25 | 4 | 10 | **1** | 28 | 4 | 20 | 28 | 24 | 25 | 24 | 28 | 8 | 20 | 23 | 18 | 28 | 28 | 18 | 14 | 13 | 24 | 23 |
| | STGAT-MAD | 14 | 17 | 9 | 20 | **2** | 16 | 14 | 13 | 7 | 14 | 4 | 4 | 10 | 14 | 4 | **3** | 7 | 4 | 10 | 8 | **2** | 5 | 5 | **3** |
| | AnomalyTransformer | 27 | 27 | 19 | 27 | **3** | 25 | 27 | 26 | 20 | 19 | 28 | 25 | 22 | 27 | 17 | 22 | 26 | 25 | 14 | 19 | 19 | 28 | 27 | 27 |
| prediction | LSTM-P | 19 | 12 | 23 | 8 | 27 | 13 | 25 | **1** | **1** | **2** | 14 | 6 | 9 | **2** | **2** | 4 | **2** | 11 | 7 | 11 | 12 | 4 | **2** | 10 |
| | LSTM-S2S-P | 13 | 19 | **1** | 24 | 13 | 6 | 11 | 6 | 16 | **3** | 19 | 23 | 23 | 24 | 25 | 10 | 15 | 17 | 18 | 15 | 13 | 21 | 18 | 18 |
| | DeepAnt | 12 | 10 | 7 | 12 | 12 | 14 | 7 | 10 | 12 | 12 | 5 | 17 | 15 | 15 | 9 | 7 | 14 | 10 | 13 | 6 | 10 | 20 | 12 | 11 |
| | TCN-S2S-P | 15 | 7 | 13 | 13 | 24 | 17 | 19 | 16 | **2** | 4 | 6 | 9 | 7 | 16 | 12 | **2** | **1** | **2** | 5 | 5 | 11 | **1** | **3** | 5 |
| | GDN | **2** | 6 | 17 | 16 | 9 | 5 | **2** | **2** | 14 | 7 | 11 | 7 | 14 | 13 | 14 | 15 | 10 | 14 | 11 | 4 | 20 | 10 | 10 | **2** |
| VAEs | LSTM-VAE | 20 | 14 | 14 | **2** | 8 | 18 | 6 | 15 | 11 | 17 | 21 | **2** | 5 | 7 | 13 | 9 | 20 | 8 | **2** | 7 | 24 | 14 | 9 | 8 |
| | Donut | 23 | 22 | 8 | 4 | 20 | 10 | 16 | 17 | 6 | 9 | **3** | **3** | 6 | 19 | 5 | 18 | 5 | **1** | 9 | **1** | 26 | 9 | 6 | 6 |
| | LSTM-DVAE | 18 | 24 | 18 | **3** | 23 | 19 | 17 | 25 | 10 | 15 | 22 | 8 | 4 | 18 | **3** | 12 | 23 | 9 | **3** | 10 | 21 | 24 | 13 | 13 |
| | GMM-GRU-VAE | 21 | 11 | 20 | 6 | 6 | 4 | 5 | 11 | 5 | 11 | **2** | 5 | **1** | 17 | 6 | 14 | 4 | 6 | 8 | **3** | 15 | 6 | 4 | **1** |
| | OmniAnomaly | 25 | 21 | 27 | **1** | 5 | 12 | 21 | 18 | 4 | 16 | 8 | 16 | **2** | **1** | 15 | 5 | 21 | 18 | 16 | 14 | 27 | 7 | 11 | 14 |
| | SIS-VAE | 17 | 16 | 6 | 7 | 7 | 22 | 12 | 5 | 9 | 10 | 7 | 12 | 12 | 11 | 7 | 8 | 8 | 7 | 12 | 13 | **3** | 8 | 8 | 9 |
| GANs | BeatGAN | 6 | **3** | 16 | 18 | 15 | 15 | 8 | 19 | 18 | 18 | 15 | 13 | 16 | 12 | 18 | 21 | 12 | 16 | 17 | 23 | 17 | 17 | 16 | 15 |
| | MAD-GAN | 9 | 15 | 12 | 23 | 10 | 23 | 18 | 22 | 23 | 13 | 17 | 18 | 24 | 22 | 16 | 17 | 24 | 13 | 26 | 27 | 18 | 26 | 23 | 24 |
| | LSTM-VAE-GAN | 11 | 8 | 5 | 25 | 16 | 7 | 13 | 14 | 17 | 21 | **1** | 15 | 19 | **3** | 22 | 25 | 22 | 21 | 19 | 25 | 6 | 12 | 19 | 20 |
| | TadGAN | **1** | 4 | 21 | 17 | 18 | 21 | 9 | 12 | 26 | 27 | 16 | 14 | 21 | 5 | 21 | 19 | 17 | 19 | 21 | 17 | 5 | 25 | 22 | 21 |
| other | LSTM-AE OC-SVM | 16 | 9 | 25 | 26 | 26 | 26 | 26 | 27 | 25 | 25 | 26 | 27 | 18 | 26 | 27 | 27 | 19 | 23 | 27 | 24 | 9 | 16 | 26 | 26 |
| | MTAD-GAT | 22 | 13 | 26 | 5 | 11 | 9 | 24 | 23 | 8 | 26 | 13 | 10 | **3** | 20 | 11 | 11 | 25 | 20 | 4 | 16 | 16 | 22 | 14 | 19 |
| | NCAD | 28 | 28 | 28 | 28 | 28 | 27 | 28 | 28 | 27 | 28 | 27 | 28 | 27 | 28 | 28 | 28 | 28 | 26 | 24 | 28 | 28 | 27 | 28 | 28 |
| | THOC | 26 | 26 | **3** | 22 | 25 | **3** | 23 | 8 | 22 | 23 | 24 | 26 | 20 | 25 | 24 | 26 | 27 | 27 | 22 | 26 | 25 | 19 | 25 | 25 |

**Generating data**—normal or anomalous—could be used to address certain shortcomings of datasets. The task of creating large real-world datasets of high quality is difficult for many reasons, often involving complex systems with many interactions. Thus, recent advances in data generation could be used to augment small datasets. Even fully artificial datasets could be used to evaluate algorithms with respect to specific aspects of the data (Schmidl et al., 2022). On the other hand, generated anomalies could be purposefully injected into datasets to expand the range of anomalous situations or simulate and recreate anomalies without having to observe these anomalies in the system. Anomalies in large systems are often expensive to induce or tied to critical system failures making the latter option appealing for data generation outside simulations.

**A common evaluation metric** is another critical tool to make evaluations comparable, eliminating the need to reevaluate many methods repeatedly. In Section 4.2 we presented a recall consistent implementation of recall for time-series data and an accompanying precision with adjusted bias. We illustrate its capabilities experimentally through examples and the benchmark presented in Section 4.3, and justify its definition theoretically in Section 4.2. However, an in-depth experimental comparison to its alternatives on a wide range of use cases could help steer the community in a unified direction. Particularly interesting could be an analysis beyond the standard setting, for example, early detection.

**The TimeSeAD library** provides, at the time of writing, a shared evaluation protocol, a collection of 28 methods, and several analysis tools. New methods are proposed constantly, and the library and future benchmarks need to adjust accordingly. Thus, if adopted by the community, we will continue to expand the library, including more methods, metrics, and datasets. Of particular interest are shallow baselines. To truly justify using large deep models, a solid collection of easier-to-train shallow methods beyond the trivial baselines currently implemented is needed to provide a well-rounded benchmark. Especially if datasets become more complex and high-dimensional, we expect shallow methods to quickly fall behind in performance, as has been the case for other settings. We encourage researchers to contribute their methods and experiments to grow the library.

**Explanation and robustness** play an essential role in safety-critical applications, such as self-driving cars. The correlation and dependencies between features in multivariate time series, in particular, offer great opportunities and challenges for explanations and robustness. Some methods offer the necessary mechanisms to enable explanations, such as feature-based attention or graph-based structures (Zhao et al., 2020; Deng & Hooi, 2021; Hua et al., 2022; Zhan et al., 2022), leaving ample room to explore these concepts further. Robustness to corrupted training samples can be important when large-scale data collection is noisy or unreliable. Recently, Li et al. (2022) analyzed robustness in time-series AD, however, they consider only four methods on four datasets relying on point-wise metrics. Thus, much more research is needed to explore robustness further.

**Other settings** beyond the one presented in this paper appear in many applications. Some require anomalies to be detected as early as possible, where our metric can be adapted by changing the bias function. Other settings require anomalies to be detected even before an anomalous event occurs. This requires more than adjusted metrics. At least the datasets need to include anomalies that are detectable ahead of time. Another interesting question is that of generalization. Our benchmark revealed that no method consistently performs best or worst across all datasets. Simply training on multiple datasets at once seems infeasible since datasets can contain different sets of features. Moreover, even if they contain the same amount of features, individual samples can differ remarkably, spreading them thin. Another emerging setting considers unequally spaced time series (Jeong et al., 2022), where we can technically apply classical methods without much effort. However, developing alternatives that explicitly address temporal irregularities could be an interesting line of research.

## 6    Conclusion

Many datasets are severely flawed and form a shaky foundation for AD evaluations. Even carefully constructed datasets (such as those in Exathlon) reveal flaws under careful scrutiny. In addition, despite their well-known problems, point-wise metrics are still the de-facto standard in most evaluations. Together with inconsistent evaluations, these three main issues create an illusion of progress in time-series AD. We have proposed TimeSeAD, a library for anomaly detection on multivariate time-series data specialized in

deep-learning based methods. TimeSeAD contains a new metric that considers temporal dependencies and produces reliable results, as we demonstrate, a collection of analysis tools for datasets and methods, implementations of 28 methods, and a general evaluation framework. The metric is provably recall-consistent and allows for customization through the bias function. Using our library, we created a substantial benchmark revealing no method that consistently outperforms any competitors. We found that modern approaches often struggle to reach the performance of older methods. We hope that our comprehensive `TimeSeAD` library aids the community in measuring the gains of new algorithms in the future and thus helps to shed some light on the actual progress in (deep) multivariate time-series AD.

**Acknowledgments**

Part of this work was conducted within the DFG research unit FOR 5359 on Deep Learning on Sparse Chemical Process Data (KL 2698/6-1 and KL 2698/7-1). FCFS acknowledges support from TU Berlin and BASF SE under the BASLEARN - TU Berlin (BASF Joint Lab for Machine Learning) project. MK acknowledges support by the Carl-Zeiss Foundation, the DFG awards KL 2698/2-1, KL 2698/5-1, KL 2698/6-1, and KL 2698/7-1, and the BMBF awards 03|B0770E and 01|S21010C.

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

## A   Metrics

Let $X \in \mathbb{R}^{T \times F}$ be a time series of length $T$ and dimension $F$ and let $y \in \mathbb{R}^T$ be the corresponding point-wise labels. Given an online anomaly detector, $s \colon \mathbb{R}^{T \times F} \to \mathbb{R}^T$ that computes a score for each time step in $X$ based only on points that came before it. The set of anomalies is

$$\mathcal{A} = \{[a,b] \subset [T] \mid \forall t \in [a,b] \colon y[t] = 1; \nexists [a',b'] \supsetneq [a,b] \colon \forall t \in [a',b'] \colon y[t] = 1\}$$

and the set of all predictions for a threshold $\lambda \in \mathbb{R}$ is

$$\mathcal{P}^\lambda = \{[a,b] \subset [T] \mid \forall t \in [a,b] \colon s(x)[t] \geq \lambda; \nexists [a',b'] \supsetneq [a,b] \colon \forall t \in [a',b'] \colon s(x)[t] \geq \lambda\}.$$

Given a cardinality function $\gamma \colon \mathbb{N} \times \mathcal{P}([T]) \to \mathbb{R}_{\geq 0}$ and a bias function $\delta \colon \mathbb{R} \to \mathbb{R}_{\geq 0}$, where $\mathcal{P}([T])$ is the power set of $[T]$, the time-series recall is given by

$$TRec(\mathcal{A}, \mathcal{P}) = \frac{1}{|\mathcal{A}|} \sum_{A \in \mathcal{A}} \left[ \alpha \mathbb{1}(|\mathcal{P}_A| > 0) + (1 - \alpha)\gamma(|\mathcal{P}_A|, A) \sum_{P \in \mathcal{P}} \frac{\sum\limits_{t \in P \cap A} \delta(t - \min A, |A|)}{\sum\limits_{t \in A} \delta(t - \min A, |A|)} \right]$$

with $\mathcal{P}_A = \{P \in P \mid |A \cap P| > 0\}$. The cardinality function is monotone decreasing in its first argument and $\gamma(1, \cdot) = 1$.

**Proof of Theorem 1:** It is straightforward to see, that $\gamma$ is monotone decreasing, as the maximum is over all values with smaller inputs multiplied by a factor smaller than one. It remains to show, that the resulting $TRec$ is recall consistent.

Since the terms within the sum are all non-negative, it suffices to show, that each individual term only ever decreases. Consider two thresholds $\lambda, \lambda' \in \mathbb{R}$ with $\lambda' > \lambda$ and anomaly $A \in \mathcal{A}$ such that $\mathcal{P}_A^\lambda \neq \mathcal{P}_A^{\lambda'}$. Note that $|\mathcal{P}_A^\lambda| = 0$ implies $|\mathcal{P}_A^{\lambda'}| = 0$. If $|\mathcal{P}_A^\lambda| = 0$, the inner sum is zero, and the statement is true. Thus, we assume $|\mathcal{P}_A^{\lambda'}| > 0$ and can therefore ignore the first term inside the outer sum, since $\mathbb{1}(|\mathcal{P}_A^\lambda| > 0) = \mathbb{1}(|\mathcal{P}_A^{\lambda'}| > 0)$ always holds.

First, we consider the case $|\mathcal{P}_A^{\lambda'}| \geq |\mathcal{P}_A^\lambda|$. Since $\gamma$ is monotone decreasing in its first argument and the inner sum loses at least one non-negative term, the second term can either decrease or stay the same.

Next, we consider the case $|\mathcal{P}_A^{\lambda'}| < |\mathcal{P}_A^\lambda|$. We want to show that each term only ever decreases with an increasing threshold, i.e.

$$\gamma(|\mathcal{P}_A^\lambda|, A) \sum_{P \in \mathcal{P}_A^\lambda} \frac{\sum\limits_{t \in P \cap A} \delta(t - \min A, |A|)}{\sum\limits_{t \in A} \delta(t - \min A, |A|)} \geq \gamma(|\mathcal{P}_A^{\lambda'}|, A) \sum_{P \in \mathcal{P}_A^{\lambda'}} \frac{\sum\limits_{t \in P \cap A} \delta(t - \min A, |A|)}{\sum\limits_{t \in A} \delta(t - \min A, |A|)}.$$

If $\gamma(|\mathcal{P}_A^{\lambda'}|, A) = 0$, the recall does not not change, because $\gamma$ is monotone decreasing. Thus we assume $\gamma(|\mathcal{P}_A^{\lambda'}|, A) > 0$, in which case the inequality above holds if and only if

$$\frac{\gamma(|\mathcal{P}_A^\lambda|, A)}{\gamma(|\mathcal{P}_A^{\lambda'}|, A)} \geq \frac{\sum\limits_{P \in \mathcal{P}_A^{\lambda'}} \sum\limits_{t \in P \cap A} \delta(t - \min A, |A|)}{\sum\limits_{P \in \mathcal{P}_A^\lambda} \sum\limits_{t \in P \cap A} \delta(t - \min A, |A|)}.$$

Consider $\Delta\delta = \sum\limits_{P \in \mathcal{P}_A^\lambda} \sum\limits_{t \in P \cap A} \delta(t - \min A, |A|) - \sum\limits_{P \in \mathcal{P}_A^{\lambda'}} \sum\limits_{t \in P \cap A} \delta(t - \min A, |A|) > 0$. Then it holds

$$\Delta\delta \geq |\bigcup_{\substack{P \in \mathcal{P}_A^\lambda \\ P' \in \mathcal{P}_A^{\lambda'}}} P \setminus P'| \geq |\mathcal{P}_A^\lambda| - |\mathcal{P}_A^{\lambda'}|$$

Since $P \cap A \subset A$, we also know

$$
\frac{\sum\limits_{P \in \mathcal{P}_A^{\lambda'}} \sum\limits_{t \in P \cap A} \delta(t - \min A, |A|)}{\sum\limits_{P \in \mathcal{P}_A^\lambda} \sum\limits_{t \in P \cap A} \delta(t - \min A, |A|)} = \frac{\sum\limits_{P \in \mathcal{P}_A^\lambda} \sum\limits_{t \in P \cap A} \delta(t - \min A, |A|) - \Delta\delta}{\sum\limits_{P \in \mathcal{P}_A^\lambda} \sum\limits_{t \in P \cap A} \delta(t - \min A, |A|)}
$$

$$
\leq \frac{\sum\limits_{t \in A} \delta(t - \min A, |A|) - \Delta\delta}{\sum\limits_{t \in A} \delta(t - \min A, |A|)}
$$

$$
\leq \frac{\sum\limits_{t \in A} \delta(t - \min A, |A|) - (|\mathcal{P}_A^\lambda| - |\mathcal{P}_A^{\lambda'}|)}{\sum\limits_{t \in A} \delta(t - \min A, |A|)}
$$

Thus, if

$$
\gamma(|\mathcal{P}_A^\lambda|, A) \geq \frac{\sum\limits_{t \in A} \delta(t - \min A, |A|) - (|\mathcal{P}_A^\lambda| - |\mathcal{P}_A^{\lambda'}|)}{\sum\limits_{t \in A} \delta(t - \min A, |A|)} \gamma(|\mathcal{P}_A^{\lambda'}|, A)
$$

holds true for all $0 < |\mathcal{P}_A^{\lambda'}| < |\mathcal{P}_A^\lambda|$, the resulting recall is recall consistent. $\qquad\square$

**Lemma 1** For any $x \in \mathbb{R}_{\geq 1}$ it holds $\left(\frac{x-1}{x}\right)^n \geq \frac{x-n}{x}$ for all $n \in \mathbb{N}$.
**Proof:** We know $\left(\frac{x-1}{x}\right)^1 = \frac{x-1}{x}$. By induction over $n$, it holds

$$
\left(\frac{x-1}{x}\right)^{n+1} \geq \frac{x-1}{x}\frac{x-n}{x} = \frac{x-(n+1)}{x} + \frac{n}{x^2} \geq \frac{x-(n+1)}{x}
$$

$\qquad\square$

**Proof of Theorem 2:** We show the proposition by induction. First, note that $\gamma^*(n, A) = \left(\frac{|A|-1}{|A|}\right)^0 = 1$.
Now assume $\gamma^*(m, A) = \left(\frac{|A|-1}{|A|}\right)^{m-1}$ for all $m \leq n$. Then it holds

$$
\gamma^*(n+1, A) = \max_{0 < m < n+1} \frac{\sum\limits_{t \in A} \delta(t - \min A, |A|) - n - 1 + m}{\sum\limits_{t \in A} \delta(t - \min A, |A|)} \gamma(m, A)
$$

$$
= \max_{0 < m < n+1} \frac{|A| - n - 1 + m}{|A|} \left(\frac{|A|-1}{|A|}\right)^{m-1}
$$

$$
\geq \frac{|A|-1}{|A|} \left(\frac{|A|-1}{|A|}\right)^{n-1} = \left(\frac{|A|-1}{|A|}\right)^n
$$

Furthermore, it holds

$$
\gamma^*(n+1, A) = \max_{0 < m < n+1} \frac{\sum\limits_{t \in A} \delta(t - \min A, |A|) - n - 1 + m}{\sum\limits_{t \in A} \delta(t - \min A, |A|)} \gamma(m, A)
$$

$$
= \max_{0 < m < n+1} \frac{|A| - n - 1 + m}{|A|} \left(\frac{|A|-1}{|A|}\right)^{m-1}
$$

$$
\overset{\text{Lemma 1}}{\leq} \max_{0 < m < n+1} \left(\frac{|A|-1}{|A|}\right)^{n+1-m} \left(\frac{|A|-1}{|A|}\right)^{m-1}
$$

$$
= \left(\frac{|A|-1}{|A|}\right)^n
$$

$\qquad\square$

# B  Datasets

In this section, we provide a more detailed description and analysis of all considered datasets. Additionally, we provide further examples of any issues we found. First and foremost, we provide the general statistics of each dataset, see Table 2.

Table 2: Statistics of each dataset

| Dataset | Features | train size | test size | Anomalies | |
|---|---|---|---|---|---|
| SWaT | 51 | 495000 | 449919 | 35 | 12.1% |
| WADI | 123 | 784571 | 172801 | 14 | 5.8% |
| SMAP | 25 | 140825 | 444035 | 69 | 12.8% |
| MSL | 55 | 58317 | 73729 | 36 | 10.5% |
| SMD 0 | 38 | 28479 | 28479 | 8 | 9.5 % |
| SMD 1 | 38 | 23694 | 23694 | 10 | 2.3 % |
| SMD 2 | 38 | 23702 | 23703 | 12 | 3.4 % |
| SMD 3 | 38 | 23706 | 23707 | 12 | 3.0 % |
| SMD 4 | 38 | 23705 | 23706 | 7 | 0.4 % |
| SMD 5 | 38 | 23688 | 23689 | 30 | 15.7 % |
| SMD 6 | 38 | 23697 | 23697 | 13 | 10.1 % |
| SMD 7 | 38 | 23698 | 23699 | 20 | 3.2 % |
| SMD 8 | 38 | 23693 | 23694 | 13 | 4.9 % |
| SMD 9 | 38 | 23699 | 23700 | 11 | 12.0 % |
| SMD 10 | 38 | 23688 | 23689 | 10 | 1.1 % |
| SMD 11 | 38 | 23689 | 23689 | 20 | 7.2 % |
| SMD 12 | 38 | 23688 | 23689 | 21 | 4.1 % |
| SMD 13 | 38 | 28743 | 28743 | 8 | 1.5 % |
| SMD 14 | 38 | 23696 | 23696 | 20 | 1.8 % |
| SMD 15 | 38 | 23702 | 23703 | 1 | 0.7 % |
| SMD 16 | 38 | 28722 | 28722 | 10 | 6.1 % |
| SMD 17 | 38 | 28700 | 28700 | 4 | 1.1 % |
| SMD 18 | 38 | 23692 | 23693 | 13 | 4.4 % |
| SMD 19 | 38 | 28695 | 28696 | 3 | 0.7 % |
| SMD 20 | 38 | 23702 | 23703 | 10 | 4.7 % |
| SMD 21 | 38 | 23703 | 23703 | 26 | 2.7 % |
| SMD 22 | 38 | 23687 | 23687 | 8 | 4.1 % |
| SMD 23 | 38 | 23690 | 23691 | 11 | 1.8 % |
| SMD 24 | 38 | 28726 | 28726 | 11 | 4.2 % |
| SMD 25 | 38 | 28705 | 28705 | 5 | 1.5 % |
| SMD 26 | 38 | 28703 | 28704 | 6 | 4.8 % |
| SMD 27 | 38 | 28713 | 28713 | 4 | 1.1 % |
| Exathlon 1 | 19 | 41382 | 49810 | 9 | 17.1 % |
| Exathlon 2 | 19 | 68917 | 96535 | 9 | 17.6 % |
| Exathlon 3 | 19 | 115160 | 15270 | 7 | 16.0 % |
| Exathlon 4 | 19 | 208720 | 133223 | 11 | 12.6 % |
| Exathlon 5 | 19 | 133411 | 190372 | 21 | 9.5 % |
| Exathlon 6 | 19 | 303087 | 97221 | 11 | 9.6 % |
| Exathlon 9 | 19 | 273247 | 103511 | 14 | 13.0 % |
| Exathlon 10 | 19 | 178685 | 106251 | 13 | 13.9 % |

Papers that evaluate on SWaT include (Li et al., 2018a; 2019; Audibert et al., 2020; Shen et al., 2020; Faber et al., 2021; Zhang et al., 2021; Xiao et al., 2021; Deng & Hooi, 2021; Carmona et al., 2021; Xu et al., 2022; Li et al., 2021b; Fährmann et al., 2022; Doshi et al., 2022; Zhan et al., 2022; Zhang et al., 2022b;a).

Papers that evaluate on WADI include (Li et al., 2019; Audibert et al., 2020; Faber et al., 2021; Deng & Hooi, 2021; Xu et al., 2022; Li et al., 2021b; Fährmann et al., 2022; Zhan et al., 2022; Zhang et al., 2022b;a).

Papers that evaluate on SMAP include (Hundman et al., 2018; Audibert et al., 2020; Geiger et al., 2020; Zhao et al., 2020; Shen et al., 2020; Zhang et al., 2021; Xiao et al., 2021; Carmona et al., 2021; Xu et al., 2022; Challu et al., 2022; Chen et al., 2022; Doshi et al., 2022; Hua et al., 2022; Chambaret et al., 2022; Zhang et al., 2022a)

Papers that evaluate on MSL include (Hundman et al., 2018; Su et al., 2019; Audibert et al., 2020; Geiger et al., 2020; Zhao et al., 2020; Shen et al., 2020; Zhang et al., 2021; Xiao et al., 2021; Wang et al., 2021; Xu et al., 2022; Challu et al., 2022; Chen et al., 2022; Doshi et al., 2022; Hua et al., 2022; Chambaret et al., 2022; Zhang et al., 2022a)

Papers that evaluate on SMD include (Su et al., 2019; Audibert et al., 2020; Xiao et al., 2021; Wang et al., 2021; Carmona et al., 2021; Xu et al., 2022; Li et al., 2021b; Challu et al., 2022; Chen et al., 2022; Doshi et al., 2022; Hua et al., 2022; Zhan et al., 2022; Zhang et al., 2022b)

Papers that evaluate on Exathlon include (Schmidl et al., 2022).

## B.1 Secure Water Treatement (SWaT)

The Secure Water Treatment (SWaT) dataset Goh et al. (2016) originates from the operation of a miniature water-treatment plant. 51 sensors were recorded during 11 days of plant operation at a sampling rate of 1 Hz. The dataset is split into a training set and a test set. The training set corresponds to the first six days of operation, during which no incidents occurred. The remaining five days make up the test set. During this time, 36 attacks on the miniature plant were conducted, both against the plant's physical components and its control software. A time step is labeled anomalous if an attack occurred at that time. In total, Goh et al. (2016) conducted 36 attacks against the system, of which two overlap, so they make up a single anomaly window. The average attack length is around 600 time steps (10 minutes). Goh et al. (2016) note that the data recording started when the plant was offline, and the first 5 hours correspond to the plant's start-up procedure. They have already removed the first 30 minutes from the dataset, but we follow Li et al. (2019) and remove the 4.5 hours after that as well. Otherwise, those data points could hamper some methods attempting to model the distribution or process that generates normal data.

With roughly 12% of points anomalous, SWaT's anomaly density is barely acceptable. The distribution of anomaly positions in SWaT, see Figure 5a, reveals no clear bias, except for one large cluster in the middle of the time series. Looking at the lengths of anomaly windows, see Figure 5b reveals an extremely long anomaly

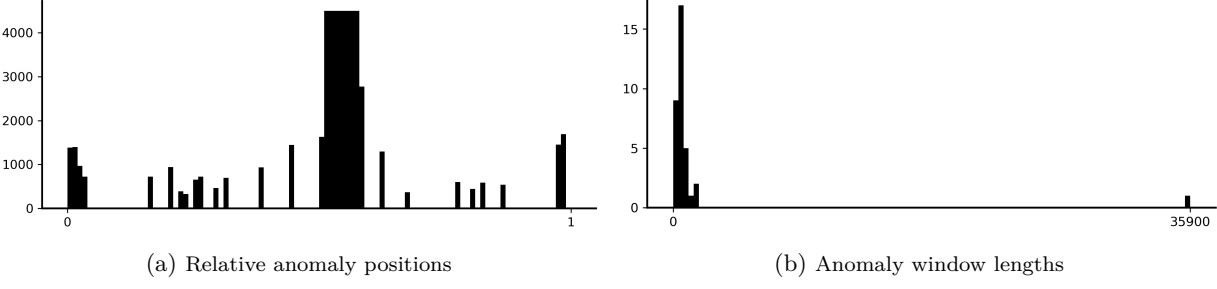

(a) Relative anomaly positions     (b) Anomaly window lengths

Figure 5: Relative position of anomalies (a) and the distribution of anomaly lengths (b) in the test set of SWaT.

window containing more than 35,000 points ( 8.5 hours). These lengths by far exceed any reasonable setting for window sizes, which usually are smaller than 100.

Looking at the mean and standard deviation of the features in SWaT, see Figure 6, reveals clear instances of distributional shift, some features that are constant throughout training and test set, and several features that seem trivial to solve. The consistently constant features mostly correspond to backup actuators that only become active when their primary counterpart fails for some reason. As this does not occur during the

training period, the backup actuators never activate. We test a trivial thresholding method on each feature,

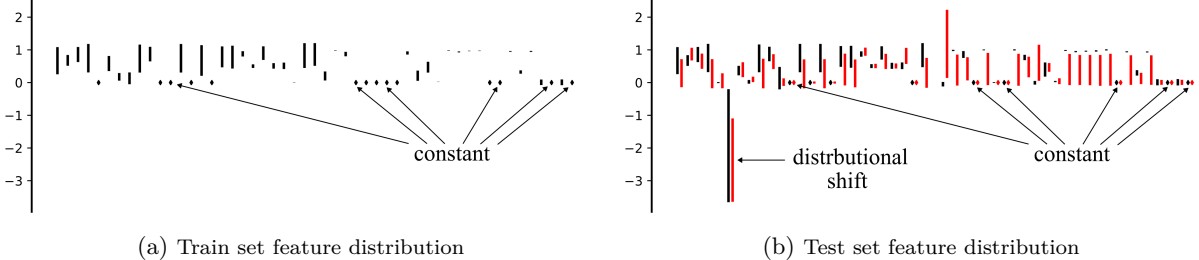

(a) Train set feature distribution

(b) Test set feature distribution

Figure 6: Mean and standard deviation for each feature in SWaT of normal points (black) and anomalies (red). Multiple features are constant across the entire dataset (diamond), some even across training and test set. Other features suggest a distributional shift between the training and test set.

by using the distance to the mean computed from the training set as the anomaly score. Indeed, for several features, we can achieve comparable performance to several deep methods. However, many modern methods still outperform the trivial baseline. Inspecting the features reveals one large anomaly responsible for the deviating mean. Multiple smaller anomalies, however, are not trivially reflected in the feature alone, see Figure 7. The specification reveals, that these sensors are flow meters. Even though a thresholding method

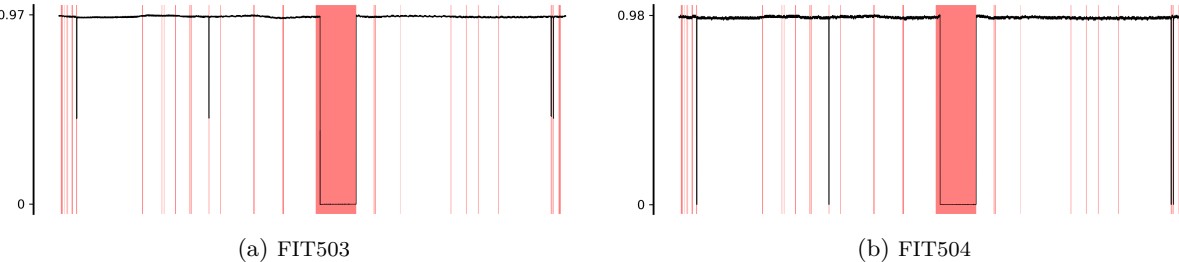

(a) FIT503

(b) FIT504

Figure 7: Two features from the test set of SWaT.

on these features presents a strong baseline, especially with respect to point-wise metrics, these features do not seem trivial.

Looking at other features in SWaT reveals some instances, where anomalies seem to cause late- or long-term effects, see Figure 8. In on instance (Figure 8b), the behavior of a feature drastically changes after some

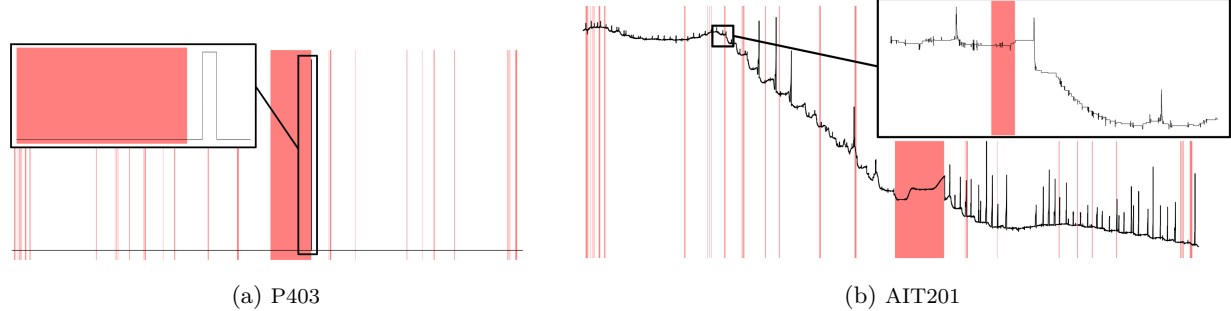

(a) P403

(b) AIT201

Figure 8: Two features from the test set of SWaT. Each feature was normalized based on the statistics of the training set.

anomalies have occurred, causing a severe distributional shift in that feature. Another example shows a sudden abnormal spike shortly after an anomaly window. Since such sikes or distributional shift does not appear in the training set, it does not seem reasonable for a fully trained anomaly detection algorithm to ignore such instances.

## B.2 WAter DIstribution (WADI)

The Water Distribution (WADI) (Ahmed et al., 2017) dataset is similar to SWaT. Its 123 features correspond to sensor values/actuator states in a miniature water-distribution grid connected to the SWaT water-treatment plant. Ahmed et al. (2017) recorded the operation of the grid for 16 continuous days at a sample rate of 1 Hz and launched a total of 15 attacks in the last two days. Thus, the test set is a single time series corresponding to the last two days of operation, whereas the training set consists of the first 14 days. We use version A2 of the dataset, where the authors removed a good chunk of the original data (425,030 of 1,209,601 data points) from the middle of the training set, which is now split into two TS. Note that the dataset file actually contains 127 columns. However, 4 of those do not contain any value at any time step, so we remove them entirely. There are also some spurious missing values in the remaining data, which we simply replace with the last available value for the affected feature.

The anomaly density in WADI seems reasonable with 6%, and the anomalies seem reasonably distributed, even if they are mostly clustered at the beginning, end, and middle of the time series, see Figure 9a. There is

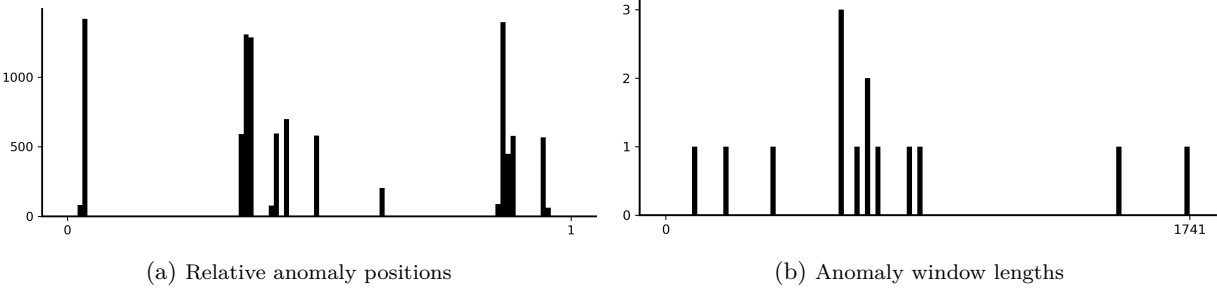

(a) Relative anomaly positions          (b) Anomaly window lengths

Figure 9: Relative position of anomalies (a) and the distribution of anomaly lengths (b) in the test set of WADI.

no extremely long anomaly window, such as in SWaT. However, several windows contain over 1,000 points, which we should still consider too long in general. Looking at the feature distribution, see Figure 10, we can see one feature in particular, for which the distributions vastly differ. Examining the piping diagram

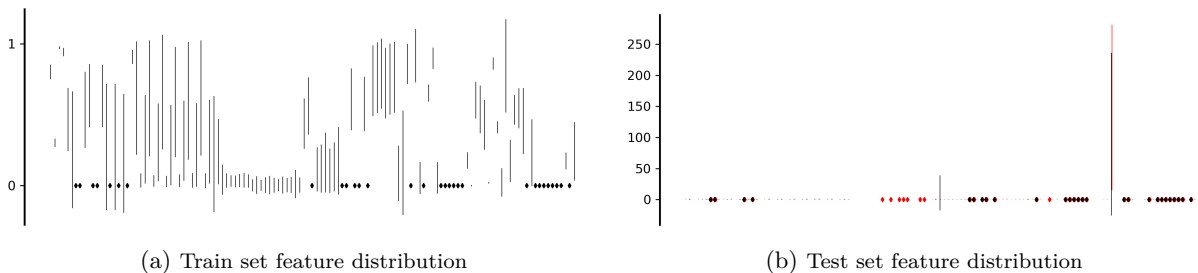

(a) Train set feature distribution          (b) Test set feature distribution

Figure 10: Mean and standard deviation for each feature in SWaT of normal points (black) and anomalies (red). Multiple features are constant across the entire dataset (diamond).

of WADI reveals that the exploding feature belongs to a turbidity sensor, and the authors claim that the previous attack introduces contaminated water to the grid. Therefore, it is not unlikely that the attack was the cause of the explosion of that feature. We are no experts on the subject, but the sensor data jumps to about 200 after normalization, see Figure 11b. If this is intended behavior, there is no way to infer this based on the training set. In other sensors we can observe possible late effects as well, see Figure 11a.

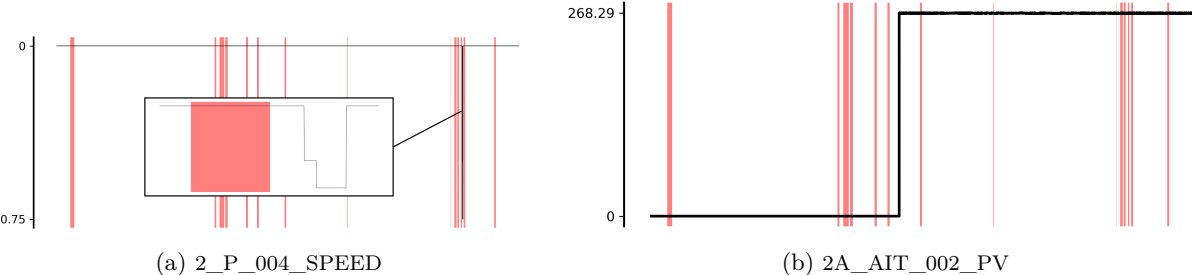

(a) 2_P_004_SPEED                                      (b) 2A_AIT_002_PV

Figure 11: Two features from the test set of WADI. Each feature was normalized based on the statistics of the training set.

### B.3 Soil Moisture Active Passive (SMAP)

The Soil Moisture Active Passive dataset (Hundman et al., 2018) contains 55 time series. All but one feature in each time series corresponds to commands sent to a satellite at a given point in time and are represented by binary features. The remaining feature contains the actual sensor values reported by the satellite. Each time series corresponds to a possibly different telemetry channel of that satellite. Thus, at least one feature is different for all time series. Furthermore, we were not able to find a specification of the remaining features. Thus, we have no way of verifying their consistency across different time series. All things considered, the time series in SMAP are technically generated by different processes and should be treated as such. To illustrate the extent of the differences between individual time series, we visualize the sensor feature from multiple time series in the training set, see Figure 12.

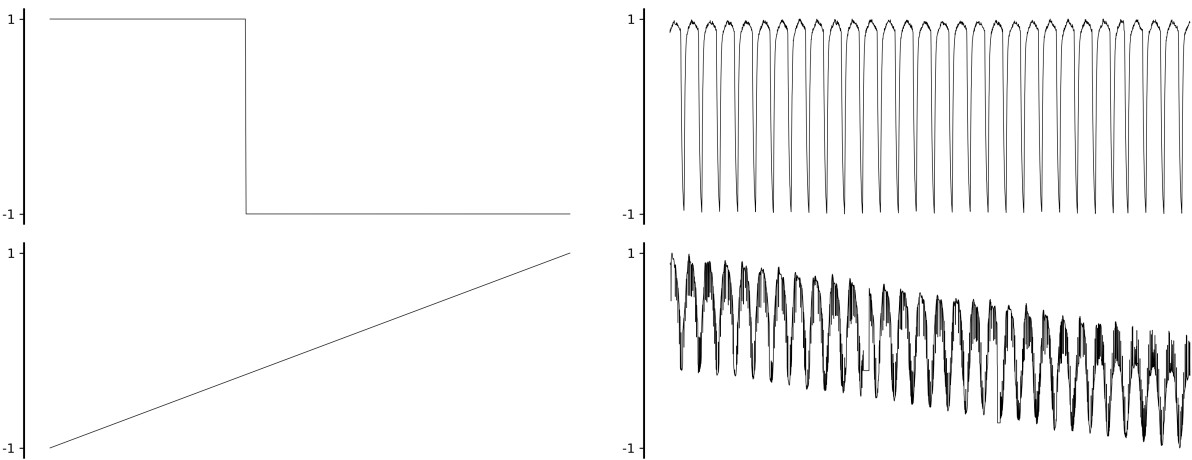

Figure 12: The sensor feature from different time series in the training set of SMAP.

Beyond the conceptual flaws, we still find a clear positional bias towards the latter half of the time series, see Figure 13a, and several anomalies longer than 2000 time points, see Figure 13b.

Since the time series are generated by different processes, we examine the distributional changes within each time series. We find instances, where all command features are constant throughout the test time series or at least after some initial period, see Figure 14. Since all methods rely on windowing, the sensor feature provides the only context for prediction, see Figure 15. Since the feature is constant before and after the anomaly and no additional information is provided by the command features, this does not seem to be a reasonable task for anomaly detection. The example shows an instance, where anomalies seem to cause long-term effects, which are not reflected in the label. We can find more examples of this behavior throughout the dataset, see Figure 16.

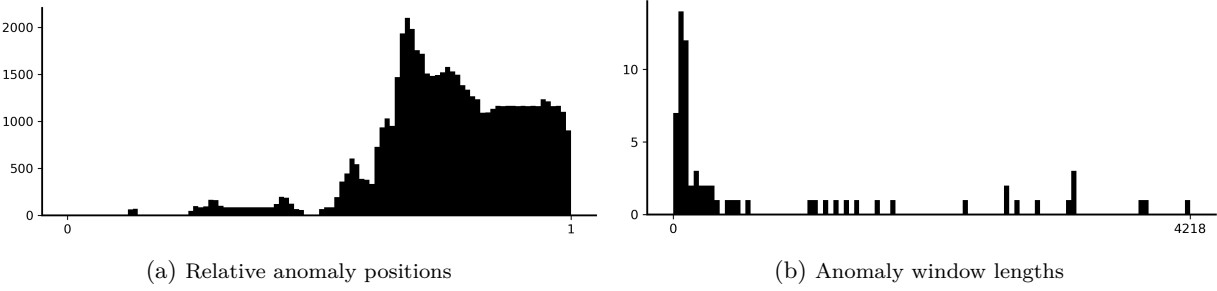

(a) Relative anomaly positions

(b) Anomaly window lengths

Figure 13: Relative position of anomalies (a) and the distribution of anomaly lengths (b) in the test set of SMAP.

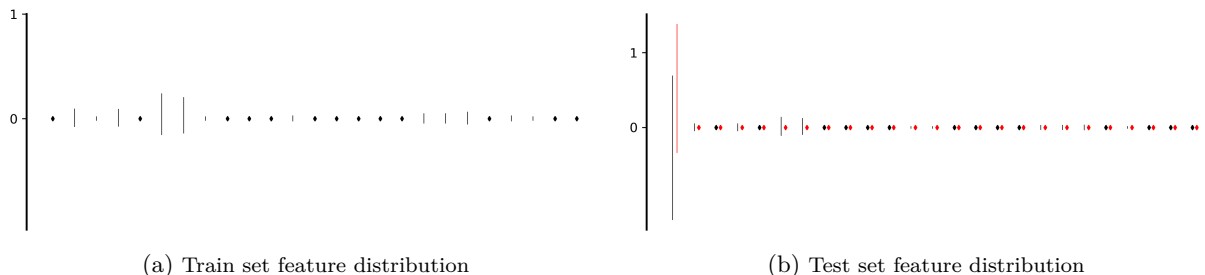

(a) Train set feature distribution

(b) Test set feature distribution

Figure 14: Example of the feature means and standard deviations

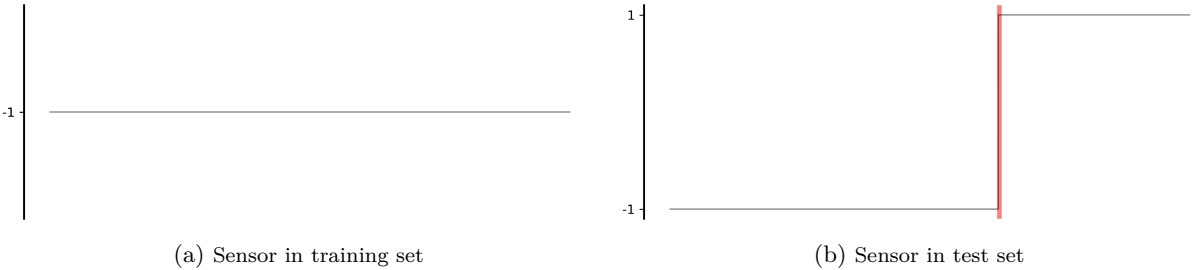

(a) Sensor in training set

(b) Sensor in test set

Figure 15: Example of the sensor feature in training and test set.

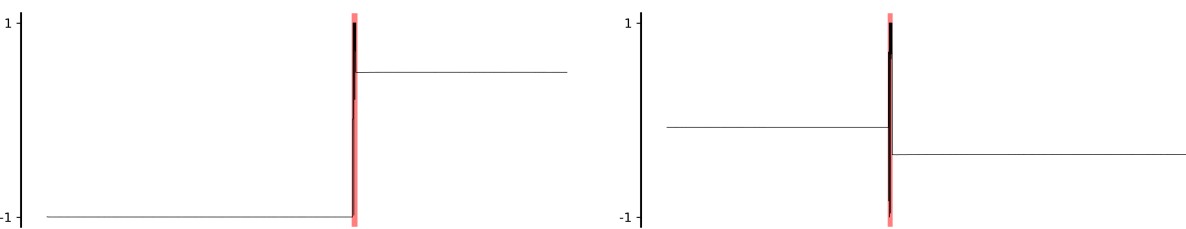

Figure 16: Example of a distributional shift in the sensor feature in the test set of SMAP.

### B.4   Mars Science Lab (MSL)

The Mars Science Lab dataset (Hundman et al., 2018) is similarly constructed as SMAP. It contains 27 time series, each containing a single telemetry value feature and binary encoded command for the rest. Thus, it shares many of the same problems as SMAP, see Figure 17.

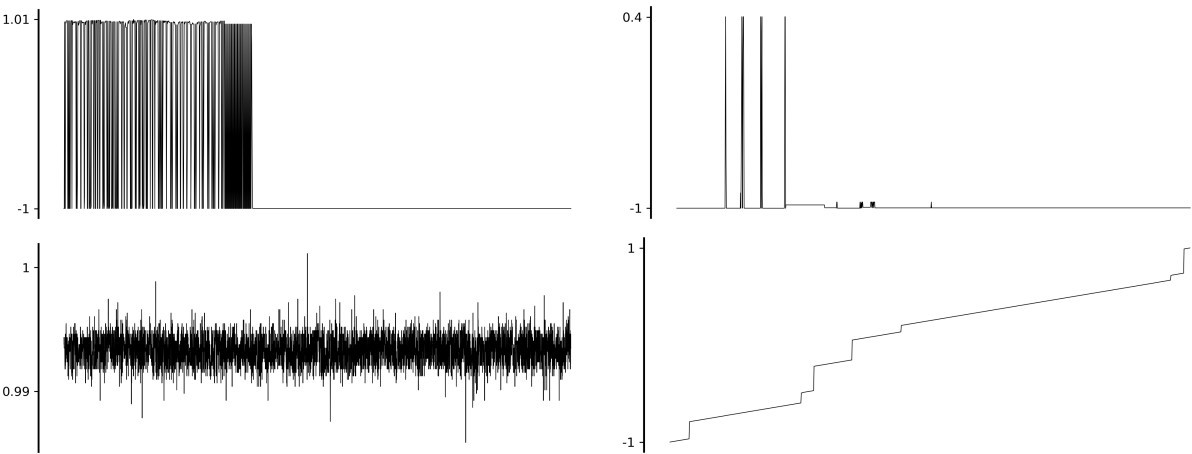

Figure 17: The sensor feature from different time series in the training set of MSL.

The positional bias, see Figure 18a, and long anomalies, see Figure 18b, are not as pronounced as in SMAP, but still noticeable.

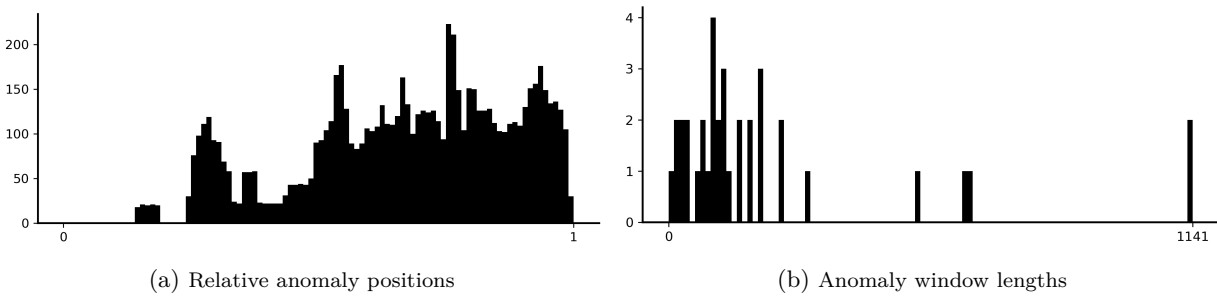

(a) Relative anomaly positions                                (b) Anomaly window lengths

Figure 18: Relative position of anomalies (a) and the distribution of anomaly lengths (b) in the test set of SMAP.

Similarly to SMAP, we can identify instances of possible distributional shift and long-term effects, see Figure 19.

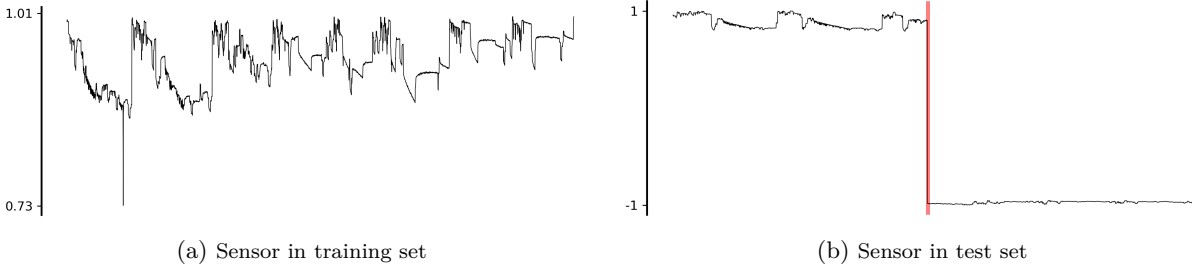

(a) Sensor in training set                                    (b) Sensor in test set

Figure 19: Example of the sensor feature in training and test set of MSL. In a wide window around the anomaly window all other features are constant.

Taking all these issues together, both SMAP and MSL do not seem suited for the evaluation of general deep time-series AD. Specialized methods that exploit the intricacies of the data will most likely outperform any general algorithms. Because the datasets are lacking detailed documentation it is difficult to assess the extent to which the long-term effects seemingly caused by anomalies are intended behavior. Even then, the datasets might be more suited to change point detection methods. Since the command features are constant for a large portion of both datasets, it begs the question of how much they can contribute to general algorithms and if the problem is as complex as their presence suggests.

## B.5 Server Machine Data (SMD)

The Server Machine Data dataset (Su et al., 2019) consists of 28 time series. According to the authors, the dataset was collected from a large internet company over a period of five weeks. The first half of the dataset comprises the training set and the latter half the test set. Unfortunately, we were not able to find any more information on this dataset. However, because each time series was apparently sampled under different conditions, each time series in this dataset should be considered independently.

Looking at the distribution of anomaly positions in each time series, we can identify three instances with a clear positional bias towards the end of the time series, see Figure 20a top. For one server in particular, see Figure 20 middle, the distribution is dominated by one large anomaly. For most time series, however,

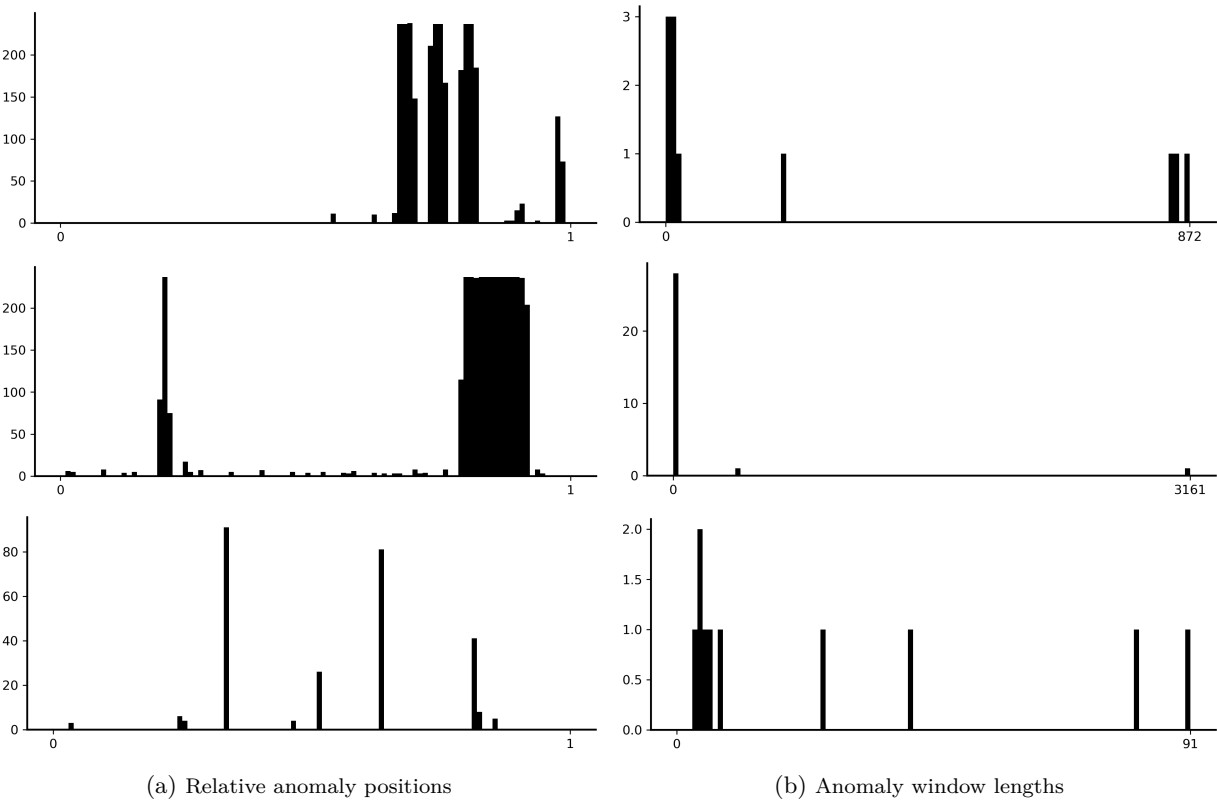

(a) Relative anomaly positions       (b) Anomaly window lengths

Figure 20: Relative position of anomalies (a) and the distribution of anomaly lengths (b) in the test set of different time series in SMD.

anomaly windows never exceed 1,000 time steps and usually not even 500 time steps. Most time series contain only few anomalies, making any definitive statement on their distribution difficult. One time series even contains just one anomaly. Time series, for which we can identify positional bias include: machine-1-1, machine-2-1, machine-2-2, machine-2-9, and machine-3-8.

Most time series suffer from consistently constant features, see Figure 21. Several of the constant features

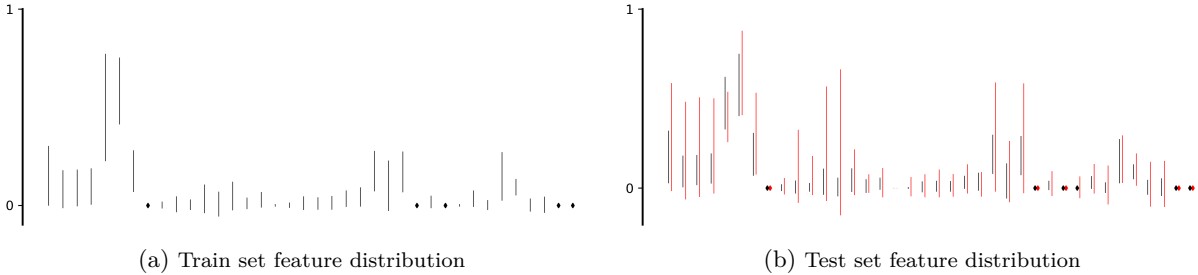

(a) Train set feature distribution      (b) Test set feature distribution

Figure 21: Example of the feature distribution of a time series in SMD.

are constant for all time series. Since we cannot say for certain what those features represent due to the lack of documentation, we are reluctant to outright remove those features. We do not expect the performance to suffer much from their inclusion as they make up only very small percentage of features.

In several time series we have found possible delayed effects of anomalies, see Figure 22. In other time series

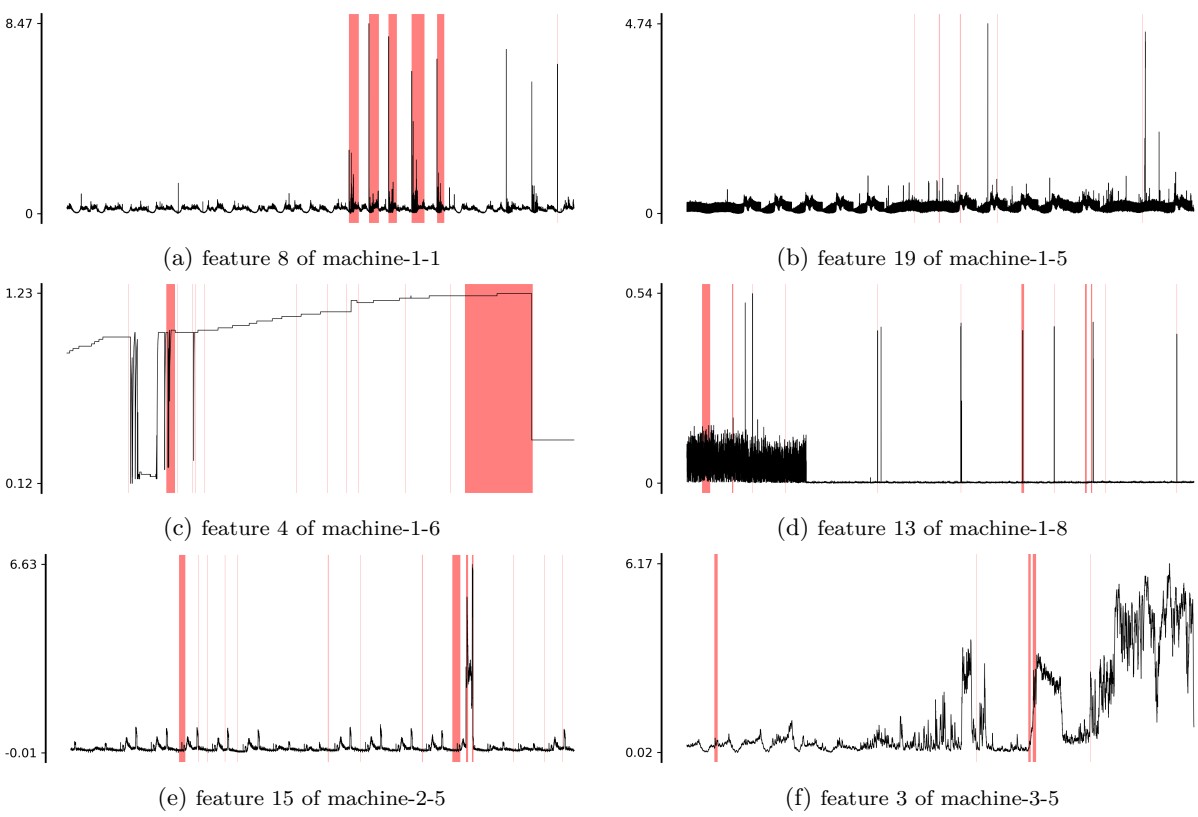

(a) feature 8 of machine-1-1      (b) feature 19 of machine-1-5

(c) feature 4 of machine-1-6      (d) feature 13 of machine-1-8

(e) feature 15 of machine-2-5      (f) feature 3 of machine-3-5

Figure 22: Instances where we suspect anomalies had delayed effects.

we strongly suspect anomalies had long term effects on the system, see Figure 23. In most instances, we can observe the effects across multiple features. That and the unusual ranges seem to affirm our assessment.

For one time series, in particular, we could observe a feature dropping to zero and staying constant directly after an anomaly occurs, see Figure 24. However, we can also observe a constant period at the start of the training set. This effect might be caused by a startup period in the training set and a crash in the test set. Without full knowledge of the underlying process, we can not give a definitive judgment on this case. However, this illustrates, that this dataset needs to undergo careful scrutiny by experts familiar with the underlying process. Until then we exclude machine-1-1, machine-1-3, machine-1-4, machine-1-5, machine-1-6,

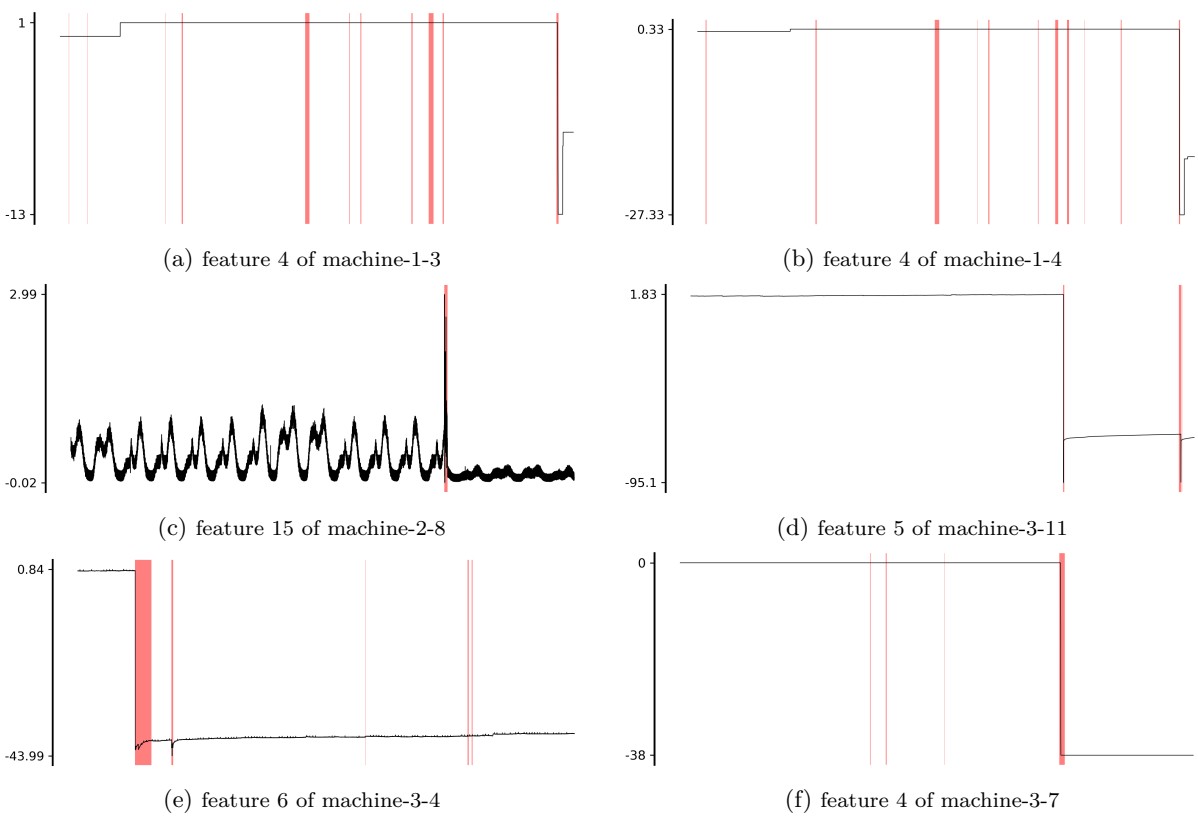

(a) feature 4 of machine-1-3

(b) feature 4 of machine-1-4

(c) feature 15 of machine-2-8

(d) feature 5 of machine-3-11

(e) feature 6 of machine-3-4

(f) feature 4 of machine-3-7

Figure 23: Instances where we suspect anomalies had long-term effects.

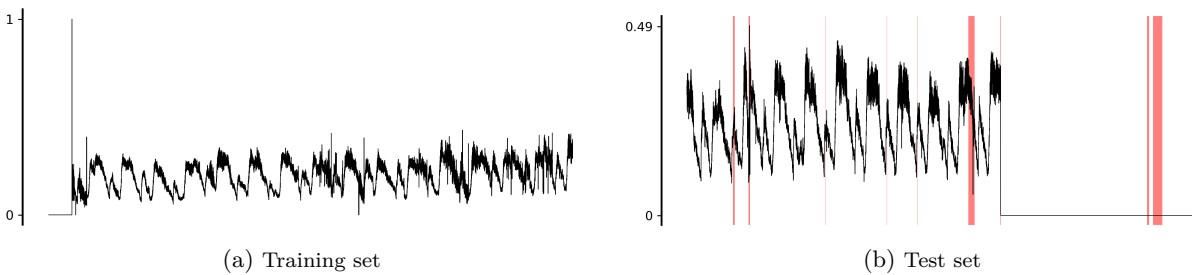

(a) Training set

(b) Test set

Figure 24: Feature 34 of machine-3-10.

machine-1-8, machine-2-5, machine-2-8, machine-3-4, machine-3-5, machine-3-7, machine-3-10, and machine-3-11 from our final report. However, we still evaluate and report on all datasets from SMD. We will provide the missing results in Appendix E.

### B.6 Exathlon

The Exathlon dataset (Jacob et al., 2020) was created from ten applications running on a Spark cluster with four nodes. The authors collected 2,283 metrics from the monitoring system and the underlying operating system. They remark, that the collected metrics could very well be correlated and suggest a curated subset of 19 features to use instead. Furthermore, in their implementation, they remove all time series from two applications (ids 7 and 8). One application contains no anomalies in the test set and the other has no training set. Thus, we only consider applications 1-6 and 9-10. The final dataset thus consists of eight datasets, each consisting of the execution traces of a single application. For the test set, they insert six types of anomalies in the cluster. One anomaly, in particular, uses up memory until the application crashes due to memory constraints, which means at least seven timer series suffer from positional bias, which is generally weakened by the other time series of each application. A detailed description of all applications and anomalies can be found on the GitHub page of the original implementation[4].

Overall we find a slight positional bias in several datasets, mostly attributed to the one anomaly discussed prior. We found no consistently constant features throughout the entire dataset. Most time series in the dataset contain unusual spikes, which the authors attribute to background activities on the cluster. Since such background activity is considered normal we ignore such cases in general. However, we would like to draw attention to one particular instance, where the spike reaches a new high directly after an anomaly occurs, see Figure 25. Since the effect of this spike is reflected in multiple features and we found no other

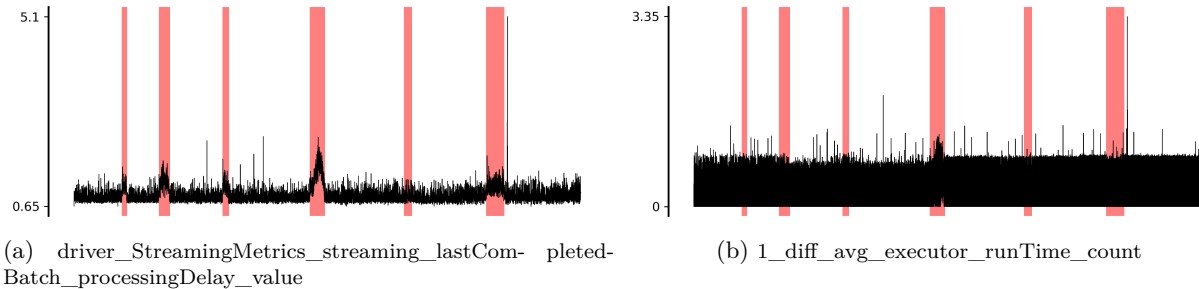

(a) driver_StreamingMetrics_streaming_lastCom-  pleted-Batch_processingDelay_value

(b) 1_diff_avg_executor_runTime_count

Figure 25: Two features from a time series in the test set of application 1 in Exathlon. The anomalies injected in this time series are of the type cpu_contention anomaly.

such example, we believe this instance warrants a closer inspection by experts in the future.

In total, we omit only one additional application. In the test set of application 10, we can observe a strong distributional shift in one feature, see Figure 26. Since this change persists throughout the entire test time

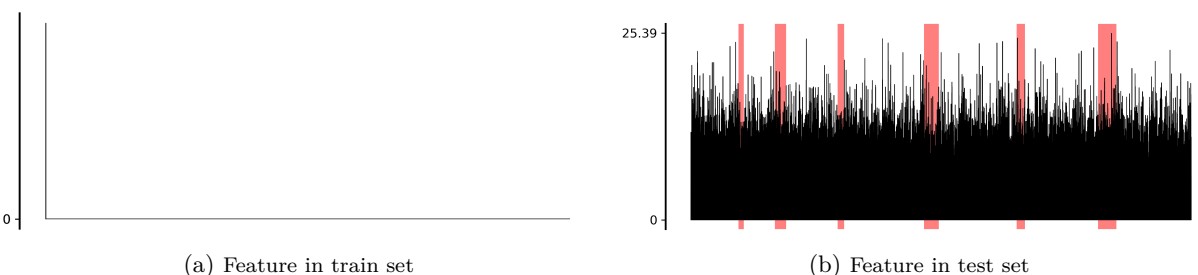

(a) Feature in train set

(b) Feature in test set

Figure 26: Feature 1_diff_avg_executor_shuffleRecordsRead_count of a time series in the test set of application 10.

series independent of any anomalies present, we suspect this might be unintentional. We still report our results on this application in Appendix E, but exclude the application from our main evaluation. Lastly, we also omit application 3 from our main evaluation. The time series in the test set of this application is

---

[4]https://github.com/exathlonbenchmark/exathlon/wiki/Dataset

comparatively short, leaving only about 500 to 1,000 time steps for evaluation folds. Together with sparse anomalies, leaves several folds with no anomalies at all, complicating the evaluation.

## C  Methods

Most approaches, in particular recent ones, rely on a combination of multiple architectural elements. Thus we focus on the method for computing the final anomaly scores. In our setting, we expect each method to compute an anomaly score for each time step based only on the knowledge from prior time steps. Some methods are built to compute anomaly scores for entire time series or windows. For most, we can adapt the method to produce local scores based on the context of a window. We discuss global methods separately at the end of this section. In the following, we will discuss each class of methods in its own section.

### C.1  Reconstruction-based methods

Based on the idea of the classical autoencoder (AE), some methods use an encoder network followed by a decoder network to map the input data into a smaller latent space and back into the input space. The idea is based on the intuition, that the information in the latent space should be enough to reconstruct the input data adequately, and, because the latent space is smaller than the input space, the networks can thus not simply learn an identity function. Since the method is generally only trained on normal data, we expect the reconstruction to fail for anomalous inputs. Thus, such methods rely on the reconstruction error to compute the anomaly score. Most of the time, the mean squared error (MSE) is used to train such methods and is later used for the anomaly score. Since squaring is strictly monotone for non-negative values, the resulting order is equivalent to the absolute error, sometimes used in its stead.

**LSTM-AE**  Malhotra et al. (2016) propose to use an LSTM network as the encoder and as the decoder. The decoder LSTM takes the final hidden state of the encoder LSTM as the initial hidden state and reconstructs the input in reverse order. During training, it uses the true input data as inputs, but during testing, it uses its own predictions.

**LSTM-Max-AE**  Mirza & Cosan (2018) propose to use use the mean or maximum of the hidden states of the encoder instead. Additionally, they use the latent representation as input for all time steps during reconstruction. Contrary to Malhotra et al. (2016), they reconstruct the inputs in the same order.

**MSCRED**  Instead of raw inputs, Zhang et al. (2019) capture the correlation of time-series segments in signature matrices before applying a fully 2D-convolutional network and feeding its output into a 2D-convolutional LSTM encoder and decoder.

**USAD**  Audibert et al. (2020) use two autoencoders with a shared encoder. Training consists of two phases: First, both train to minimize the reconstruction error. Afterward, training shifts to an adversarial setting. Here, the second autoencoder aims to distinguish real samples from those generated by the first autoencoder, whereas the first autoencoder tries to fool the adversary. During inference, a combination of reconstruction and adversarial loss yields the anomaly score for each point.

**TCN-S2S-AE**  Thill et al. (2020) propose a fully convolutional AE architecture with a temporal convolutional network (TCN) in the encoder and a transposed TCN in the decoder. Instead of the usual MSE loss, they use the LogCosh loss as their training objective. A Gaussian is fitted on the errors over the test set during testing. However, this avoids using the method in an online setting. Therefore, we fit the Gaussian to a held-out validation set to be comparable to other methods.

**IDEAL**  Homayouni et al. (2020) propose another LSTM-based AE that determines its ideal window size based on the input time series autocorrelation. However, it seems to us that eq.(2) in the paper has some mistakes, as there is a sum over index $i$ and $i$ is never used, and the authors attempt to compute a confidence interval using the cumulative distribution function of a standard normal distribution instead of its inverse. Furthermore, the authors do not specify any details regarding the dimensionality of the latent space and

how the decoder uses the latent vector to produce the reconstructed sequence. Thus, we cannot implement IDEAL for our library.

**GenAD** Hua et al. (2022) split an input TS into 5 folds of equal size. During training, they mask a random fraction of input features in the last fold by replacing them with the values of another randomly chosen feature. After that, they apply several multi-head self-attention layers to the masked input sequence. Each layer computes attention along the time and feature dimensions separately. Their outputs are interpolated with a learnable weight to produce the final reconstructed sequence. During training, GENAD computes the LogCosh reconstruction loss over the previously masked features. The paper does not clearly describe the detection procedure. Hence, the following is what we implemented in the absence of specific details: We mask each feature in the input TS once and let the GENAD model compute its reconstruction. If the reconstruction error, measured by the LogCosh metric, is larger than some threshold, we consider that feature anomalous. Finally, if more than a predetermined fraction of the input features at a certain point in time is anomalous, we consider the entire TS to be anomalous at that time point.

**STGAT-MAD**[5] Zhan et al. (2022) process an input TS by applying several 1D-convolution layers with different kernel sizes before passing each of the resulting sequences through several graph attention and graph convolution layers in parallel. Then they concatenate the output of those layers and feed them to a bi-LSTM decoder, which attempts to reconstruct the original input TS. STGAT-MAD uses the squared error as both its training loss and anomaly score.

**AnomalyTransformer**[6] Xu et al. (2022) introduce a novel anomaly attention layer that defines a Gaussian prior with learnable bandwidth over the temporal self-attention weights. Their architecture consists of alternating anomaly attention and fully connected layers with residual skip connections. Aside from the MSE between input and reconstructed time series, the authors add a symmetrized KL divergence between the prior and actual attention weights for each anomaly attention layer to the training loss. They update the prior parameters to minimize the KL divergence, whereas the non-prior parameters should maximize it. The anomaly score incorporates both the divergence and reconstruction error.

## C.2 Prediction-based methods

Prediction-based methods—sometimes also called forecasting methods—attempt to predict the next $k \geq 1$ time steps, called prediction horizon, when given an input time series. After training on normal data, they should be capable of accurately predicting the next time steps as long as the input time series and the points that are to be predicted are not anomalous. However, if any point in the prediction horizon is anomalous, the model will usually produce a higher prediction error for those points. Methods in this category use this prediction error as the basis for their anomaly score. Most methods measure the prediction error in terms of the MSE or mean absolute error (MAE).

**LSTM-P** Malhotra et al. (2015) use a multilayer LSTM to extract features and an FC NN to generate l-steps ahead predictions. An MSE loss is used during training, and at inference, a multivariate Gaussian is fitted to the errors of the held-out validation set. Given the learned distribution, the negative log-likelihood corresponds to the anomaly scores.

**LSTM-S2S-P** Similar to LSTM-P, Filonov et al. (2016) use a multilayer LSTM. However, they use the hidden features at each time step to predict the forecast, making their model a sequence-to-sequence predictor. An exponentially weighted moving average of the reconstruction errors yields the anomaly scores.

**DeepANT/TCN-P** Munir et al. (2018) use a TCN with max pooling and an MLP after that to predict the next k points from the input window $x$. They train the model with the MAE. However, the anomaly score is the MSE between a point and its prediction. If the prediction horizon $k > 1$ and there are multiple predictions for a single time step, we take their average and compute the MSE for that.

---

[5]https://github.com/zhanjun717/STGAT
[6]https://github.com/thuml/Anomaly-Transformer

**TCN-S2S-P**   He & Zhao (2019) pass the input window through a dilated causal TCN and concatenate the outputs of the last three layers along the feature dimension to pass this to a final convolution layer with kernel size one and $D$ filters. The output of their method is a window of size $w \times D$ shifted by one time step compared to the input window. TCN-S2S-P uses the MSE loss during training and fits a Gaussian distribution to the prediction errors, just like LSTM-P. Note that during detection, we can only use the last point in the predicted window due to the requirement that the detector must work in an online setting.

**GDN**[7]   Deng & Hooi (2021) construct a graph with features as its nodes and edges representing relations between features. They train an embedding vector for each feature and add directed edges from each feature to the top $m \in \mathbb{N}$ features based on cosine similarity between the feature embeddings. Thus the graph is dynamically recreated for each input batch. After that, they apply a graph attention mechanism (Veličković et al., 2018) to this dynamic graph and pass the outputs to an MLP that returns the prediction for the next time step. The authors use the MSE as their training loss and the MAE, which they normalize using each feature's median and interquartile range, as their anomaly score. They compute the two statistics over the test set, making GDN an offline method. However, computing the statistics over held-out normal data performed poorly due to constant features in the datasets. Hence, we decided to use the unscaled MSE as the anomaly score instead.

## C.3   Generative methods

Generative methods model the data-generating distribution directly by training a generative model on some latent space with a predefined prior that produces samples close to the real data. Those models usually offer some way of computing the marginal likelihood of a data point under the model they learned, which can be used to derive anomaly scores.

### C.3.1   VAE-based methods

**LSTM-VAE**   Sölch et al. (2016) choose both the likelihood $p(x \mid z)$ and the posterior approximation $q(z \mid x)$ to be Gaussian and instantiate all NNs as single-layer LSTMs. Their encoder returns a mean and covariance component for each time step. Furthermore, they use $p(z) = \mathcal{N}(\mu, I)$ as a prior, where $\mu = (\mu_1, \dots, \mu_T)$ is produced by another LSTM. The anomaly score is the negative ELBO.

**Donut**[8]   Xu et al. (2018) chose to use MLPs for both the encoder and decoder. Furthermore, they mask some time steps in the input by setting them to zero. During training, Donut maximizes a modified version of the ELBO that accounts for the input masking. Their anomaly score is the so-called "reconstruction probability" $\mathbb{E}_{z \sim q(z|x)}[-\log p(x \mid z)]$, although they combine it with elaborate mechanisms to reconstruct missing data. However, those are irrelevant to this work since we do not have to deal with missing data. Note that the original formulation only supports univariate TS. We extend it to the multivariate case by simply applying the MLPs to the flattened multivariate input window. We further do not mask entire time steps but random features in random time steps instead.

**LSTM-DVAE**   Park et al. (2018) apply zero-mean Gaussian noise to any input before feeding it to the encoder, and their prior mean for each time step is computed as

$$\mu_t = \left(1 - \frac{t}{T}\right) v_1 + \frac{t}{T} v_T,$$

where $v_1, v_T \in \mathbb{R}^{D'}$ are learnable parameters of the model. Furthermore, they use the reconstruction probability as their anomaly score. Apart from that, the method is exactly the same as the LSTM-VAE.

**GMM-GRU-VAE**   Guo et al. (2018) use GRUs for both their encoder and decoder. Additionally, they chose a Gaussian mixture distribution with $K$ components as their variational posterior approximation.

---

[7] https://github.com/d-ailin/GDN
[8] https://github.com/NetManAIOps/donut

Their prior is also a Gaussian mixture with learnable parameters $\mu_k, \Sigma_k$ for each of the $K$ components. GMM-GRU-VAE uses the reconstruction probability as its anomaly score.

**BI-LSTM-VAE** Pereira & Silveira (2018) propose to use a bi-directional LSTM for both the encoder and decoder. They compute mean and variance for the latent Gaussian distribution from the last hidden state of the encoder. Additionally, the authors apply self-attention to the sequence of encoder hidden states and use the results to instantiate another Gaussian distribution at each time step. The samples from those distributions are combined with the sample from the original latent distribution at each time step to form the input for the decoder. However, the paper does not explicitly say how this sample is combined with the samples from the attention results at each time step. We contacted the authors on this matter but did not receive any response. Thus, we decided not to implement BI-LSTM-VAE as part of our library.

**OmniAnomaly**[9] Su et al. (2019) use a GRU-based encoder and decoder. They also apply a planar normalizing flow (Rezende & Mohamed, 2015) to the latent variable $z$ after they sample it from a multivariate normal distribution with parameters defined by the encoder. Furthermore, they choose a linear Gaussian state space model, i.e., a Kalman filter, for the prior $p(z)$. OmniAnomaly also uses the reconstruction probability as its anomaly score.

**SIS-VAE** Li et al. (2021a) propose another GRU-based VAE. They encourage the VAE to reconstruct smooth TS by adding a KL-divergence term between adjacent time steps term to the ELBO. Intuitively, this term encourages that the distributions for two neighboring points in the predicted TS are similar. Like most other VAE-based methods, SIS-VAE uses the reconstruction probability as its anomaly score.

### C.3.2 GAN-based methods

**BeatGAN**[10] Zhou et al. (2019) use a TCN-based AE as the generator and a TCN-based discriminator. Technically speaking, their method is not really generative since they simply pass the input TS $x$ through a deterministic AE and treat the reconstructed sample $\hat{x}$ as the "generated" input for a GAN discriminator. Furthermore, they only use the reconstruction error (MSE) of the AE as their anomaly score, completely discarding the discriminator after training. Nevertheless, we decided to put BeatGAN in the GAN category because it shares some architectural elements with the other GAN-based approaches. However, it would also be justified to think of BeatGAN as a reconstruction-based method with adversarial regularisation, similar to USAD. The TCN AE trains to minimize the MSE between input $x$ and reconstruction $\hat{x}$ as well as the MSE between their feature maps in the discriminator's second-to-last layer. The discriminator, on the other hand, is trained on the standard GAN loss. Note that the authors augment the input dataset during training by applying dynamic time warping (Vintsyuk, 1968) to each input window and concatenating the resulting distorted window to the original dataset.

**MAD-GAN**[11] Li et al. (2019) use LSTM as a generator and discriminator in their GAN-based approach. Besides the usual discriminator score, they also use a "reconstruction" score. They start with a random latent variable $z \sim \mathcal{N}(0, I)$ and pass it through the generator to obtain $\hat{x}$. Now they use a Gaussian/RBF kernel to compute the similarity between the current input $x$ and the generated sample $\hat{x}$ and use $1-\text{sim}(x, \hat{x})$ as the reconstruction error. They minimize this error using gradient-based methods on $z$ until it falls below a certain threshold. Then, they compute the MAE between the original and reconstructed input and use it as the anomaly score together with the discriminator's output.

**Conv-GAN** Jiang et al. (2019) extract a fixed set of 16 features from an input time series and pass this vector through a fully convolutional AE. Like BeatGAN, they consider the reconstructed vector their "generated" sample. Conv-GANS's discriminator is also a CNN. Furthermore, they add an additional encoder to the generator that takes the reconstructed input and transforms it into the latent space again, trying to match the latent vector of the original AE. However, from table 3 in the paper, it seems that the authors

---

[9]https://github.com/NetManAIOps/OmniAnomaly
[10]https://github.com/hi-bingo/BeatGAN
[11]https://github.com/LiDan456/MAD-GANs

input the 16 extracted features as a $4 \times 4$ matrix into the model, but they do not specify which extracted feature goes where in the matrix. Furthermore, they also write that they do not use any feature extraction on some datasets but do not specify how the model works in that case. Thus, we decided not to implement Conv-GAN.

**LSTM-VAE-GAN**   Niu et al. (2020) use the decoder of an LSTM-based VAE as the generator of a GAN with an LSTM discriminator. Instead of computing the likelihood of the VAE's output on the input $x$ directly, they pass both the original and reconstructed sequences through all but the last layer of the discriminator. The discriminator should not just be capable of detecting a transformed sample generated from a standard normal distribution but also transformed samples from the posterior approximation. Therefore, its loss has an additional term to detect samples from the posterior approximation. LSTM-VAE-GAN's anomaly score is a convex combination of the MAE between $x$ and its reconstruction, and the negative discriminator score.

**TadGAN**[12]   Geiger et al. (2020) propose to use bidirectional LSTMs as decoder and encoder of an AE. Additionally, they consider the decoder and the encoder as generators of two separate Wasserstein GANs (Arjovsky et al., 2017). One GAN uses the decoder as its generator, which maps random samples $z \in \mathcal{N}(0, I)$ to the input data space, and its TCN-based discriminator then attempts to distinguish a real input $x$ from the generated sample $\hat{x}$. However, the generator of the second GAN is the encoder, which maps a data point $x$ to the latent space. The TCN discriminator of that second GAN must now distinguish if its input is a random sample from a standard normal distribution or an encoded data point. Note that the loss function also contains the reconstruction error of the AE measured by the MSE. The authors compute the MSE and the discriminator score during detection and normalize both using their means and standard deviations in the test set. After taking the absolute value, they return a convex combination of the two scores as their final anomaly score. Like TCN-AE, we compute the statistics of both scores during training on a held-out part of the training set instead, turning TADGAN into an online method.

## C.4   Other and hybrid methods

Some methods cannot be directly assigned to one of the classes mentioned above, since they use principles of more than one class. For example, models could compute both reconstruction and prediction errors for an input time series before combining them into a single anomaly score. We decided to place such methods in their own "hybrid" category. Additionally, some methods do not fall into any of the above categories here. This includes, for example, one-class approaches, which are more widely used for AD on other data types.

**LSTM-AE OC-SVM**   Said Elsayed et al. (2020) train an AE with multi-layer LSTMs as encoder and decoder. However, instead of deriving their anomaly score from the reconstruction error of the AE, the authors train an OC-SVM (Schölkopf et al., 2001) on the latent vectors produced by applying the encoder to the held-out clean validation set instead. This OC-SVM then yields the anomaly scores during detection. Note that we return the raw scores, i.e., a point's signed distance from the OC-SVM's separating hyperplane, instead of predictions (0 or 1) to stay consistent with the other methods in this thesis and to avoid putting the OC-SVM at a disadvantage by using a fixed threshold. Unfortunately, the authors do not describe how they pass the latent vector to the decoder in detail, so we decided to use the same architecture as the LSTM-Max-AE. Although this method shares many similarities with some of the reconstruction-based methods (especially LSTM-AE and LSTM-Max-AE), we do not consider it a reconstruction-based method since its anomaly score is not derived from the AE's reconstruction error.

**MTAD-GAT**[13]   Zhao et al. (2020) apply two graph attention modules (Veličković et al., 2018) on top of a TCN, one taking features as nodes and one taking time points in a window as nodes. Both feed their output concatenated to the original input into a GRU. Unlike GDN, they use a fully connected graph as the input and do not build a dynamic graph. The final hidden state of the GRU serves as the input for an MLP to predict the next time point and as the latent variable for a VAE with an MLP decoder. They additively

---

[12]https://github.com/sintel-dev/Orion
[13]The authors included a link to https://github.com/Azure/Multivariate-AD in their paper, claiming that this repository contains their code and data, but as of now (20.12.2022) it is just an empty repository.

combine the MSE of the prediction with the VAE's ELBO loss to train the model. The anomaly score also combines the MSE of the prediction and the reconstruction probability of the VAE using a trade-off coefficient $\gamma \in [0, 1]$.

**THOC** Shen et al. (2020) use a multilayered, dilated RNN to extract features from different temporal scales. For each layer, they attempt to cluster the latent features using a predefined number of centers. The assignment probabilities are then used to fuse the features with features from the previous layers. At the last layer, they then fit an extension of the deep SVDD (Ruff et al., 2018) loss for multiple centers. Additionally, they include the objective of predicting the next time point using the latent features from each layer separately during training. THOC's anomaly score is then the modified SVDD loss without the additional prediction objective.

**NCAD**[14] Carmona et al. (2021) use a TCN network to extract a fixed-size representation from a TS window. The authors derive their anomaly score by comparing representaions for an entire window and the same window without the last $k$ points using an adapted hypersphere classifier loss (Ruff et al., 2020). A high discrepancy between those two representations indicates an anomaly in the last $k$ points of the window. During training, they inject random point anomalies into the training data and augment a batch by randomly replacing some features in the last $k$ points using the values of another time series in the batch and labelling the resulting TS as anomalous. Additionally, they generate new windows by randomly interpolating between existing ones, similar to the mixup (Zhang et al., 2018) procedure.

**GRELEN** Zhang et al. (2022b) use multi-head self-attention along the feature dimension to encode an input TS. More specifically, they compute the attention weights but use a softmax normalization along the attention head axis. Those weights are then used as probabilities for a Gumbel softmax distribution, which the authors sample from. They consider this sample as $h$ adjacency matrices, where $h$ is the number of attention heads. GRELEN uses those adjacency matrices as inputs to a DCGRU (Li et al., 2018b) layer that aims to predict the next time step for each point in the input TS. During training, GRELEN uses a VAE style loss, where the DCGRU output is the mean of a normal distribution with constant variance, and the Gumbel softmax distribution is considered the latent distribution. During testing, GRELEN uses the KL divergence between the latent distribution and a predefined prior. However, the paper lacks many important details, e.g., the value of the constant variance, and we could not understand how a second anomaly score described in the paper works.

## D  Details about our evaluation procedure

Each time-series AD dataset consists of an unlabelled training set $\mathcal{D}_{ul} := \{x^{(1)}, \ldots, x^{(N)}\}$, where each $x^{(i)} \in \mathbb{R}^{T_i \times D}$ is one time series and a labelled test set $\mathcal{D}_l := \{(x^{(1)}, y^{(i)}), \ldots, (x^{(N)}, y^{(N')})\}$, where $x^{(i)} \in \mathbb{R}^{T_i \times D}$ and $y^{(i)} \in \{0, 1\}^{T'_i}$ are the ground truth anomaly labels. We split the unlabelled data into two distinct sets $\mathcal{D}_{train}$ and $\mathcal{D}_{val1}$ such that $\mathcal{D}_{train}$ contains 75% of the available time points and $\mathcal{D}_{val1}$ contains 25%. If $N > 1$, we can achieve this split (approximately) by assigning the entire time series to either set. However, several datasets (e.g., SWaT, WADI, SMD) contain only as a single time series in both their labeled and unlabelled data. Hence, we decided to split each time series along the time dimension and assigned the resulting sub-sequences to $\mathcal{D}_{train}$ and $\mathcal{D}_{val1}$, respectively. We train each method for up to 100 epochs on $\mathcal{D}_{train}$, using early stopping on the validation loss calculated over $\mathcal{D}_{val1}$ for all methods except USAD and the GAN-based approaches. Some methods also require $\mathcal{D}_{val1}$ for fitting parameters of their anomaly detection module, e.g., mean and covariance matrices over reconstruction errors to use in a Gaussian distribution. USAD, BeatGAN, MAD-GAN, and LSTM-VAE-GAN use neither early stopping nor do their detectors require any parameter fitting, so we train them on the entire unlabelled data.

Since we perform a grid search to tune each method's hyperparameters, we also need to split the labeled data into another validation set $\mathcal{D}_{val2}$ and a test set $\mathcal{D}_{test}$. Using a simple split here might introduce an unwanted bias into our evaluation for the case where only one time series of labeled data is available. In this case, the anomalies in the validation set might be of a different type compared to the ones in the test

---

[14]https://github.com/Francois-Aubet/gluon-ts/tree/adding_ncad_to_nursery/src/gluonts/nursery/ncad/src/ncad

set. Since the split is arbitrary, this might put some methods at an unfair disadvantage if we report only the scores on the test set. We cannot entirely eliminate this issue, but we attempt to mitigate it by performing a modified 5-fold cross validation. For that, we split the time series into five equally sized folds and use each fold as the validation set once. The remaining folds, excluding the ones directly next to the validation fold to reduce possible statistical interdependencies, form the test set. We choose the hyperparameters that perform best on the validation set in terms of the best $F_1*$-score and evaluate the corresponding model on the test set. The scores reported in our tables are averages over all five folds. To ensure a fair comparison between methods that incorporate their run time/computational complexity, we adapt the hyperparameter grid size of each method, s.t. it takes roughly 48h to evaluate them on a dataset collection like Exathlon or SMD. We provide the hyperparameter grid for each method as part of our source code repository[15].

All methods use sliding windows as their inputs, although window size and step size may differ between them as we consider them to be hyperparameters. Furthermore, we sub-sample the Exathlon dataset by partitioning each time series into windows of size five and computing the mean over each window.

We implemented all methods and datasets as part of our TimeSeAD library based on PyTorch (Paszke et al., 2019). To keep track of our training and evaluation experiments, we also developed a plugin for our library based on sacred (Greff et al., 2017). This plugin automatically saves all results, configuration, random seeds, and artifacts, e.g., model weights, that our experiments produce. Furthermore, we provide a list of all packages we use with their corresponding version numbers as part of our source code repository[15].

## E   Detailed Benchmark Results

In the following, we present additional results from our benchmark experiments. Table Table 3 shows the ranked average scores for Exathlon and SMD. Here, we average the scores in Exathlon and SMD only over the datasets which we think fit the purpose of evaluating time series AD methods. Details about why and which specific datasets are excluded can be found in Appendix B. To fit on one paper, we use the following abbreviations: $F_1^{pw}$ and $AUPRC^{pw}$ stand for the point-wise best $F_1$ score and area under the precision-recall curve (AUPRC), respectively, with $ts$ as superscript for the appropriate $F_1$, and AUPRC introduce by (Tatbul et al., 2018), and, with $our$ as superscript for our modified metric.

Table 4, Table 5 and Table 6 display detailed results for Exathlon based on point-wise, (Tatbul et al., 2018) and our metric, respectively. Whereas, Table 7 to Table 12 show results on the SMD datasets. For a clear visual appearance, all scores are multiplied by 100. Note that due to spatial constraints, we have limited ourselves only to presenting the SMD datasets, which we find to be best applicable for evaluation. See Appendix B for a detailed analysis of which and why we ignore specific datasets. Full results on all SMD servers will be made available together with the publication of our library.

---

[15]We provide the code in the supplementary material, and later through a GitHub link.

Table 3: Ranked average scores of relevant datasets for Exathlon and SMD on six different evaluation metrics.

| | Exathlon | | | | | | SMD | | | | | |
|---|---|---|---|---|---|---|---|---|---|---|---|---|
| | $F_1^{pw}$ | $AUPRC^{pw}$ | $F_1^{ts}$ | $AUPRC^{ts}$ | $F_1^{our}$ | $AUPRC^{our}$ | $F_1^{pw}$ | $AUPRC^{pw}$ | $F_1^{ts}$ | $AUPRC^{ts}$ | $F_1^{our}$ | $AUPRC^{our}$ |
| LSTM-AE | 21 | 20 | 16 | 22 | 22 | 21 | **1** | **1** | 5 | **1** | **1** | **1** |
| LSTM-Max-AE | 14 | 13 | 14 | 17 | 20 | 16 | 21 | 22 | 21 | 19 | 17 | 20 |
| MSCRED | **1** | **3** | **3** | **1** | **1** | **3** | 16 | 13 | 9 | 13 | 20 | 19 |
| FC-AE | 5 | 4 | 23 | 12 | 10 | 4 | 7 | 9 | 16 | 5 | 7 | 9 |
| USAD | 9 | 5 | 21 | 14 | 15 | 5 | 19 | 17 | 12 | 14 | 15 | 16 |
| TCN-AE | 4 | 17 | **1** | **2** | **3** | 17 | 18 | 24 | **2** | 24 | 21 | 24 |
| GenAD | **3** | **1** | 28 | 4 | 4 | **1** | 24 | 14 | 27 | 12 | 24 | 14 |
| STGAT-MAD | 6 | 6 | 12 | 6 | 14 | 9 | 5 | 4 | 7 | 4 | 5 | 4 |
| AnomalyTransformer | 27 | 27 | 26 | 21 | 27 | 27 | 27 | 27 | 14 | 25 | 27 | 25 |
| LSTM-P | 22 | 22 | 6 | 26 | 25 | 24 | **2** | **2** | 4 | **2** | **2** | **2** |
| LSTM-S2S-P | 18 | 21 | **2** | **3** | 11 | 22 | 13 | 15 | **1** | 10 | 18 | 21 |
| DeepAnt | 10 | 10 | 17 | 9 | 7 | 7 | 10 | 10 | 18 | 11 | 12 | 10 |
| TCN-S2S-P | 19 | 19 | 13 | 13 | 19 | 19 | **3** | **3** | 6 | **3** | **3** | **3** |
| GDN | **2** | **2** | 15 | 10 | **2** | **2** | 9 | 6 | 15 | 9 | 10 | 9 |
| LSTM-VAE | 11 | 11 | 22 | 16 | 6 | 11 | 11 | 11 | 20 | 15 | 9 | 11 |
| Donut | 15 | 16 | 5 | 9 | 16 | 15 | 6 | 5 | 11 | 6 | 6 | 6 |
| LSTM-DVAE | 16 | 15 | 24 | 25 | 17 | 18 | 12 | 19 | 24 | 22 | 13 | 15 |
| GMM-GRU-VAE | 7 | 12 | 18 | 23 | 5 | 10 | 4 | 9 | 10 | 7 | 4 | 5 |
| OmniAnomaly | 23 | 24 | 19 | 24 | 21 | 20 | 15 | 16 | 23 | 21 | 11 | 12 |
| SIS-VAE | 13 | 9 | 11 | 11 | 12 | 6 | 9 | 7 | 17 | 9 | 9 | 7 |
| BeatGAN | 20 | 14 | 27 | 19 | 9 | 9 | 17 | 18 | 19 | 17 | 16 | 17 |
| MAD-GAN | 12 | 9 | 20 | 18 | 18 | 14 | 23 | 23 | **3** | 18 | 23 | 23 |
| LSTM-VAE-GAN | 17 | 18 | 9 | 7 | 13 | 13 | 22 | 20 | 25 | 23 | 19 | 18 |
| TadGAN | 9 | 7 | 10 | 5 | 9 | 12 | 20 | 21 | 9 | 16 | 22 | 22 |
| LSTM-AE OC-SVM | 26 | 25 | 25 | 27 | 26 | 26 | 26 | 26 | 26 | 26 | 26 | 27 |
| MTAD-GAT | 25 | 23 | 7 | 15 | 24 | 23 | 14 | 12 | 22 | 20 | 14 | 13 |
| NCAD | 28 | 28 | 4 | 28 | 28 | 28 | 28 | 28 | 13 | 28 | 28 | 28 |
| THOC | 24 | 26 | 9 | 20 | 23 | 25 | 25 | 25 | 28 | 27 | 25 | 26 |

Table 4: Cross-validation results on Exathlon evaluated with the point wise metric.

| | Best $F_1$-score (point wise) | | | | | | | | | AUPRC (point wise) | | | | | | | | |
| | App ID | | | | | | | | | App ID | | | | | | | | |
| | 1 | 2 | 3 | 4 | 5 | 6 | 9 | 10 | avg(rank) | 1 | 2 | 3 | 4 | 5 | 6 | 9 | 10 | avg(rank) |
|---|---|---|---|---|---|---|---|---|---|---|---|---|---|---|---|---|---|---|
| LSTM-AE | 47.3 | 77.4 | 57.1 | 77.0 | 45.0 | 48.1 | 34.5 | 43.7 | 53.8(20) | 49.5 | 75.0 | 51.7 | 68.5 | 43.8 | 34.9 | 32.3 | 31.6 | 48.4(19) |
| LSTM-Max-AE | 64.0 | 63.1 | 55.4 | 76.1 | 45.1 | 50.8 | 47.2 | 45.2 | 55.9(14) | 69.9 | 60.8 | 51.6 | 73.8 | 42.8 | 36.4 | 38.0 | 38.2 | 51.4(12) |
| MSCRED | 65.0 | 72.3 | 79.1 | 90.5 | 50.4 | 63.1 | 48.9 | 54.3 | 65.4(**1**) | 57.0 | 66.3 | 75.9 | 89.1 | 48.0 | 40.9 | 44.2 | 46.9 | 58.5(**1**) |
| FC-AE | 64.4 | 62.4 | 55.2 | 85.5 | 48.5 | 54.5 | 42.3 | 46.9 | 57.5(5) | 69.7 | 61.6 | 51.3 | 83.9 | 45.9 | 38.9 | 39.7 | 40.4 | 53.9(6) |
| USAD | 61.8 | 62.2 | 49.5 | 87.7 | 52.1 | 53.3 | 38.5 | 48.3 | 56.7(8) | 67.2 | 61.5 | 46.8 | 84.1 | 48.4 | 40.4 | 36.4 | 40.5 | 53.2(8) |
| TCN-AE | 55.8 | 66.5 | 40.7 | 83.6 | 59.2 | 51.7 | 49.0 | 44.8 | 56.4(12) | 52.1 | 56.5 | 35.1 | 72.9 | 57.3 | 32.5 | 42.6 | 34.5 | 47.9(20) |
| GenAD | 68.5 | 57.4 | 42.4 | 91.7 | 50.9 | 68.5 | 34.2 | 38.2 | 56.5(10) | 70.8 | 59.4 | 52.1 | 84.7 | 51.8 | 67.0 | 37.9 | 32.4 | 57.0(**3**) |
| STGAT-MAD | 56.0 | 62.5 | 62.3 | 87.7 | 46.7 | 64.4 | 39.1 | 46.2 | 58.1(**3**) | 60.3 | 61.7 | 58.4 | 83.6 | 45.5 | 50.1 | 36.0 | 39.6 | 54.4(4) |
| AnomalyTransformer | 33.2 | 41.6 | 48.9 | 83.9 | 29.4 | 63.6 | 30.5 | 26.5 | 44.7(27) | 27.0 | 43.4 | 42.7 | 80.3 | 27.1 | 51.4 | 25.3 | 14.4 | 38.9(27) |
| LSTM-P | 47.9 | 71.2 | 46.5 | 75.2 | 47.4 | 38.4 | 41.0 | 43.4 | 51.4(22) | 48.7 | 66.2 | 50.5 | 62.6 | 46.3 | 25.3 | 38.2 | 29.2 | 45.9(23) |
| LSTM-S2S-P | 58.3 | 42.8 | 54.1 | 91.8 | 49.0 | 52.9 | 43.1 | 42.3 | 54.3(18) | 53.8 | 31.6 | 45.6 | 88.8 | 46.8 | 38.0 | 31.1 | 30.5 | 45.8(24) |
| DeepAnt | 57.7 | 60.4 | 45.9 | 89.2 | 50.3 | 55.1 | 41.4 | 46.2 | 55.8(16) | 62.8 | 54.7 | 39.3 | 84.1 | 49.5 | 39.0 | 38.1 | 38.2 | 50.7(14) |
| TCN-S2S-P | 53.1 | 67.5 | 53.3 | 86.0 | 44.8 | 47.7 | 38.1 | 39.6 | 53.8(19) | 56.2 | 62.5 | 52.6 | 77.7 | 44.0 | 32.1 | 35.2 | 28.2 | 48.6(18) |
| GDN | 74.8 | 64.3 | 68.2 | 83.3 | 47.7 | 55.4 | 48.5 | 48.8 | 61.4(**2**) | 79.7 | 62.6 | 65.5 | 80.5 | 47.4 | 38.5 | 43.1 | 42.0 | 57.4(**2**) |
| LSTM-VAE | 47.8 | 62.1 | 59.9 | 84.8 | 63.1 | 60.2 | 33.7 | 46.8 | 57.3(6) | 49.9 | 61.0 | 55.0 | 81.9 | 63.9 | 48.2 | 22.7 | 40.4 | 52.9(10) |
| Donut | 45.2 | 61.2 | 60.4 | 88.9 | 54.2 | 53.3 | 40.1 | 48.2 | 56.4(11) | 48.4 | 54.3 | 56.3 | 83.8 | 53.5 | 38.8 | 36.4 | 41.7 | 51.6(11) |
| LSTM-DVAE | 51.6 | 57.4 | 60.5 | 86.7 | 59.2 | 50.6 | 35.1 | 46.0 | 55.9(13) | 53.0 | 57.5 | 55.8 | 81.8 | 55.5 | 42.2 | 26.9 | 37.8 | 51.3(13) |
| GMM-GRU-VAE | 50.0 | 62.9 | 48.0 | 82.7 | 47.2 | 63.7 | 49.2 | 42.9 | 55.8(15) | 52.0 | 59.0 | 45.8 | 78.2 | 36.1 | 56.9 | 43.0 | 29.8 | 50.1(15) |
| OmniAnomaly | 45.5 | 58.5 | 62.0 | 41.0 | 61.6 | 62.3 | 40.1 | 44.3 | 51.9(21) | 42.9 | 59.0 | 59.7 | 31.1 | 60.3 | 51.2 | 35.7 | 40.0 | 47.5(21) |
| SIS-VAE | 51.9 | 61.7 | 63.9 | 88.3 | 51.1 | 58.8 | 36.8 | 43.8 | 57.0(7) | 57.2 | 61.2 | 59.0 | 84.9 | 48.3 | 44.4 | 33.8 | 38.6 | 53.5(7) |
| BeatGAN | 58.8 | 61.3 | 37.0 | 82.7 | 44.3 | 50.9 | 35.4 | 40.0 | 51.3(23) | 58.9 | 61.7 | 37.6 | 81.7 | 42.8 | 41.4 | 33.2 | 33.9 | 48.9(17) |
| MAD-GAN | 61.3 | 61.7 | 60.1 | 85.9 | 48.8 | 54.9 | 37.8 | 42.6 | 56.6(9) | 66.5 | 63.3 | 56.8 | 79.2 | 45.5 | 38.6 | 36.4 | 38.3 | 53.1(9) |
| LSTM-VAE-GAN | 58.6 | 56.1 | 57.2 | 89.3 | 35.1 | 54.6 | 45.2 | 45.4 | 55.2(17) | 57.8 | 51.4 | 52.4 | 87.5 | 30.4 | 41.0 | 40.8 | 33.6 | 49.4(16) |
| TadGAN | 74.8 | 62.6 | 65.3 | 80.1 | 48.9 | 50.3 | 38.4 | 44.1 | 58.1(4) | 79.4 | 61.4 | 60.1 | 77.8 | 44.6 | 37.7 | 34.3 | 37.3 | 54.1(5) |
| LSTM-AE OC-SVM | 51.3 | 67.2 | 36.3 | 73.8 | 41.4 | 38.7 | 35.0 | 40.8 | 48.0(26) | 52.7 | 64.4 | 29.7 | 63.0 | 35.4 | 30.6 | 32.0 | 35.1 | 42.8(26) |
| MTAD-GAT | 50.3 | 60.2 | 49.5 | 51.3 | 51.2 | 57.6 | 37.7 | 46.4 | 50.5(24) | 55.0 | 61.0 | 45.3 | 39.3 | 47.3 | 47.4 | 32.3 | 40.4 | 46.0(22) |
| NCAD | 29.7 | 23.1 | 23.2 | 22.2 | 16.7 | 19.0 | 26.2 | 24.5 | 23.1(28) | 13.0 | 8.6 | 9.8 | 12.2 | 6.3 | 10.5 | 16.2 | 12.1 | 11.1(28) |
| THOC | 32.5 | 54.7 | 41.1 | 90.5 | 37.1 | 47.4 | 46.4 | 49.3 | 49.9(25) | 22.8 | 50.4 | 30.8 | 89.6 | 29.0 | 36.3 | 44.6 | 42.7 | 43.3(25) |

Table 5: Cross-validation results on Exathlon evaluated with the metric from Tatbul et al. (2018).

| | Best $F_1$-score (Tatbul et al.) | | | | | | | | | AUPRC (Tatbul et al.) | | | | | | | | |
| | App ID | | | | | | | | | | App ID | | | | | | | | |
| | 1 | 2 | 3 | 4 | 5 | 6 | 9 | 10 | avg(rank) | 1 | 2 | 3 | 4 | 5 | 6 | 9 | 10 | avg(rank) |
|---|---|---|---|---|---|---|---|---|---|---|---|---|---|---|---|---|---|---|
| LSTM-AE | 60.5 | 47.2 | 28.4 | 27.6 | 27.2 | 44.1 | 40.0 | 57.4 | 41.5(22) | 43.9 | 51.7 | 28.0 | 51.1 | 30.8 | 42.4 | 22.9 | 22.9 | 36.7(23) |
| LSTM-Max-AE | 61.6 | 47.6 | 45.0 | 35.3 | 17.3 | 25.5 | 61.7 | 66.0 | 45.0(12) | 49.0 | 41.8 | 39.2 | 73.0 | 31.5 | 33.6 | 27.7 | 24.6 | 40.0(14) |
| MSCRED | 68.6 | 69.4 | 76.4 | 72.1 | 48.7 | 64.9 | 77.0 | 56.1 | 66.7(**2**) | 67.2 | 60.7 | 60.9 | 86.7 | 49.1 | 58.1 | 52.0 | 39.0 | 59.2(**1**) |
| FC-AE | 62.3 | 37.9 | 44.8 | 33.8 | 20.9 | 32.5 | 45.1 | 61.0 | 42.3(15) | 50.4 | 38.8 | 30.6 | 71.9 | 34.3 | 44.6 | 27.1 | 24.9 | 40.3(13) |
| USAD | 51.0 | 38.9 | 54.7 | 38.2 | 26.5 | 38.3 | 45.3 | 47.8 | 42.6(14) | 49.7 | 39.8 | 39.7 | 72.1 | 37.5 | 42.4 | 23.4 | 25.7 | 41.3(12) |
| TCN-AE | 65.3 | 69.5 | 64.4 | 79.3 | 77.3 | 66.9 | 75.2 | 72.3 | 71.3(**1**) | 56.3 | 50.7 | 34.1 | 76.0 | 58.9 | 47.5 | 48.8 | 40.2 | 51.6(**2**) |
| GenAD | 38.1 | 34.8 | 37.5 | 54.3 | 19.3 | 34.0 | 16.4 | 34.3 | 33.6(28) | 53.4 | 47.1 | 42.6 | 81.9 | 38.4 | 75.2 | 28.8 | 25.1 | 49.1(4) |
| STGAT-MAD | 60.9 | 47.7 | 41.9 | 33.4 | 25.2 | 41.2 | 49.4 | 69.9 | 46.2(8) | 46.7 | 43.8 | 36.4 | 72.4 | 35.7 | 59.6 | 24.4 | 26.1 | 43.1(7) |
| AnomalyTransformer | 45.1 | 15.4 | 30.0 | 43.4 | 13.3 | 56.3 | 56.1 | 21.0 | 35.1(27) | 33.3 | 34.4 | 32.9 | 73.7 | 23.2 | 52.7 | 29.0 | 13.8 | 36.6(24) |
| LSTM-P | 57.7 | 49.7 | 39.1 | 26.4 | 26.7 | 45.1 | 79.0 | 68.5 | 49.1(6) | 43.0 | 48.9 | 17.0 | 48.1 | 34.5 | 27.9 | 32.7 | 27.7 | 35.0(26) |
| LSTM-S2S-P | 60.5 | 75.0 | 60.9 | 82.2 | 75.4 | 58.6 | 55.5 | 55.7 | 65.5(**3**) | 54.6 | 43.7 | 35.2 | 89.8 | 60.4 | 44.8 | 39.0 | 35.8 | 50.4(**3**) |
| DeepAnt | 60.5 | 44.5 | 46.2 | 38.2 | 32.9 | 38.5 | 31.2 | 46.4 | 42.3(16) | 49.2 | 41.7 | 29.3 | 72.6 | 41.3 | 49.9 | 24.8 | 24.9 | 41.7(11) |
| TCN-S2S-P | 54.8 | 52.8 | 36.3 | 27.5 | 32.4 | 45.1 | 42.7 | 61.1 | 44.1(13) | 45.9 | 44.8 | 25.3 | 64.8 | 36.2 | 45.7 | 27.8 | 20.9 | 38.9(17) |
| GDN | 61.1 | 41.8 | 35.9 | 32.9 | 24.1 | 38.2 | 48.6 | 50.9 | 41.7(20) | 57.3 | 38.5 | 39.9 | 69.0 | 36.2 | 44.2 | 31.7 | 24.9 | 42.7(8) |
| LSTM-VAE | 51.1 | 42.1 | 39.9 | 23.8 | 31.8 | 38.8 | 48.5 | 58.2 | 41.8(18) | 39.8 | 42.6 | 30.0 | 66.6 | 42.5 | 50.1 | 16.6 | 22.4 | 38.8(18) |
| Donut | 58.4 | 59.1 | 42.8 | 32.6 | 31.1 | 48.6 | 78.9 | 65.4 | 52.1(5) | 45.0 | 45.2 | 33.5 | 67.4 | 36.3 | 50.1 | 34.7 | 24.0 | 42.0(9) |
| LSTM-DVAE | 50.2 | 34.7 | 41.3 | 30.9 | 30.2 | 33.8 | 51.1 | 48.4 | 40.1(24) | 45.4 | 31.6 | 36.1 | 68.3 | 37.6 | 37.3 | 18.8 | 21.5 | 37.1(22) |
| GMM-GRU-VAE | 52.6 | 35.3 | 33.8 | 26.8 | 25.2 | 42.8 | 60.2 | 56.4 | 41.7(21) | 37.2 | 38.7 | 22.9 | 59.1 | 27.3 | 45.4 | 33.8 | 20.1 | 35.6(25) |
| OmniAnomaly | 58.6 | 35.3 | 43.3 | 21.6 | 33.8 | 36.1 | 57.3 | 42.1 | 41.0(23) | 52.0 | 40.0 | 39.9 | 27.7 | 43.1 | 50.2 | 26.9 | 20.2 | 37.5(20) |
| SIS-VAE | 56.1 | 33.9 | 43.5 | 36.0 | 33.9 | 39.8 | 63.5 | 68.5 | 46.9(7) | 43.5 | 39.5 | 38.3 | 73.3 | 39.0 | 50.4 | 28.6 | 23.3 | 42.0(10) |
| BeatGAN | 28.2 | 58.0 | 55.9 | 35.2 | 22.2 | 25.0 | 39.8 | 38.9 | 37.9(26) | 37.1 | 43.0 | 19.8 | 75.1 | 36.3 | 40.2 | 22.3 | 24.5 | 37.3(21) |
| MAD-GAN | 45.5 | 38.6 | 40.8 | 26.7 | 48.5 | 32.2 | 46.8 | 55.2 | 41.8(17) | 47.5 | 42.4 | 37.1 | 65.9 | 38.8 | 38.1 | 23.3 | 22.9 | 39.5(16) |
| LSTM-VAE-GAN | 58.4 | 56.0 | 46.4 | 45.1 | 30.2 | 40.1 | 43.5 | 49.7 | 46.2(9) | 50.1 | 45.7 | 35.9 | 77.9 | 30.6 | 42.4 | 34.6 | 28.9 | 43.3(6) |
| TadGAN | 56.7 | 48.4 | 49.9 | 43.1 | 47.4 | 32.1 | 38.2 | 49.4 | 45.7(10) | 60.9 | 47.1 | 38.8 | 75.8 | 45.7 | 39.8 | 29.4 | 26.5 | 45.5(5) |
| LSTM-AE OC-SVM | 50.5 | 30.4 | 36.9 | 49.0 | 30.1 | 37.3 | 32.6 | 46.4 | 39.1(25) | 39.8 | 38.0 | 30.5 | 55.6 | 34.0 | 30.2 | 22.0 | 19.7 | 33.7(27) |
| MTAD-GAT | 62.4 | 43.3 | 38.2 | 35.2 | 40.4 | 39.5 | 55.8 | 47.2 | 45.2(11) | 45.7 | 44.9 | 29.7 | 60.8 | 36.1 | 46.0 | 27.5 | 25.6 | 39.5(15) |
| NCAD | 79.1 | 64.1 | 58.3 | 54.8 | 52.6 | 38.7 | 68.0 | 47.9 | 57.9(4) | 46.2 | 27.5 | 20.4 | 52.8 | 27.2 | 15.0 | 23.8 | 18.1 | 28.9(28) |
| THOC | 66.4 | 37.6 | 40.9 | 31.7 | 37.1 | 41.9 | 54.6 | 23.5 | 41.7(19) | 30.0 | 33.5 | 25.1 | 78.1 | 33.2 | 42.1 | 36.8 | 21.4 | 37.5(19) |

Table 6: Cross-validation results on Exathlon evaluated with our metric.

| | Best $F_1$-score (ours) | | | | | | | | | AUPRC (ours) | | | | | | | | |
| | App ID | | | | | | | | | App ID | | | | | | | | |
| | 1 | 2 | 3 | 4 | 5 | 6 | 9 | 10 | avg(rank) | 1 | 2 | 3 | 4 | 5 | 6 | 9 | 10 | avg(rank) |
|---|---|---|---|---|---|---|---|---|---|---|---|---|---|---|---|---|---|---|
| LSTM-AE | 53.7 | 64.7 | 60.6 | 74.8 | 43.7 | 48.7 | 31.7 | 44.2 | 52.8(23) | 56.3 | 63.1 | 42.9 | 66.6 | 42.4 | 36.0 | 29.1 | 35.7 | 46.5(21) |
| LSTM-Max-AE | 67.3 | 50.8 | 59.1 | 73.2 | 44.0 | 51.7 | 41.5 | 37.4 | 53.1(21) | 72.2 | 49.3 | 53.6 | 72.7 | 44.4 | 40.4 | 33.9 | 31.6 | 49.7(16) |
| MSCRED | 64.2 | 64.7 | 84.0 | 90.9 | 49.0 | 63.5 | 50.1 | 54.8 | 65.1(**1**) | 56.4 | 62.4 | 29.8 | 89.6 | 47.7 | 41.6 | 44.4 | 48.3 | 52.5(7) |
| FC-AE | 68.1 | 52.1 | 71.3 | 84.0 | 47.4 | 55.1 | 38.9 | 42.6 | 57.4(5) | 72.8 | 52.7 | 69.9 | 85.4 | 47.8 | 42.7 | 37.8 | 36.7 | 55.8(**3**) |
| USAD | 65.5 | 52.6 | 54.6 | 85.6 | 51.3 | 53.4 | 34.4 | 43.8 | 55.1(16) | 70.4 | 53.4 | 56.5 | 84.3 | 50.0 | 43.6 | 32.9 | 36.8 | 53.5(4) |
| TCN-AE | 64.8 | 57.3 | 61.9 | 83.1 | 52.8 | 52.2 | 49.8 | 41.5 | 57.9(**3**) | 59.6 | 51.2 | 36.9 | 73.0 | 51.8 | 32.6 | 41.2 | 29.3 | 47.0(19) |
| GenAD | 70.6 | 47.7 | 42.4 | 89.7 | 52.3 | 68.0 | 25.2 | 33.1 | 53.6(19) | 74.5 | 53.1 | 51.6 | 87.2 | 53.1 | 67.4 | 35.7 | 32.3 | 56.8(**2**) |
| STGAT-MAD | 59.5 | 52.6 | 69.3 | 86.2 | 43.9 | 66.3 | 34.7 | 40.8 | 56.7(11) | 64.2 | 53.3 | 55.7 | 84.7 | 43.3 | 51.4 | 33.3 | 35.5 | 52.7(6) |
| AnomalyTransformer | 40.1 | 33.3 | 44.3 | 81.0 | 28.9 | 63.9 | 29.2 | 24.9 | 43.2(27) | 35.3 | 37.1 | 44.0 | 77.6 | 26.3 | 50.8 | 24.6 | 16.7 | 39.1(27) |
| LSTM-P | 55.1 | 54.1 | 68.6 | 73.2 | 52.8 | 38.3 | 37.3 | 43.1 | 52.8(22) | 57.5 | 53.2 | 55.2 | 61.4 | 51.5 | 26.0 | 34.5 | 32.6 | 46.5(22) |
| LSTM-S2S-P | 61.8 | 52.2 | 60.5 | 91.2 | 41.4 | 55.2 | 43.5 | 38.1 | 55.5(14) | 55.1 | 34.9 | 53.1 | 87.0 | 39.5 | 37.0 | 32.0 | 30.3 | 46.1(24) |
| DeepAnt | 63.4 | 54.8 | 64.7 | 87.3 | 51.0 | 56.0 | 36.6 | 41.0 | 56.8(10) | 66.9 | 50.5 | 47.2 | 85.1 | 49.9 | 42.9 | 36.2 | 33.4 | 51.5(10) |
| TCN-S2S-P | 58.3 | 56.5 | 68.8 | 83.7 | 50.0 | 47.7 | 34.6 | 38.4 | 54.8(17) | 62.0 | 54.7 | 35.2 | 76.0 | 48.1 | 32.1 | 33.2 | 30.9 | 46.5(20) |
| GDN | 76.5 | 56.9 | 69.4 | 81.7 | 46.6 | 57.9 | 46.1 | 42.5 | 59.7(**2**) | 80.4 | 55.4 | 68.7 | 82.0 | 45.1 | 44.3 | 40.5 | 38.4 | 56.8(**1**) |
| LSTM-VAE | 54.7 | 53.0 | 68.6 | 83.2 | 65.5 | 59.1 | 34.6 | 40.3 | 57.4(6) | 57.9 | 53.5 | 40.4 | 81.8 | 65.3 | 47.2 | 22.8 | 37.6 | 50.8(14) |
| Donut | 54.2 | 50.9 | 71.0 | 86.8 | 57.0 | 51.9 | 39.3 | 44.5 | 56.9(8) | 56.5 | 46.7 | 51.8 | 82.4 | 57.3 | 37.5 | 33.6 | 41.9 | 51.0(12) |
| LSTM-DVAE | 56.2 | 50.0 | 67.9 | 81.3 | 63.8 | 47.8 | 34.4 | 42.4 | 55.5(15) | 55.6 | 48.4 | 53.8 | 77.2 | 61.1 | 39.5 | 26.4 | 38.2 | 50.0(15) |
| GMM-GRU-VAE | 54.7 | 54.7 | 67.3 | 79.4 | 53.5 | 61.9 | 47.9 | 40.5 | 57.5(4) | 58.2 | 52.8 | 53.9 | 76.6 | 42.1 | 56.2 | 42.8 | 34.1 | 52.1(9) |
| OmniAnomaly | 53.4 | 51.6 | 69.5 | 44.7 | 68.3 | 62.3 | 37.8 | 39.5 | 53.4(20) | 53.5 | 52.7 | 57.2 | 37.9 | 67.4 | 51.7 | 33.4 | 35.4 | 48.7(18) |
| SIS-VAE | 56.8 | 52.7 | 70.3 | 88.2 | 53.0 | 61.1 | 32.4 | 38.8 | 56.6(12) | 61.6 | 53.3 | 55.9 | 87.1 | 52.1 | 48.7 | 30.8 | 33.9 | 52.9(5) |
| BeatGAN | 65.9 | 64.7 | 68.9 | 82.3 | 44.0 | 54.3 | 36.5 | 38.4 | 56.9(9) | 67.1 | 55.4 | 31.0 | 81.0 | 49.1 | 44.4 | 33.5 | 32.7 | 49.3(17) |
| MAD-GAN | 64.4 | 52.7 | 67.9 | 84.0 | 43.3 | 56.5 | 32.1 | 36.2 | 54.6(18) | 68.3 | 54.4 | 54.7 | 79.3 | 42.4 | 43.2 | 32.4 | 33.1 | 51.0(13) |
| LSTM-VAE-GAN | 63.9 | 55.7 | 62.1 | 89.3 | 39.5 | 54.0 | 41.6 | 44.7 | 56.3(13) | 67.0 | 49.7 | 49.1 | 89.7 | 35.4 | 42.5 | 37.9 | 38.3 | 51.2(11) |
| TadGAN | 78.6 | 58.8 | 70.4 | 77.7 | 46.1 | 52.7 | 33.0 | 39.5 | 57.1(7) | 82.7 | 54.3 | 55.1 | 75.1 | 44.0 | 40.8 | 31.5 | 34.3 | 52.2(8) |
| LSTM-AE OC-SVM | 57.5 | 54.9 | 46.0 | 71.7 | 38.5 | 39.7 | 29.1 | 37.2 | 46.8(26) | 58.2 | 53.1 | 48.9 | 63.5 | 33.3 | 32.4 | 27.3 | 34.2 | 43.9(26) |
| MTAD-GAT | 54.6 | 54.0 | 67.7 | 53.0 | 55.5 | 56.4 | 40.0 | 36.6 | 52.2(25) | 59.5 | 53.4 | 52.0 | 42.4 | 50.0 | 46.6 | 33.3 | 34.3 | 46.4(23) |
| NCAD | 34.1 | 23.1 | 38.9 | 22.3 | 17.8 | 19.9 | 26.5 | 27.2 | 26.2(28) | 17.5 | 7.3 | 21.1 | 12.6 | 6.8 | 11.4 | 18.2 | 16.9 | 14.0(28) |
| THOC | 41.3 | 45.9 | 59.3 | 89.9 | 43.7 | 43.6 | 49.3 | 45.9 | 52.4(24) | 32.7 | 43.8 | 41.3 | 91.4 | 37.4 | 33.0 | 45.4 | 43.3 | 46.0(25) |

Table 7: Cross-validation results on SMD evaluated with point-wise F1.

| | \multicolumn{15}{c}{Server ID} | |
| | 1 | 6 | 8 | 9 | 10 | 11 | 13 | 14 | 16 | 17 | 20 | 21 | 24 | 26 | 27 | avg(rank) |
|---|---|---|---|---|---|---|---|---|---|---|---|---|---|---|---|---|
| LSTM-AE | 47.1 | 68.9 | 29.4 | 33.9 | 60.4 | 30.6 | 56.8 | 64.5 | 51.9 | 74.6 | 14.7 | 42.3 | 65.3 | 23.4 | 78.1 | 49.5(**1**) |
| LSTM-Max-AE | 52.7 | 32.2 | 23.0 | 33.9 | 33.0 | 29.7 | 58.1 | 44.4 | 35.8 | 56.2 | 14.0 | 11.8 | 52.4 | 25.4 | 61.3 | 37.6(21) |
| MSCRED | 53.9 | 49.5 | 43.7 | 28.0 | 40.2 | 22.1 | 48.8 | 37.6 | 48.2 | 60.8 | 12.3 | 24.4 | 47.5 | 20.5 | 57.7 | 39.7(16) |
| FC-AE | 49.0 | 56.4 | 30.7 | 34.2 | 48.7 | 27.7 | 56.9 | 58.4 | 44.8 | 64.8 | 14.4 | 32.7 | 57.0 | 23.5 | 79.4 | 45.2(7) |
| USAD | 37.7 | 48.6 | 20.9 | 33.8 | 35.7 | 38.0 | 56.9 | 58.6 | 34.6 | 56.8 | 13.5 | 27.9 | 46.6 | 17.9 | 58.4 | 39.1(19) |
| TCN-AE | 47.0 | 52.7 | 36.6 | 23.9 | 33.9 | 23.6 | 44.7 | 35.9 | 44.4 | 63.3 | 16.8 | 24.2 | 55.2 | 31.5 | 52.8 | 39.1(18) |
| GenAD | 44.6 | 21.0 | 12.7 | 26.6 | 22.4 | 15.7 | 52.5 | 46.7 | 31.2 | 55.2 | 10.5 | 7.1 | 48.4 | 21.1 | 59.6 | 31.7(24) |
| STGAT-MAD | 48.1 | 64.5 | 24.2 | 34.7 | 55.9 | 26.6 | 56.7 | 58.1 | 49.5 | 65.6 | 15.9 | 33.0 | 59.5 | 25.4 | 78.2 | 46.4(5) |
| AnomalyTransformer | 22.9 | 49.1 | 21.8 | 20.6 | 25.7 | 24.8 | 11.5 | 21.0 | 25.1 | 27.8 | 11.4 | 27.1 | 49.6 | 23.7 | 13.8 | 25.1(27) |
| LSTM-P | 54.7 | 73.8 | 36.5 | 32.7 | 58.2 | 31.8 | 56.6 | 63.0 | 49.5 | 69.5 | 14.4 | 37.1 | 59.2 | 23.9 | 78.7 | 49.3(**2**) |
| LSTM-S2S-P | 53.0 | 54.5 | 43.1 | 27.2 | 28.2 | 28.7 | 43.8 | 34.8 | 47.7 | 54.4 | 14.4 | 34.8 | 58.0 | 29.5 | 55.7 | 40.5(13) |
| DeepAnt | 50.6 | 60.4 | 26.8 | 35.0 | 42.2 | 28.9 | 56.2 | 59.6 | 45.7 | 61.4 | 14.3 | 30.0 | 60.6 | 24.8 | 61.0 | 43.8(10) |
| TCN-S2S-P | 41.9 | 73.2 | 34.6 | 34.1 | 52.5 | 31.9 | 54.0 | 55.1 | 51.5 | 78.3 | 16.2 | 36.3 | 60.7 | 26.0 | 80.9 | 48.5(**3**) |
| GDN | 58.9 | 58.6 | 32.0 | 34.2 | 53.9 | 26.3 | 56.4 | 56.6 | 43.9 | 63.3 | 13.4 | 30.0 | 59.9 | 20.2 | 68.2 | 45.0(9) |
| LSTM-VAE | 45.7 | 62.1 | 22.9 | 26.7 | 60.5 | 32.0 | 57.7 | 54.1 | 44.9 | 50.0 | 14.5 | 39.2 | 57.2 | 21.9 | 63.9 | 43.6(11) |
| Donut | 42.7 | 62.8 | 27.6 | 35.4 | 63.2 | 40.0 | 50.1 | 60.9 | 40.6 | 70.7 | 17.1 | 29.5 | 64.3 | 14.5 | 74.1 | 46.2(6) |
| LSTM-DVAE | 40.1 | 61.3 | 24.0 | 23.4 | 51.7 | 35.7 | 52.8 | 62.2 | 42.4 | 39.1 | 15.7 | 38.0 | 58.7 | 22.9 | 49.0 | 41.1(12) |
| GMM-GRU-VAE | 44.4 | 64.2 | 25.3 | 34.1 | 62.0 | 36.0 | 55.6 | 58.7 | 40.3 | 71.2 | 16.1 | 32.1 | 60.2 | 22.9 | 77.7 | 46.7(4) |
| OmniAnomaly | 38.9 | 63.1 | 23.6 | 33.3 | 32.7 | 31.2 | 60.6 | 50.8 | 50.1 | 42.1 | 13.7 | 24.1 | 54.1 | 13.8 | 73.4 | 40.4(15) |
| SIS-VAE | 53.9 | 63.0 | 27.5 | 34.6 | 47.4 | 27.4 | 56.9 | 59.8 | 43.5 | 64.0 | 15.0 | 28.8 | 55.4 | 23.7 | 75.8 | 45.1(8) |
| BeatGAN | 44.5 | 48.8 | 24.1 | 33.2 | 45.5 | 33.2 | 57.0 | 45.1 | 38.3 | 58.8 | 13.4 | 24.7 | 42.8 | 23.0 | 57.8 | 39.3(17) |
| MAD-GAN | 41.5 | 33.9 | 30.2 | 33.9 | 44.6 | 20.3 | 38.8 | 45.7 | 42.4 | 47.6 | 15.4 | 9.8 | 32.1 | 25.7 | 32.7 | 33.0(23) |
| LSTM-VAE-GAN | 47.8 | 45.0 | 21.2 | 36.7 | 51.8 | 25.3 | 58.0 | 35.9 | 28.4 | 51.6 | 14.0 | 19.6 | 31.3 | 27.7 | 60.6 | 37.0(22) |
| TadGAN | 49.7 | 28.2 | 14.1 | 33.9 | 51.1 | 31.1 | 58.3 | 45.8 | 35.3 | 54.9 | 14.9 | 21.9 | 54.3 | 27.0 | 48.9 | 38.0(20) |
| LSTM-AE OC-SVM | 16.9 | 25.2 | 13.7 | 19.2 | 18.3 | 31.5 | 22.0 | 35.2 | 19.7 | 50.5 | 14.6 | 10.7 | 40.3 | 27.6 | 56.7 | 26.8(26) |
| MTAD-GAT | 40.3 | 67.8 | 13.2 | 33.2 | 51.6 | 50.2 | 44.8 | 47.7 | 40.7 | 39.8 | 12.6 | 31.9 | 50.9 | 26.5 | 55.1 | 40.4(14) |
| NCAD | 11.6 | 21.4 | 11.4 | 18.4 | 9.7 | 12.3 | 4.1 | 3.9 | 8.6 | 7.1 | 10.7 | 13.9 | 7.1 | 9.6 | 13.3 | 10.9(28) |
| THOC | 52.7 | 37.6 | 20.6 | 23.8 | 23.9 | 24.2 | 25.6 | 35.1 | 21.7 | 17.1 | 11.2 | 18.8 | 31.9 | 20.5 | 54.0 | 27.9(25) |

Table 8: Cross-validation results on SMD evaluated with point-wise AUPRC.

| | 1 | 6 | 8 | 9 | 10 | 11 | 13 | 14 | 16 | 17 | 20 | 21 | 24 | 26 | 27 | avg(rank) |
|---|---|---|---|---|---|---|---|---|---|---|---|---|---|---|---|---|
| | | | | | | | | | Server ID | | | | | | | |
| LSTM-AE | 32.9 | 68.9 | 23.6 | 23.4 | 59.3 | 26.8 | 48.7 | 62.8 | 51.9 | 73.7 | 8.3 | 32.2 | 62.4 | 23.8 | 76.8 | 45.0(**1**) |
| LSTM-Max-AE | 41.6 | 17.2 | 20.3 | 23.2 | 21.5 | 26.2 | 49.4 | 35.6 | 35.9 | 54.0 | 6.5 | 7.0 | 48.9 | 24.6 | 57.5 | 31.3(22) |
| MSCRED | 48.3 | 36.6 | 41.0 | 15.0 | 25.7 | 16.2 | 42.8 | 47.4 | 45.6 | 60.2 | 5.7 | 19.3 | 47.1 | 17.4 | 55.5 | 34.9(13) |
| FC-AE | 42.0 | 46.1 | 26.2 | 24.7 | 41.8 | 22.8 | 47.1 | 51.7 | 42.7 | 63.5 | 5.4 | 24.1 | 53.6 | 23.6 | 77.6 | 39.5(9) |
| USAD | 31.3 | 39.6 | 15.7 | 21.3 | 29.7 | 36.8 | 46.8 | 50.4 | 28.4 | 54.4 | 7.1 | 19.5 | 42.6 | 18.5 | 54.3 | 33.1(17) |
| TCN-AE | 27.4 | 36.8 | 24.1 | 13.3 | 14.4 | 17.4 | 22.2 | 21.8 | 41.1 | 57.9 | 8.8 | 13.3 | 36.1 | 27.6 | 33.9 | 26.4(24) |
| GenAD | 40.6 | 21.3 | 25.8 | 19.3 | 35.3 | 23.0 | 49.7 | 55.5 | 34.6 | 60.9 | 8.7 | 12.6 | 50.5 | 20.8 | 55.4 | 34.3(14) |
| STGAT-MAD | 38.2 | 56.8 | 22.0 | 25.2 | 50.9 | 23.2 | 48.9 | 55.1 | 49.4 | 65.0 | 8.2 | 24.5 | 59.9 | 25.6 | 73.8 | 41.8(4) |
| AnomalyTransformer | 25.0 | 37.6 | 15.7 | 14.3 | 23.8 | 17.0 | 11.1 | 11.4 | 19.6 | 26.5 | 6.1 | 18.5 | 42.6 | 22.0 | 7.7 | 19.9(27) |
| LSTM-P | 48.3 | 71.8 | 29.4 | 23.7 | 54.9 | 29.7 | 49.1 | 54.2 | 50.2 | 67.6 | 7.6 | 28.0 | 59.5 | 23.1 | 75.5 | 44.8(**2**) |
| LSTM-S2S-P | 50.2 | 37.0 | 39.0 | 16.3 | 20.1 | 19.2 | 39.8 | 28.4 | 46.2 | 50.4 | 7.1 | 27.8 | 49.2 | 26.2 | 52.2 | 34.0(15) |
| DeepAnt | 42.0 | 52.3 | 21.3 | 25.2 | 35.2 | 23.3 | 48.4 | 52.9 | 43.6 | 58.3 | 6.5 | 18.7 | 61.7 | 24.0 | 58.7 | 38.1(10) |
| TCN-S2S-P | 31.0 | 72.3 | 27.2 | 24.0 | 47.2 | 29.2 | 48.0 | 46.7 | 51.6 | 76.7 | 9.1 | 27.1 | 60.5 | 23.5 | 77.3 | 43.4(**3**) |
| GDN | 50.6 | 50.6 | 28.4 | 24.0 | 50.7 | 23.8 | 48.6 | 48.9 | 40.6 | 61.3 | 5.9 | 19.4 | 61.6 | 20.2 | 64.7 | 40.0(6) |
| LSTM-VAE | 27.1 | 51.1 | 17.4 | 19.3 | 48.3 | 31.4 | 50.3 | 43.5 | 41.8 | 47.6 | 7.2 | 24.8 | 53.0 | 17.1 | 60.7 | 36.0(11) |
| Donut | 27.1 | 52.1 | 19.7 | 26.8 | 59.1 | 33.1 | 41.2 | 53.8 | 40.4 | 70.6 | 10.7 | 21.1 | 66.0 | 12.0 | 71.5 | 40.3(5) |
| LSTM-DVAE | 22.8 | 51.1 | 16.6 | 14.7 | 34.3 | 34.9 | 45.7 | 45.4 | 39.5 | 32.9 | 6.5 | 24.1 | 53.2 | 19.1 | 43.0 | 32.2(19) |
| GMM-GRU-VAE | 31.8 | 53.8 | 19.2 | 26.4 | 44.9 | 35.7 | 47.2 | 51.2 | 35.9 | 68.3 | 6.8 | 17.3 | 61.3 | 22.3 | 74.8 | 39.8(8) |
| OmniAnomaly | 22.2 | 54.1 | 17.7 | 26.5 | 18.0 | 28.4 | 52.2 | 36.8 | 52.0 | 41.3 | 8.3 | 12.5 | 56.2 | 10.0 | 70.5 | 33.8(16) |
| SIS-VAE | 43.8 | 57.9 | 22.2 | 24.8 | 41.5 | 22.3 | 49.2 | 53.6 | 42.7 | 60.9 | 6.4 | 19.8 | 55.0 | 24.1 | 73.3 | 39.8(7) |
| BeatGAN | 35.8 | 34.8 | 21.3 | 21.4 | 39.5 | 29.0 | 47.0 | 37.8 | 33.2 | 55.5 | 6.2 | 16.4 | 39.1 | 23.6 | 52.2 | 32.8(18) |
| MAD-GAN | 30.8 | 26.6 | 20.5 | 20.8 | 39.9 | 16.0 | 27.1 | 32.9 | 43.2 | 48.2 | 6.5 | 4.4 | 26.9 | 22.9 | 31.2 | 26.5(23) |
| LSTM-VAE-GAN | 38.7 | 37.3 | 18.1 | 30.5 | 46.2 | 16.4 | 51.0 | 33.4 | 31.2 | 49.0 | 6.9 | 13.6 | 25.4 | 29.1 | 55.9 | 32.2(20) |
| TadGAN | 34.7 | 16.3 | 9.1 | 22.1 | 44.3 | 26.8 | 47.5 | 40.4 | 31.2 | 52.0 | 8.0 | 13.0 | 51.4 | 24.9 | 48.6 | 31.4(21) |
| LSTM-AE OC-SVM | 8.4 | 19.7 | 9.9 | 9.2 | 12.8 | 23.7 | 13.3 | 28.2 | 17.5 | 45.8 | 7.5 | 4.8 | 32.1 | 26.7 | 48.3 | 20.5(26) |
| MTAD-GAT | 28.1 | 63.4 | 9.2 | 22.5 | 49.7 | 49.3 | 40.2 | 38.5 | 35.8 | 33.7 | 6.4 | 24.6 | 49.0 | 22.6 | 51.0 | 34.9(12) |
| NCAD | 5.2 | 14.4 | 6.4 | 8.4 | 3.4 | 4.3 | 1.4 | 1.8 | 4.2 | 3.1 | 4.9 | 7.8 | 2.3 | 5.3 | 5.5 | 5.2 (28) |
| THOC | 42.6 | 24.8 | 16.5 | 13.7 | 14.5 | 14.3 | 19.3 | 32.8 | 17.6 | 10.8 | 4.9 | 10.8 | 20.5 | 15.0 | 52.6 | 20.7(25) |

Table 9: Cross-validation results on SMD evaluated with the $F_1$ metric introduced by (Tatbul et al., 2018).

| | \multicolumn{15}{c}{Server ID} | |
| | 1 | 6 | 8 | 9 | 10 | 11 | 13 | 14 | 16 | 17 | 20 | 21 | 24 | 26 | 27 | avg(rank) |
|---|---|---|---|---|---|---|---|---|---|---|---|---|---|---|---|---|
| LSTM-AE | 17.8 | 47.5 | 28.8 | 69.6 | 54.3 | 39.8 | 52.0 | 43.2 | 60.1 | 50.7 | 23.8 | 42.8 | 48.3 | 61.5 | 51.1 | 46.1(5) |
| LSTM-Max-AE | 26.7 | 50.3 | 22.5 | 72.8 | 35.7 | 15.0 | 54.4 | 25.7 | 62.9 | 34.6 | 14.4 | 15.6 | 45.5 | 59.3 | 35.3 | 38.0(21) |
| MSCRED | 37.9 | 37.4 | 36.9 | 58.6 | 41.5 | 32.9 | 50.3 | 9.7 | 61.7 | 71.2 | 25.1 | 8.1 | 52.4 | 64.4 | 64.9 | 43.5(8) |
| FC-AE | 23.1 | 36.1 | 24.6 | 66.3 | 31.1 | 26.5 | 53.3 | 42.9 | 58.2 | 45.6 | 26.5 | 33.1 | 46.7 | 53.7 | 42.8 | 40.7(16) |
| USAD | 25.8 | 43.2 | 29.3 | 76.3 | 44.3 | 20.9 | 53.1 | 42.6 | 57.9 | 36.8 | 19.5 | 29.3 | 53.4 | 59.2 | 42.2 | 42.3(12) |
| TCN-AE | 46.5 | 48.3 | 47.7 | 60.2 | 41.3 | 46.8 | 44.3 | 18.1 | 67.5 | 63.1 | 39.6 | 29.7 | 53.0 | 68.2 | 57.1 | 48.8(**2**) |
| GenAD | 23.4 | 17.0 | 22.6 | 53.7 | 29.0 | 5.9 | 47.1 | 33.1 | 51.6 | 25.4 | 4.7 | 2.8 | 43.6 | 56.9 | 38.1 | 30.3(27) |
| STGAT-MAD | 19.8 | 44.4 | 27.7 | 67.9 | 47.4 | 41.8 | 52.6 | 40.0 | 58.1 | 47.9 | 27.0 | 32.4 | 48.7 | 59.5 | 50.7 | 44.4(7) |
| AnomalyTransformer | 28.5 | 41.2 | 49.2 | 42.6 | 36.6 | 43.8 | 34.9 | 56.0 | 67.7 | 15.8 | 16.2 | 42.8 | 51.3 | 70.4 | 18.1 | 41.0(14) |
| LSTM-P | 34.0 | 55.1 | 30.6 | 68.7 | 56.7 | 44.0 | 53.2 | 40.4 | 55.2 | 52.4 | 22.1 | 35.5 | 51.9 | 55.1 | 49.4 | 47.0(4) |
| LSTM-S2S-P | 58.1 | 50.2 | 46.0 | 64.0 | 42.5 | 40.0 | 50.9 | 29.0 | 66.4 | 67.7 | 45.7 | 28.2 | 62.4 | 59.3 | 60.2 | 51.4(**1**) |
| DeepAnt | 23.3 | 39.3 | 27.0 | 66.8 | 28.1 | 23.8 | 52.3 | 42.2 | 58.0 | 52.6 | 20.7 | 30.0 | 44.1 | 51.1 | 38.9 | 39.9(18) |
| TCN-S2S-P | 16.2 | 51.3 | 29.5 | 70.0 | 47.1 | 46.2 | 51.2 | 31.3 | 54.8 | 61.1 | 29.6 | 34.9 | 48.7 | 49.3 | 58.1 | 45.3(6) |
| GDN | 24.7 | 40.1 | 25.5 | 62.2 | 35.7 | 27.9 | 52.4 | 39.9 | 63.3 | 41.4 | 22.7 | 30.7 | 44.6 | 65.2 | 38.2 | 41.0(15) |
| LSTM-VAE | 18.6 | 33.6 | 34.5 | 60.5 | 37.8 | 39.6 | 48.7 | 37.5 | 54.5 | 32.7 | 23.4 | 35.6 | 38.1 | 45.2 | 39.0 | 38.6(20) |
| Donut | 18.4 | 33.7 | 25.7 | 62.4 | 44.0 | 37.0 | 43.0 | 42.0 | 47.3 | 60.3 | 27.1 | 35.7 | 53.1 | 46.7 | 61.0 | 42.5(11) |
| LSTM-DVAE | 17.2 | 34.6 | 28.0 | 58.8 | 26.4 | 39.3 | 45.2 | 36.7 | 45.4 | 34.2 | 21.7 | 33.2 | 40.4 | 49.0 | 29.6 | 36.0(24) |
| GMM-GRU-VAE | 21.0 | 35.5 | 26.9 | 69.9 | 34.8 | 47.6 | 49.1 | 41.2 | 46.9 | 64.2 | 25.0 | 29.5 | 44.6 | 51.6 | 52.1 | 42.7(10) |
| OmniAnomaly | 17.6 | 30.7 | 31.4 | 67.1 | 15.6 | 41.8 | 45.6 | 31.9 | 49.4 | 45.9 | 16.7 | 20.3 | 33.2 | 41.4 | 52.0 | 36.0(23) |
| SIS-VAE | 24.6 | 37.6 | 26.2 | 63.8 | 30.4 | 29.1 | 52.7 | 44.0 | 59.3 | 40.4 | 22.1 | 31.2 | 47.0 | 54.0 | 42.1 | 40.3(17) |
| BeatGAN | 26.9 | 39.9 | 33.8 | 78.6 | 33.4 | 16.9 | 52.6 | 32.9 | 64.6 | 34.6 | 23.1 | 26.3 | 43.1 | 49.0 | 29.4 | 39.0(19) |
| MAD-GAN | 28.5 | 47.3 | 45.9 | 74.4 | 44.9 | 66.1 | 50.4 | 37.3 | 62.9 | 60.5 | 33.4 | 25.0 | 52.8 | 60.7 | 39.9 | 48.7(**3**) |
| LSTM-VAE-GAN | 20.7 | 35.4 | 23.9 | 63.9 | 33.6 | 22.2 | 52.4 | 16.0 | 52.8 | 29.1 | 20.3 | 38.1 | 33.6 | 53.6 | 33.3 | 35.3(25) |
| TadGAN | 33.6 | 35.1 | 41.7 | 72.3 | 30.0 | 27.1 | 55.7 | 32.6 | 63.5 | 33.5 | 18.9 | 37.7 | 52.7 | 59.0 | 46.9 | 42.7(9) |
| LSTM-AE OC-SVM | 9.9 | 27.0 | 21.3 | 51.9 | 11.4 | 18.5 | 21.3 | 9.9 | 62.5 | 53.7 | 26.3 | 32.1 | 32.4 | 56.1 | 58.4 | 32.8(26) |
| MTAD-GAT | 19.2 | 44.8 | 26.1 | 56.9 | 40.6 | 42.5 | 21.1 | 48.8 | 64.4 | 17.0 | 17.1 | 38.3 | 27.1 | 45.9 | 35.1 | 36.3(22) |
| NCAD | 10.2 | 35.3 | 45.0 | 56.8 | 22.7 | 54.0 | 55.4 | 35.1 | 70.4 | 23.4 | 18.8 | 27.8 | 55.7 | 70.6 | 44.1 | 41.7(13) |
| THOC | 25.7 | 24.7 | 28.9 | 49.6 | 12.4 | 30.1 | 18.1 | 11.7 | 50.0 | 21.5 | 22.8 | 18.8 | 26.7 | 48.5 | 36.4 | 28.4(28) |

Table 10: Cross-validation results on SMD evaluated with the *AUPRC* metric introduced by (Tatbul et al., 2018).

| | 1 | 6 | 8 | 9 | 10 | 11 | 13 | 14 | 16 | 17 | 20 | 21 | 24 | 26 | 27 | avg(rank) |
|---|---|---|---|---|---|---|---|---|---|---|---|---|---|---|---|---|
| | | | | | | | | | Server ID | | | | | | | |
| LSTM-AE | 18.5 | 53.3 | 15.8 | 33.1 | 50.5 | 29.9 | 44.1 | 38.1 | 40.3 | 70.9 | 17.0 | 32.2 | 57.6 | 25.4 | 74.9 | 40.1(**1**) |
| LSTM-Max-AE | 28.4 | 24.6 | 12.8 | 30.9 | 23.5 | 11.4 | 45.7 | 13.3 | 37.5 | 53.4 | 5.9 | 7.4 | 53.0 | 28.7 | 55.8 | 28.8(19) |
| MSCRED | 38.0 | 28.3 | 25.4 | 13.4 | 23.8 | 15.8 | 45.2 | 20.9 | 33.8 | 69.7 | 8.6 | 4.9 | 52.6 | 25.2 | 62.0 | 31.2(13) |
| FC-AE | 25.3 | 35.8 | 14.7 | 30.2 | 29.1 | 18.2 | 44.2 | 31.3 | 39.8 | 60.5 | 10.7 | 21.1 | 53.8 | 26.3 | 71.7 | 34.2(5) |
| USAD | 21.2 | 39.2 | 12.5 | 31.1 | 25.6 | 19.3 | 43.3 | 31.3 | 30.6 | 54.3 | 8.2 | 17.6 | 49.4 | 24.7 | 58.3 | 31.1(14) |
| TCN-AE | 28.4 | 31.4 | 19.5 | 10.1 | 17.5 | 16.0 | 23.0 | 7.9 | 33.1 | 57.0 | 13.0 | 9.7 | 35.2 | 28.4 | 40.6 | 24.7(24) |
| GenAD | 27.4 | 25.3 | 25.1 | 22.4 | 39.8 | 16.2 | 43.4 | 41.5 | 31.5 | 56.5 | 10.0 | 13.2 | 50.9 | 26.3 | 49.2 | 31.9(12) |
| STGAT-MAD | 22.4 | 44.2 | 15.7 | 31.4 | 40.2 | 24.2 | 45.1 | 33.0 | 41.0 | 63.4 | 15.8 | 22.2 | 57.6 | 29.4 | 73.1 | 37.2(4) |
| AnomalyTransformer | 21.4 | 32.1 | 14.2 | 15.1 | 26.0 | 12.7 | 21.7 | 20.9 | 35.7 | 30.1 | 7.0 | 18.1 | 52.3 | 24.5 | 14.7 | 23.1(25) |
| LSTM-P | 35.4 | 59.4 | 17.9 | 33.2 | 46.6 | 32.3 | 46.8 | 29.0 | 34.3 | 68.6 | 13.5 | 24.0 | 58.1 | 21.4 | 75.5 | 39.7(**2**) |
| LSTM-S2S-P | 44.4 | 38.8 | 28.1 | 19.8 | 31.1 | 13.6 | 42.3 | 12.4 | 35.5 | 59.2 | 15.0 | 14.5 | 58.9 | 25.6 | 54.4 | 32.9(10) |
| DeepAnt | 24.6 | 41.3 | 14.6 | 30.8 | 27.8 | 16.6 | 43.8 | 29.9 | 39.6 | 56.3 | 11.0 | 14.3 | 54.8 | 24.4 | 55.0 | 32.3(11) |
| TCN-S2S-P | 18.2 | 55.3 | 16.7 | 32.8 | 41.2 | 33.8 | 45.1 | 21.2 | 35.6 | 72.9 | 18.4 | 21.8 | 54.2 | 18.5 | 75.5 | 37.4(**3**) |
| GDN | 29.8 | 39.3 | 15.8 | 28.7 | 33.1 | 16.1 | 43.5 | 28.5 | 40.5 | 54.6 | 10.0 | 18.3 | 52.4 | 30.7 | 55.5 | 33.1(8) |
| LSTM-VAE | 14.3 | 30.0 | 12.0 | 26.5 | 32.8 | 32.7 | 44.1 | 21.9 | 30.2 | 49.0 | 13.4 | 24.5 | 46.3 | 14.1 | 60.2 | 30.1(15) |
| Donut | 13.3 | 31.7 | 11.5 | 28.1 | 40.2 | 31.7 | 37.0 | 30.8 | 27.5 | 66.3 | 18.3 | 24.1 | 59.6 | 11.5 | 69.4 | 33.4(6) |
| LSTM-DVAE | 11.4 | 33.3 | 10.9 | 22.7 | 22.5 | 37.2 | 40.3 | 22.5 | 23.1 | 39.1 | 13.8 | 19.8 | 45.3 | 16.6 | 43.2 | 26.8(22) |
| GMM-GRU-VAE | 14.6 | 34.2 | 13.1 | 31.9 | 28.5 | 42.5 | 41.9 | 28.8 | 23.7 | 70.4 | 14.8 | 13.7 | 50.9 | 15.1 | 75.4 | 33.3(7) |
| OmniAnomaly | 10.9 | 32.0 | 11.5 | 33.6 | 9.8 | 35.1 | 40.5 | 13.8 | 31.3 | 59.6 | 10.7 | 9.2 | 46.0 | 10.1 | 68.0 | 28.1(21) |
| SIS-VAE | 25.5 | 40.7 | 14.5 | 30.4 | 28.0 | 18.4 | 44.1 | 32.2 | 39.5 | 53.7 | 10.6 | 17.7 | 49.3 | 25.6 | 65.7 | 33.0(9) |
| BeatGAN | 24.7 | 34.7 | 14.3 | 28.8 | 30.1 | 13.7 | 45.1 | 16.0 | 35.5 | 53.5 | 10.0 | 13.4 | 44.7 | 22.7 | 51.6 | 29.2(17) |
| MAD-GAN | 29.1 | 33.1 | 18.9 | 31.1 | 28.7 | 11.8 | 43.2 | 22.0 | 32.5 | 53.9 | 13.0 | 9.5 | 42.4 | 24.2 | 41.0 | 29.0(18) |
| LSTM-VAE-GAN | 23.0 | 35.6 | 11.7 | 31.7 | 27.7 | 9.5 | 45.3 | 7.1 | 32.3 | 43.9 | 7.9 | 12.3 | 30.4 | 27.2 | 53.4 | 26.6(23) |
| TadGAN | 32.7 | 19.8 | 13.1 | 32.3 | 29.4 | 12.4 | 44.9 | 17.6 | 41.1 | 51.7 | 9.6 | 11.7 | 53.2 | 29.6 | 50.1 | 29.9(16) |
| LSTM-AE OC-SVM | 5.6 | 20.6 | 6.8 | 7.7 | 6.6 | 10.7 | 13.9 | 4.5 | 25.9 | 46.9 | 7.7 | 7.3 | 29.6 | 26.5 | 52.9 | 18.2(26) |
| MTAD-GAT | 16.4 | 43.8 | 7.8 | 21.5 | 33.3 | 46.5 | 32.3 | 27.9 | 29.4 | 30.5 | 10.0 | 27.5 | 38.3 | 15.3 | 47.3 | 28.5(20) |
| NCAD | 3.7 | 14.8 | 8.4 | 11.0 | 3.7 | 9.9 | 16.9 | 2.0 | 22.5 | 24.8 | 10.1 | 6.1 | 23.7 | 8.7 | 33.4 | 13.3(28) |
| THOC | 24.4 | 17.2 | 8.2 | 11.2 | 6.0 | 10.2 | 17.9 | 5.7 | 13.7 | 15.7 | 6.2 | 8.3 | 19.4 | 15.6 | 50.7 | 15.3(27) |

Table 11: Cross-validation results on SMD evaluated with our adapted best $F_1$ score metric, using $TRec^*$ and $TPrec^*$.

| | | | | | | | Server ID | | | | | | | | | |
|---|---|---|---|---|---|---|---|---|---|---|---|---|---|---|---|---|
| | 1 | 6 | 8 | 9 | 10 | 11 | 13 | 14 | 16 | 17 | 20 | 21 | 24 | 26 | 27 | avg(rank) |
| LSTM-AE | 53.1 | 70.2 | 42.4 | 65.1 | 68.6 | 39.8 | 55.0 | 62.3 | 79.7 | 80.7 | 25.0 | 48.7 | 64.0 | 62.4 | 84.6 | 60.1**(1)** |
| LSTM-Max-AE | 55.8 | 28.6 | 30.3 | 64.8 | 34.9 | 24.6 | 54.3 | 26.1 | 63.1 | 68.8 | 16.1 | 10.6 | 58.8 | 61.4 | 72.4 | 44.7(17) |
| MSCRED | 51.3 | 46.3 | 47.0 | 58.5 | 36.1 | 14.0 | 47.0 | 10.1 | 70.3 | 73.0 | 14.3 | 14.4 | 52.1 | 54.3 | 64.3 | 43.5(20) |
| FC-AE | 52.2 | 61.1 | 40.3 | 65.1 | 51.0 | 29.8 | 53.0 | 48.2 | 75.7 | 77.2 | 18.0 | 36.4 | 60.3 | 61.2 | 85.4 | 54.3(7) |
| USAD | 43.0 | 45.0 | 31.7 | 63.9 | 36.0 | 33.2 | 52.9 | 45.2 | 59.4 | 68.1 | 18.7 | 26.3 | 52.4 | 53.7 | 68.6 | 46.5(15) |
| TCN-AE | 43.7 | 49.4 | 42.1 | 56.1 | 34.4 | 17.1 | 44.3 | 17.2 | 67.1 | 63.5 | 23.1 | 19.1 | 51.4 | 66.7 | 56.1 | 43.4(21) |
| GenAD | 44.8 | 17.7 | 26.6 | 50.6 | 28.2 | 12.3 | 53.8 | 25.7 | 60.2 | 61.7 | 5.3 | 2.9 | 53.6 | 58.3 | 68.9 | 38.0(24) |
| STGAT-MAD | 48.3 | 66.5 | 36.4 | 66.0 | 62.6 | 34.1 | 52.5 | 52.5 | 77.9 | 77.0 | 24.0 | 34.7 | 60.6 | 65.0 | 83.8 | 56.1(5) |
| AnomalyTransformer | 22.3 | 45.3 | 33.0 | 44.5 | 27.3 | 21.0 | 10.9 | 32.4 | 60.7 | 27.5 | 13.2 | 28.4 | 53.4 | 56.3 | 11.6 | 32.5(27) |
| LSTM-P | 61.7 | 73.4 | 46.7 | 64.2 | 61.5 | 37.6 | 56.7 | 59.2 | 77.0 | 81.4 | 18.8 | 35.5 | 60.0 | 60.0 | 83.9 | 58.5**(2)** |
| LSTM-S2S-P | 52.4 | 48.5 | 46.5 | 60.1 | 29.0 | 20.1 | 40.3 | 14.4 | 71.5 | 67.5 | 17.1 | 23.2 | 56.1 | 58.3 | 61.7 | 44.5(18) |
| DeepAnt | 50.6 | 62.0 | 37.7 | 65.9 | 40.8 | 28.4 | 52.0 | 47.0 | 75.5 | 68.1 | 19.2 | 29.1 | 61.3 | 60.2 | 63.8 | 50.8(12) |
| TCN-S2S-P | 47.2 | 73.3 | 44.6 | 65.5 | 57.7 | 42.6 | 51.6 | 42.3 | 78.8 | 84.5 | 27.2 | 37.2 | 61.6 | 60.0 | 85.5 | 57.3**(3)** |
| GDN | 57.8 | 58.3 | 40.9 | 64.9 | 58.6 | 29.2 | 52.7 | 41.5 | 67.9 | 72.5 | 18.0 | 33.8 | 61.6 | 55.4 | 74.5 | 52.5(10) |
| LSTM-VAE | 47.6 | 63.7 | 35.2 | 58.4 | 64.6 | 48.8 | 53.9 | 42.3 | 73.8 | 58.8 | 20.5 | 46.3 | 60.6 | 52.7 | 68.8 | 53.1(9) |
| Donut | 46.5 | 66.7 | 38.9 | 67.1 | 62.7 | 47.0 | 50.2 | 51.6 | 65.2 | 77.5 | 29.6 | 34.8 | 68.7 | 48.4 | 76.2 | 55.4(6) |
| LSTM-DVAE | 42.2 | 63.8 | 35.4 | 56.9 | 58.4 | 49.6 | 50.8 | 54.8 | 70.5 | 48.8 | 20.1 | 41.2 | 60.1 | 55.2 | 52.5 | 50.7(13) |
| GMM-GRU-VAE | 49.1 | 67.0 | 38.1 | 67.1 | 62.0 | 60.6 | 51.2 | 49.4 | 68.7 | 79.2 | 21.4 | 35.0 | 63.3 | 58.2 | 83.4 | 56.9(4) |
| OmniAnomaly | 45.3 | 67.7 | 35.3 | 65.3 | 42.9 | 57.7 | 62.0 | 40.7 | 77.0 | 57.1 | 16.9 | 25.7 | 56.8 | 44.6 | 82.4 | 51.8(11) |
| SIS-VAE | 52.6 | 65.7 | 38.4 | 65.4 | 50.8 | 30.9 | 52.9 | 49.0 | 74.9 | 73.0 | 21.2 | 32.8 | 58.1 | 63.8 | 79.1 | 53.9(8) |
| BeatGAN | 45.3 | 46.5 | 34.0 | 64.1 | 47.1 | 25.7 | 52.7 | 31.6 | 61.6 | 68.7 | 17.6 | 24.7 | 49.0 | 57.8 | 65.5 | 46.1(16) |
| MAD-GAN | 43.4 | 35.9 | 37.6 | 64.1 | 40.3 | 19.1 | 46.8 | 32.6 | 65.7 | 48.6 | 18.5 | 9.4 | 34.3 | 57.3 | 32.9 | 39.1(23) |
| LSTM-VAE-GAN | 47.9 | 46.9 | 31.0 | 67.9 | 44.4 | 23.1 | 56.3 | 17.8 | 57.1 | 56.6 | 16.1 | 19.5 | 37.9 | 61.6 | 69.8 | 43.6(19) |
| TadGAN | 49.0 | 27.3 | 22.9 | 64.1 | 45.4 | 22.0 | 54.4 | 23.9 | 64.0 | 63.3 | 16.6 | 18.7 | 54.5 | 62.2 | 49.5 | 42.5(22) |
| LSTM-AE OC-SVM | 19.0 | 28.2 | 24.6 | 49.3 | 13.3 | 23.4 | 23.5 | 8.4 | 50.7 | 61.6 | 14.6 | 8.8 | 40.5 | 60.4 | 67.3 | 32.9(26) |
| MTAD-GAT | 43.0 | 65.9 | 24.1 | 64.5 | 52.8 | 57.1 | 50.1 | 43.2 | 71.0 | 41.7 | 16.3 | 39.6 | 55.7 | 57.9 | 56.6 | 49.3(14) |
| NCAD | 13.9 | 25.7 | 21.3 | 48.5 | 10.9 | 12.5 | 4.8 | 3.7 | 38.6 | 6.8 | 12.5 | 14.3 | 8.7 | 40.0 | 13.4 | 18.4(28) |
| THOC | 51.4 | 38.2 | 27.7 | 54.0 | 23.7 | 22.9 | 24.7 | 16.7 | 51.2 | 25.7 | 12.2 | 17.1 | 35.0 | 49.8 | 64.2 | 34.3(25) |

Table 12: Cross-validation results on SMD evaluated with our adapted AUPRC metric, using $TRec^*$ and $TPrec^*$.

| | | | | | | Server ID | | | | | | | | | | |
|---|---|---|---|---|---|---|---|---|---|---|---|---|---|---|---|---|
| | 1 | 6 | 8 | 9 | 10 | 11 | 13 | 14 | 16 | 17 | 20 | 21 | 24 | 26 | 27 | avg(rank) |
| LSTM-AE | 39.5 | 70.9 | 25.6 | 29.2 | 65.5 | 33.8 | 46.1 | 59.2 | 48.8 | 81.1 | 17.8 | 37.1 | 64.9 | 31.0 | 81.8 | 48.8(**1**) |
| LSTM-Max-AE | 43.0 | 18.5 | 18.5 | 28.7 | 24.7 | 19.4 | 45.9 | 14.5 | 34.7 | 64.5 | 6.4 | 6.1 | 53.0 | 30.9 | 67.8 | 31.8(20) |
| MSCRED | 45.4 | 34.9 | 36.1 | 13.5 | 20.1 | 10.3 | 40.4 | 22.5 | 37.7 | 68.7 | 5.5 | 10.1 | 49.9 | 21.0 | 61.1 | 31.8(19) |
| FC-AE | 46.3 | 54.1 | 24.3 | 30.0 | 43.2 | 23.5 | 43.8 | 37.6 | 43.5 | 73.0 | 7.9 | 24.5 | 57.3 | 30.6 | 82.4 | 41.4(8) |
| USAD | 35.8 | 41.9 | 16.3 | 26.6 | 28.9 | 29.9 | 43.2 | 33.4 | 25.5 | 63.5 | 9.2 | 15.6 | 46.2 | 23.3 | 61.3 | 33.4(16) |
| TCN-AE | 25.8 | 30.5 | 19.0 | 8.5 | 12.0 | 12.5 | 20.0 | 9.2 | 37.1 | 56.3 | 9.4 | 10.3 | 34.0 | 31.5 | 36.1 | 23.5(24) |
| GenAD | 41.8 | 28.3 | 27.6 | 21.5 | 39.5 | 21.3 | 48.7 | 40.1 | 33.7 | 65.6 | 10.1 | 12.8 | 52.5 | 28.0 | 63.1 | 35.6(14) |
| STGAT-MAD | 42.6 | 61.1 | 23.7 | 30.1 | 56.6 | 27.2 | 45.3 | 48.6 | 47.6 | 77.2 | 15.7 | 24.9 | 62.9 | 33.5 | 80.4 | 45.2(4) |
| AnomalyTransformer | 26.0 | 37.8 | 16.3 | 15.8 | 26.4 | 14.2 | 15.2 | 16.5 | 23.3 | 28.2 | 7.2 | 17.7 | 47.7 | 24.2 | 6.7 | 21.5(25) |
| LSTM-P | 57.8 | 71.4 | 30.2 | 29.0 | 55.6 | 34.3 | 48.6 | 49.9 | 46.8 | 78.8 | 12.1 | 27.6 | 63.0 | 27.9 | 82.0 | 47.7(**2**) |
| LSTM-S2S-P | 45.5 | 37.3 | 33.5 | 13.9 | 22.4 | 12.4 | 36.6 | 10.1 | 38.5 | 59.6 | 8.1 | 15.3 | 50.1 | 27.7 | 56.9 | 31.2(21) |
| DeepAnt | 45.4 | 58.3 | 21.9 | 30.4 | 34.1 | 22.7 | 45.0 | 36.5 | 43.1 | 64.4 | 10.8 | 15.8 | 62.0 | 29.5 | 63.3 | 38.9(10) |
| TCN-S2S-P | 37.6 | 71.6 | 28.0 | 29.5 | 51.5 | 35.8 | 45.5 | 33.6 | 48.1 | 84.8 | 19.1 | 24.6 | 63.9 | 26.9 | 82.9 | 45.6(**3**) |
| GDN | 52.8 | 53.7 | 27.1 | 29.1 | 52.2 | 25.0 | 45.2 | 30.8 | 37.3 | 70.7 | 8.6 | 21.8 | 63.8 | 25.1 | 73.3 | 41.1(9) |
| LSTM-VAE | 32.5 | 53.8 | 18.3 | 25.5 | 56.1 | 44.0 | 47.0 | 27.5 | 40.2 | 53.6 | 12.7 | 29.2 | 57.6 | 18.5 | 62.6 | 38.6(11) |
| Donut | 33.0 | 59.9 | 20.0 | 29.8 | 58.3 | 40.0 | 40.6 | 41.3 | 36.0 | 75.0 | 19.7 | 25.0 | 69.4 | 15.0 | 73.8 | 42.5(6) |
| LSTM-DVAE | 26.8 | 55.6 | 16.9 | 20.5 | 46.3 | 46.4 | 43.7 | 37.4 | 38.0 | 40.7 | 12.0 | 24.3 | 56.9 | 21.7 | 46.4 | 35.6(15) |
| GMM-GRU-VAE | 37.4 | 60.6 | 20.8 | 31.6 | 51.1 | 56.3 | 43.9 | 39.8 | 36.0 | 75.7 | 12.5 | 18.0 | 64.6 | 27.4 | 81.1 | 43.8(5) |
| OmniAnomaly | 28.9 | 57.6 | 18.5 | 33.1 | 25.0 | 49.1 | 52.4 | 23.2 | 47.8 | 49.1 | 11.4 | 12.7 | 57.7 | 11.3 | 82.2 | 37.3(12) |
| SIS-VAE | 47.0 | 63.9 | 22.4 | 30.0 | 42.8 | 23.1 | 45.8 | 36.2 | 43.3 | 68.9 | 11.0 | 20.2 | 57.3 | 32.4 | 78.2 | 41.5(7) |
| BeatGAN | 39.8 | 39.8 | 21.3 | 26.1 | 38.6 | 20.1 | 43.6 | 20.2 | 29.4 | 64.2 | 8.4 | 15.7 | 42.7 | 27.4 | 57.6 | 33.0(17) |
| MAD-GAN | 33.3 | 26.6 | 18.7 | 27.1 | 33.4 | 14.8 | 24.8 | 19.7 | 33.8 | 48.3 | 7.7 | 4.4 | 25.6 | 25.5 | 31.4 | 25.0(23) |
| LSTM-VAE-GAN | 40.7 | 44.2 | 18.3 | 34.7 | 35.3 | 14.3 | 49.5 | 12.2 | 29.1 | 53.1 | 9.1 | 11.3 | 29.7 | 33.2 | 65.0 | 32.0(18) |
| TadGAN | 34.9 | 17.6 | 8.3 | 28.2 | 36.9 | 17.8 | 44.0 | 17.8 | 31.1 | 56.8 | 9.9 | 9.4 | 51.6 | 30.3 | 48.6 | 29.6(22) |
| LSTM-AE OC-SVM | 10.2 | 22.8 | 9.5 | 7.3 | 8.8 | 16.9 | 14.0 | 6.7 | 17.8 | 53.0 | 8.0 | 3.9 | 31.9 | 28.7 | 56.9 | 19.8(27) |
| MTAD-GAT | 31.1 | 65.6 | 10.4 | 27.3 | 46.8 | 54.1 | 42.5 | 32.2 | 37.0 | 34.8 | 10.6 | 30.1 | 53.9 | 23.0 | 53.3 | 36.8(13) |
| NCAD | 6.4 | 17.0 | 6.3 | 8.2 | 3.5 | 4.3 | 1.8 | 1.2 | 3.8 | 3.2 | 9.2 | 7.4 | 2.7 | 5.9 | 5.8 | 5.8 (28) |
| THOC | 44.7 | 26.7 | 14.7 | 13.3 | 10.6 | 13.2 | 18.0 | 12.2 | 17.4 | 17.7 | 5.2 | 8.2 | 22.4 | 15.9 | 62.2 | 20.2(26) |

