# OpenReview forum: "TimeSeAD: Benchmarking Deep Multivariate Time-Series Anomaly Detection"
_TMLR — Accepted by TMLR_

### Review · Reviewer_t3jS · 2023-01-19

**Summary Of Contributions:**

This paper discusses some issues they found in common time series anomaly detection datasets and metrics.  They propose a new metric that is recall consistent.  They then provide an extensive benchmark of existing methods.

**Audience:**

Yes

**Broader Impact Concerns:**

No ethical concerns.

**Claims And Evidence:**

Yes

**Requested Changes:**

I originally missed footnote 2 on page 4.  I think its important to tell readers that there are descriptions of what the datasets are and what they collected in Appendix B.

Please soften the tone a bit or add an explicit recognition that the choices being made to filter the datasets are in the service of improving the interpretability and repeatability of the benchmark and they aren't necessarily meant to be taken as a universal declaration of what does or doesn't constitute proper anomaly detection.

I would appreciate some more intuition regarding why Recall consistency is a good thing, why the existing metric fails and how the new one avoids the failure.  The math is all there, but I'd appreciate some more words on the matter.

**Strengths And Weaknesses:**

## Strengths:

Overall, the paper appears to be correct in the sense that the claims it is making are correct and supported by evidence.

The paper contains a lot of useful detailed discussion of existing techniques, datasets and metrics and their potential shortcomings.

The paper argues convincingly that recall consistency is a useful property, the existing popular metric fails to be recall consistent and propose a new metric that is, proving that it is.

The paper includes an extensive bench-marking of existing techniques.

## Weaknesses

The biggest weakness of the paper is probably its narrow audience.  The paper is a very in-depth discussion of a particular task and set of papers.  I'm a bit concerned that it lacks much broader interest outside of that small community.  To this point, the way the paper is written, its assuming its audience already has a great deal of knowledge about the particular datasets  / metrics  and techniques being discussed, so it isn't very *inviting* for someone not familiar with these things. If I'm being honest, if I weren't assigned the review, I wouldn't care to read it.  I found it difficult to read as an outsider. It's very "inside-baseball", esoteric and highly technical for a specific sub-field.

The other thing that strikes me is that I feel as though the paper is pushing a broader philosophical point that I disagree with.  The paper points out many features of the existing datasets that it takes issue with, things like higher anomaly density, positional bias, long anomalies and constant features.  I think its right to identify that these prevent particularly challenging issues for point-wise anomaly detection, but I fear that paper and the tone it takes is as if to suggest that these real world datasets in some way aren't reflective of the real-world, while I personally feel very much the opposite.  Things like positional bias might make it difficult for many of the simple point wise anomaly detection techniques and metrics to do their job well, but it is also very much a very common phenomenon in the real world.  I mean, these datasets appear to all have been collected from real sensors living in the real world.  There is a very real sense in which all of these intricacies are in fact a *better* representation of the difficulties with anomaly detection in the real world, not that they are lesser for it.  I agree that *cleaner* datasets can make for *cleaner* comparisons, but I disagree as soon as I feel as though the paper is implicitly assuming that *cleaner* is *better*.

The paper is very strongly opinionated about what does or doesn't constitute a *proper* anomaly detection dataset, including things like < 10% anomalies.  Again, I don't really take particular issues with the choices being made here, just with the suggestion that these choices are somehow the only with any merit.  I think there should, early on in the paper be an explicit recognition that the paper is going to proceed to be highly-opinionated in the service of trying to come up with a repeatable and interpretable benchmark, while acknowledging that many might see this as a further departure from a *real-world* evaluation.  Otherwise there is an air of opinion being presented as fact.

---

> ### Author Response · Authors · 2023-02-09
> **Reply to intial review of Reviwer t3jS**
>
> Thank you for your helpful comments and constructive review! We will address your individual points below:
>
> > The biggest weakness of the paper is probably its narrow audience.
>
> We agree that our work is highly technical and probably interesting for only a niche audience. Nevertheless, we believe that our work provides novel insights and a relevant contribution for researchers working in (deep) time-series anomaly detection (TSAD) who want to benchmark a new algorithm that they came up with, for example. As we mention in Section 2, quite a few TSAD methods have been proposed, especially in the last years (see e.g., the surveys [1,2]), so we would consider it a highly active research branch. Hence, we would argue that our work meets TMLR's criteria regarding the audience and generally disagree that a "small community" working in a specific field is a major weakness.
>
> > To this point, the way the paper is written, its assuming its audience already has a great deal of knowledge about the particular datasets / metrics and techniques being discussed, so it isn't very inviting for someone not familiar with these things.
>
> We wrote the paper with a target audience in mind that is at least somewhat familiar with time-series anomaly detection. Since the scope of our work is quite broad (we consider datasets, methods, evaluation prcedures, and metrics), we believe that explaining all concepts from scratch would bloat the paper too much. However, we agree that there is at least some potential to make the paper more "inviting" and that it would strengthen its overall contribution if it were accessible to a broader audience. Hence, we updated our writing throughout the paper in the revised version with these points in mind.
>
> > The paper is very strongly opinionated about what does or doesn't constitute a proper anomaly detection dataset.
> > I fear that paper and the tone it takes is as if to suggest that these real world datasets in some way aren't reflective of the real-world
>
> We are not claiming that some of the datasets we mention are inherently bad/mislabelled or not reflective of the real world. The criteria we apply to select datasets are entirely in the interest of ensuring an interpretable, *meaningful* benchmark in the sense that they should be suitable for the anomaly detection task as defined in, e.g., [3]. Consider, for example, the WADI dataset: as we describe in Appendix B, its labeled "anomalies" are cyber attacks. Those labels are, therefore, correct in the sense that the dataset collectors simply recorded the time when they ran an attack, but as we show in Figure 3 (b), it contains an apparent distributional shift that would cause most detectors to consider the entire second half of the test set as anomalous. Thus, the WADI dataset may very well be reflective of the real world, and it might be an excellent dataset for detecting cyber attacks. Still, we would not recommend using it in the scope of a benchmark as a basis to make claims like "TSAD method [A] works significantly better than method [B]". We overhauled the introduction of Section 3.1 and made changes throughout to highlight where we make suggestions for an ideal benchmark and where we analyze real-world datasets. In particular, we removed any suggestions of specific statistics based on our intuition.
>
> >Things like positional bias might make it difficult for many of the simple point wise anomaly detection techniques and metrics to do their job well, but it is also very much a very common phenomenon in the real world.
>
> Positional bias is a common phenomenon in real world applications. For a fair and unbiased evaluation, however, extensive positional bias will inherently prefer methods accounting for that bias. The suggestions we make and the flaws we point out are only in the interest of a fair, unbiased benchmark. We hope our revisions in Section 3.1 alleviate your criticism.
>
> > I would appreciate some more intuition regarding why Recall consistency is a good thing, why the existing metric fails and how the new one avoids the failure.
>
> We added some more text to the section expanding the example and overhauled the corresponding figure to add to the intuition of recall consistency in the revised version.
>
> [1] Zahra Zamanzadeh Darban, Geoffrey I Webb, Shirui Pan, Charu C Aggarwal, and Mahsa Salehi. Deep learning for time series anomaly detection: A survey, arXiv preprint arXiv:2211.05244, 2022.
>
> [2] Ane Blázquez-García, Angel Conde, Usue Mori, and Jose A Lozano. A review on outlier/anomaly detection in time series data. ACM Computing Surveys (CSUR), 54(3):1–33, 2021.
>
> [3] Lukas Ruff, Jacob R. Kauffmann, Robert A. Vandermeulen, Grégoire Montavon, Wojciech Samek, Marius Kloft, Thomas G. Dietterich, and Klaus-Robert Müller. A unifying review of deep and shallow anomaly detection. Proc. IEEE, 109(5):756–795, 2021.

---

### Review · Reviewer_CeqJ · 2023-01-27

**Summary Of Contributions:**

In this work, the authors conduct a critical analysis of several of the most widely-used multivariate time-series anomaly detection datasets. In doing so, they identify a number of issues, and identify datasets which suffer the least from such issues as candidates for method evaluation for future research. The work also proposes a new evaluation metric for time-series anomaly detection, which is more intuitive as it is recall-consistent and more holistically evaluates approaches as it takes temporal correlations into account. Lastly, the authors critically analyze and standardize existing evaluation protocols to provide TimeSeAD, a large benchmark of deep, multivariate time-series anomaly detection methods.

**Audience:**

Yes

**Broader Impact Concerns:**

None identified

**Claims And Evidence:**

No

**Requested Changes:**

- Section 3.1 is very important to the overall message of the work, but as-written it requires significant changes to make its suggestions more objective and actionable for time-series dataset creators (see weaknesses section). This is the primary reason that I have indicated that some of the claims in this work are not clearly supported by evidence.


**Strengths And Weaknesses:**

Strengths:

- Large, thorough, and sound empirical analyses of time series datasets, metrics, and evaluation protocols are very useful for fair, single-source-of-truth comparisons across a wide variety of methods in the field. The authors should be commended for conducting such a large-scale suite of experiments across methods and datasets, and releasing the code for general use. This work also provides a great example for future researchers in this field (and others) to see how fair evaluations should be performed.

- The suggestions made in Section 4, while opinionated, appear to be sound and generally provide a path towards harmonizing the way practitioners evaluate and compare methods.

- Standardized implementations of time-series AD methods as well as fast versions of time-series precision and recall are provided for widespread adoption and reuse.

- The paper is generally clear and mostly easy to understand, albeit with some remaining typos/grammatical inconsistencies in the writing (see weaknesses section).

- Sections 3.2 and 3.3 provide a helpful and comprehensive overview of outstanding issues with time-series evaluation metrics and protocols.

- The discussion section (Section 5) outlines several important and reasonable directions for future research.

- A thorough and detailed appendix is included that expands upon most of the content in the paper for the interested reader (and for reproducibility).

Weaknesses:

- Some of the language used in the paper to refer to prior works is quite strong (see examples below) and may not convey the necessary nuance to readers who are not intimately familiar with the referenced works. The criticism may indeed be valid, but making such statements more precise will improve understanding of where exactly prior work is lacking.
    - In the abstract: "e.g., due to questionable anomaly labels" - It might be more precise to replace the word "questionable" with "incorrect" or rephrase the statement as "e.g., due to many anomaly labels being false positives" or similar.
    - In Section 1: "have not caught on in the community, primarily due to their complexity and the counterintuitive results they can produce." - Is it fair to say that a method introduced just last year has not yet caught on in the community? It takes time for ideas to propagate and be widely used.
    - In Section 6: "Many datasets are severely flawed and form a shaky foundation for AD evaluations" - I appreciate the passion by which the authors advocate for flawless datasets in this field, however, perhaps tempering such a statament as "Many datasets contain structural flaws which make it difficult to fairly compare and evaluate AD methods." or something similar might be better-received by a general audience.

- The related work section provides a brief history of recent AD approaches and their evaluation. I wonder if it is perhaps too recent, however, since time-series analysis has existed for decades now and there may be prior AD works from before 2015 that are worth inclusion and discussion.

- This is not a major weakness, but there are numerous minor grammatical inconsistencies and typos in the paper. Another round of fine-grained editing would help improve the readability of this work (which is already generally quite clear and easy to understand). Some examples include:
    - When "widely used" is an adjective, it should be hyphenated (widely-used).
    - In Section 3: "frequently used" -> "frequently-used"
    - In Section 3: "most commonly used datasets.These datasets" -> "most commonly-used datasets. These datasets"
    - The same comment as above in Section 3.2: "random predictions.Evaluation on time-series data" as well as "prediction separately.As a consequence"

- In Section 3.1: "A good dataset for evaluating anomaly detection methods should contain enough samples to train particularly data-hungry deep models." - This is quite the opinionated statement, and I do not think it is wise to tie the quality of a dataset to which methods can be applied to it. In a broad sense, real-world datasets should reflect a snapshot of phenomena in the real world. Accordingly, measures of quality such as data resolution and frequency are much more important than raw quantity as it relates to current methods.

- In Section 3.1: "We think a good multivariate time-series dataset should have at least more than ten features to allow for complex inter-feature dependencies." - Why not 3? Why not 5? This number seems very subjectively determined and it would be a much stronger statement if there were some evidence for it. Further, the number of features does not necessarily determine how complex the inter-feature dependencies are; the underlying system / temporal process determines that.

- In Section 3.1: "Ideally, anomaly density should be less than 5%." Why? Some real-world scenarios have anomalies more than 10% of the time. As an example, more than 10% of the USA suffers from diabetes [1], so a time-series dataset containing the blood-glucose levels over time of a random sample of the US population would contain abnormally high values 10% of the time. An example of a more precise statement would be that anomaly density should accurately reflect the frequency of the anomalous phenomena in the chosen setting, not more or less, or something similar.

- Long anomalies are highlighted as a problem in Section 3.1, however, it really should be the onus of the methods being evaluated to handle this. Long anomalies may simply be an inherent quality of the types of anomalies considered.

- Figure 5: "providing an ordering aligned with our intuition." This is a subjective statement, and likely does not need to be said.

[1] https://www.niddk.nih.gov/health-information/health-statistics/diabetes-statistics

---

> ### Author Response · Authors · 2023-02-09
> **Reply to intial review of Reviewer Ceqj**
>
> Thank you for your constructive comments! We will address your individual points below:
>
> >[...] there are numerous minor grammatical inconsistencies and typos in the paper.
>
> We would like to thank you for pointing out the typos, however, common style guides [1-4] advise against using a hyphen between an adverb ending in "-ly" and an adjective in a modifier. Hence, we addressed any inconsistencies by not using a hyphen in these cases. We fixed the missing spaces between sentences in the revised version. Furthermore, we will carefully read the paper for the camera-ready version to correct any remaining mistakes we have missed.
>
> >Some of the language used in the paper to refer to prior works is quite strong (see examples below) and may not convey the necessary nuance to readers who are not intimately familiar with the referenced works.
>
> We have updated the text throughout the paper to soften our tone and address any writing related issues raised by you and other reviewers.
>
> >In the abstract: "e.g., due to questionable anomaly labels" - It might be more precise to replace the word "questionable" with "incorrect" or rephrase the statement as "e.g., due to many anomaly labels being false positives" or similar.
>
> Without explicit domain knowledge, it is impossible to say if labels are wrong with absolute certainty. Based on our analysis we strongly suspect erroneous labels in multiple datasets. Thus, we try to avoid strong wording such as "incorrect" where appropriate. Replacing "questionable" with "possibly erroneous" could fit better here.
>
> >In Section 1: "have not caught on in the community, primarily due to their complexity and the counterintuitive results they can produce." - Is it fair to say that a method introduced just last year has not yet caught on in the community? It takes time for ideas to propagate and be widely used.
>
> This particular sentence mostly refers to metrics and ideas that have been around since 2015 [5] and were later extended in 2018 [6]. We think this provides enough time to be adapted in the community. For clarity, we include the corresponding references in the revised version alongside the previous reference to another recently published paper analyzing metrics that makes a similar claim [7].
>
> >The related work section provides a brief history of recent AD approaches and their evaluation. I wonder if it is perhaps too recent, however, since time-series analysis has existed for decades now, and there may be prior AD works from before 2015 that are worth inclusion and discussion.
>
> In our work, we focus on deep-learning based methods on multivariate time-series data. We conducted an extensive literature search and are not aware of any papers before 2015 that use neural networks for TSAD on multivariate data. Please let us know what kind of work you have in mind. We will gladly include any significant references we might have missed.
>
> >Comments on Section 3.1
>
> We overhauled the introduction to this section in the revised version and removed any explicit recommendations to avoid confusion. All recommendations for specific statistics of a benchmark dataset were only intended to apply to a perfect dataset for benchmarking purposes, not to all datasets in general. Note that we do not exclude any dataset based on the anomaly density or long anomalies. However, these factors have a nontrivial impact on evaluations and should be included consciously, which we pointed out in the paper: "Although not inherently negative, both effects should be kept in mind when using data containing long anomaly windows for evaluation". Thus, we still think discussing them is important overall.
>
> [1] The Chicago Manual of Style: https://ophi.org.uk/wp-content/uploads/CMS_list.pdf
>
> [2] The Guardian: https://www.theguardian.com/guardian-observer-style-guide-h
>
> [3] Queen's English Society: https://queens-english-society.org/the-hyphen-puzzle/
>
> [4] Grammarly: https://www.grammarly.com/blog/hyphen/
>
> [5] Alexander Lavin and Subutai Ahmad. Evaluating real-time anomaly detection algorithms–the numenta anomaly benchmark. In 2015 IEEE 14th international conference on machine learning and applications (ICMLA), pp. 38–44. IEEE, 2015.
>
> [6] Nesime Tatbul, Tae Jun Lee, Stan Zdonik, Mejbah Alam, and Justin Gottschlich. Precision and recall for time series. Advances in neural information processing systems, 31, 2018.
>
> [7] Alexis Huet, Jose Manuel Navarro, and Dario Rossi. Local evaluation of time series anomaly detection algorithms. In Proceedings of the 28th ACM SIGKDD Conference on Knowledge Discovery and Data Mining, pp. 635–645, 2022.

---

### Review · Reviewer_t5zx · 2023-02-06

**Summary Of Contributions:**

This paper critically reviews various existing anomaly detection evaluation techniques. The paper's scope is limited to point-wise anomaly scoring, but the authors' implicit assumption is that anomaly detection is made with a sliding window. After a lot of qualitative criticisms, the authors bring up a metric called TRec without a clear definition/justification. The authors propose an ad hoc metric in Sec 4.2, and claim it has desirable properties.

**Audience:**

Yes

**Broader Impact Concerns:**

The tone of the paper may be inappropriate.

**Claims And Evidence:**

No

**Requested Changes:**

I wouldn't say I like the overly sensationalized tone of this paper. It is just not necessary. Given the serious gap between the authors' bold claims and the lack of rigorous analysis, partial correction may not suffice to clear the bar of acceptance.

**Strengths And Weaknesses:**

**Strength**
- Provides a summary of recent deep-learning-based anomaly detection works.
- Provides a summary of popular benchmark datasets and their issues.


**Weakness**
- The problem setting is not well designed. Apparently, the authors assume windows-based anomaly detection scenarios alone, which makes the authors' claim of being the most comprehensive benchmark questionable.
- Serious lack of discussions of the existing statistical anomaly detection approaches. The authors seem to justify that omission by saying, "We focus on deep-learning-based methods," but the authors' main interest is anomaly detection METRICs, which must be model-agnostic.
- Lack of scientific rigor when comparing with existing methods. Section 3.2 is full of very basic or vague descriptions and criticisms without clear definitions/justifications. As modern anomaly detection theories are predominantly built upon rigorous statistical grounds, if the authors claim that everything but the present work is flawed, they probably need to start with basic definitions in legitimate statistical language.
- The authors do not seem to understand the basic nature of anomaly detection, which is extreme diversity in anomalies. Anomalies are defined as not being normal.  This main feature makes it very challenging to define anomalies algorithmically. One dataset or algorithm would understandably cover a few tiny portions of the entire possibilities. This is the main reason why they turned to the statistical (or information-theoretic) approach; It is an expression of their scientific conscience. The authors cannot just ignore it.

---

> ### Author Response · Authors · 2023-02-09
> **Reply to initial review of Reviewer t5zx**
>
> Thank you for your comments! We will address your points below:
>
> > [...] the authors bring up a metric called TRec without a clear definition/justification.
>
> We provide the definition of TRec in equation (1) and dedicate multiple paragraphs to the discussion of its advantages and shortcomings. In our opinion, this constitutes a clear definition and ample justification. Could you please explain what you think is missing to help us improve the paper and clear any misunderstandings?
>
> >Apparently, the authors assume windows-based anomaly detection scenarios alone, which makes the authors' claim of being the most comprehensive benchmark questionable.
>
> The anomaly detection setting we consider is by far the most common setting in the literature (see references in the paper): An anomaly detector assigns each time step a label, usually based on a thresholded score. Windowing is a common practical consideration employed for computational reasons. As such, it does not conflict with the general anomaly detection setting we consider. Some methods considered in our analysis work only with windowing, for example [1]. Thus, to ensure a fair evaluation, we use windowing in our evaluation across all methods.
>
> >Serious lack of discussions of the existing statistical anomaly detection approaches. The authors seem to justify that omission by saying,
> "We focus on deep-learning-based methods," but the authors' main interest is anomaly detection METRICs, which must be model-agnostic.
>
> The metric we propose can be applied to multivariate and univariate time-series anomaly detection with shallow and deep methods alike. However, the goals of this paper are threefold:
> - We examine existing multivariate datasets, identify the most promising candidates, and provide guidelines for constructing future benchmark datasets.
> - We examine existing metrics and propose a novel metric to address their flaws.
> - We examine various evaluation protocols proposed over the years and provide the TimeSeAD library, including a standard evaluation protocol dedicated toward an unbiased benchmark.
>
> All three goals are equally important to our work. Since we settled on examining multivariate datasets and the methods evaluated thereon, we think the focus on deep-learning based methods is justified by their abundance in the literature (see references in the paper) and their success in other complex multivariate domains [2]. With almost 30 methods considered, we provide the most extensive comparison of methods in this field to date. Evaluation and comparison to shallow baselines would exceed the scope of this work and will be left to future work, which we point out in the paper.
>
> >Section 3.2 is full of very basic or vague descriptions and criticisms without clear definitions/justifications.
>
> Introducing all related methods in detail beyond the scope of this work. It would only distract from the main point of this section: Identifying and explaining various flaws in existing metrics and providing a brief survey of metrics with respect to these flaws. Thus, we rely on the interested reader to refer to the numerous references we provide for the details of particular methods. We define the most relevant metric to our work (TRec) rigorously to provide the necessary context for our contribution. Could you please elaborate on the criticisms you deem unjustified?
>
> [1] Julien Audibert, Pietro Michiardi, Frédéric Guyard, Sébastien Marti, and Maria A Zuluaga. Usad: Unsupervised anomaly detection on multivariate time seriesammann2020anomaly. In Proceedings of the 26th ACM SIGKDD International Conference on Knowledge Discovery & Data Mining, pp. 3395–3404, 2020.
>
> [2] Chalapathy, Raghavendra, and Sanjay Chawla. "Deep learning for anomaly detection: A survey." arXiv preprint arXiv:1901.03407 (2019).

---

> > ### Author Response · Authors · 2023-02-09
> > **Continutation**
> >
> > >As modern anomaly detection theories are predominantly built upon rigorous statistical grounds, if the authors claim that everything but the present work is flawed, they probably need to start with basic definitions in legitimate statistical language.
> >
> > What modern work are you referring to?
> >
> > >As modern anomaly detection theories are predominantly built upon rigorous statistical grounds, if the authors claim that everything but the present work is flawed, they probably need to start with basic definitions in legitimate statistical language. The authors do not seem to understand the basic nature of anomaly detection, which is extreme diversity in anomalies. Anomalies are defined as not being normal. This main feature makes it very challenging to define anomalies algorithmically. One dataset or algorithm would understandably cover a few tiny portions of the entire possibilities.
> >
> > We do not understand how your comment relates to the presented work. What is the issue you are addressing here? For the exact reason you describe, we do not give a precise definition of anomalies in the presented work.
> >
> > >I wouldn't say I like the overly sensationalized tone of this paper.
> >
> > We have reworked the text throughout our paper in the revised version to make our writing more clear and more precise.

---

### Decision · Action_Editors · 2023-03-27

**Recommendation:** Accept as is

**Comment:**

The earlier version was written in a too strongly-opinionated manner, which was mainly criticized by all of reviewers. However, the authors did a good job in revising the manuscript, leveraging those comments.

**Audience:**

The biggest weakness of this paper might be its narrow audience. However, the paper provides a useful information for practitioners who are working on anomaly detection.

**Claims And Evidence:**

This paper addresses practically significant issues in multivariate time-series anomaly detection, including datasets, evaluation metric, and evaluation protocol. A new evaluation metric, which takes temporal correlations into account is presented. Moreover, it standardize existing evaluation protocols to provide a comprehensive library, 'TimeSeAD'.